# THE MARGINAL VALUE OF MOMENTUM FOR SMALL LEARNING RATE SGD

**Runzhe Wang[1], Sadhika Malladi[1], Tianhao Wang[2], Kaifeng Lyu[1], Zhiyuan Li[34]**

[1]Princeton University, [2]Yale University, [3]Stanford University,
[4]Toyota Technological Institute at Chicago
`{runzhew,smalladi,klyu}@princeton.edu`,
`tianhao.wang@yale.edu, zhiyuanli@ttic.edu`

## ABSTRACT

Momentum is known to accelerate the convergence of gradient descent in strongly convex settings without stochastic gradient noise. In stochastic optimization, such as training neural networks, folklore suggests that momentum may help deep learning optimization by reducing the variance of the stochastic gradient update, but previous theoretical analyses do not find momentum to offer any provable acceleration. Theoretical results in this paper clarify the role of momentum in stochastic settings where the learning rate is small and gradient noise is the dominant source of instability, suggesting that SGD with and without momentum behave similarly in the short and long time horizons. Experiments show that momentum indeed has limited benefits for both optimization and generalization in practical training regimes where the optimal learning rate is not very large, including small- to medium-batch training from scratch on ImageNet and fine-tuning language models on downstream tasks.

## 1 INTRODUCTION

In modern deep learning, it is standard to combine stochastic gradient methods with *heavy-ball momentum*, or *momentum* for short, to enable a more stable and efficient training of neural networks (Sutskever et al., 2013). The simplest form is *Stochastic Gradient Descent with Momentum* (SGDM). SGDM aims to minimize the training loss $\mathcal{L}(\boldsymbol{x})$ given a noisy gradient oracle $\mathcal{G}(\boldsymbol{x})$, which is usually realized by evaluating the gradient at a randomly sampled mini-batch from the training set. Specifically, let $\gamma, \beta$ be the learning rate and momentum coefficient, then SGDM can be stated as:

$$\boldsymbol{g}_k \sim \mathcal{G}(\boldsymbol{x}_k), \qquad \boldsymbol{m}_{k+1} = \beta\boldsymbol{m}_k + \boldsymbol{g}_k, \qquad \boldsymbol{x}_{k+1} = \boldsymbol{x}_k - \gamma\boldsymbol{m}_{k+1}, \tag{1}$$

where $\boldsymbol{g}_k, \boldsymbol{m}_k, \boldsymbol{x}_k$ are the gradient, momentum buffer, and parameter vector at step $k$.

For typical choices of $\beta \in (0, 1)$, the momentum buffer can be interpreted as an exponential moving average of past gradients, i.e., $\boldsymbol{m}_k = \sum_{j=0}^{k} \beta^{k-j} \boldsymbol{g}_j$. Based on this interpretation, Polyak (1964; 1987); Rumelhart et al. (1987) argued that momentum is able to cancel out oscillations along high-curvature directions and add up contributions along low-curvature directions. More concretely, for strongly convex functions without any noise in gradient estimates, Polyak (1964; 1987) showed that adding momentum can stabilize the optimization process even when the learning rate is so large that can make vanilla gradient descent diverge, and thus momentum accelerates the convergence to minimizers by allowing using a larger learning rate.

In deep learning, however, the random sampling of mini-batches inevitably introduces a large amount of stochastic gradient noise, which sometimes dominates the true gradient and may become the main source of training instability. As the above convergence results solely analyze the noiseless case, it remains unclear in theory whether momentum can likewise stabilize the stochastic optimization process in deep learning.

To understand the benefit of momentum in stochastic optimization, several prior studies (Bottou et al., 2018; Defazio, 2020; You et al., 2020) speculate that averaging past stochastic gradients through

momentum may reduce the variance of the noise in the parameter update, thus making the loss decrease faster. To approach this more rigorously, Cutkosky and Orabona (2019) proposed a variant of SGDM that provably accelerates training by leveraging the reduced variance in the updates.

Nevertheless, for SGDM without any modifications, past theoretical analyses in the stochastic optimization of convex and non-convex functions typically conclude with a convergence rate that is comparable to that of vanilla SGD, but not faster (Yan et al., 2018; Yu et al., 2019; Liu et al., 2020; Sebbouh et al., 2021; Li et al., 2022a). Besides, there also exist simple and concrete instances of convex optimization where momentum does not speed up the convergence rate of SGD, even though it is possible to optimize faster with some variants of SGDM (Kidambi et al., 2018). This naturally raises the following question on the true role of momentum:

*Does noise reduction in SGDM updates really benefit neural network training?*

To address this question, this paper delves into the training regime where the learning rate is small enough to prevent oscillations along high-curvature directions, yet the gradient noise is large enough to induce instability. This setting enables us to concentrate exclusively on the interplay between momentum and gradient noise. More importantly, this training regime is of practical significance as in many situations, such as small-batch training from scratch or fine-tuning a pre-trained model, the optimal learning rate is indeed relatively small (Liu et al., 2019; Malladi et al., 2023).

**Main Contributions.** In this paper, we present analyses of the training trajectories of SGD with and without momentum, in the regime of small learning rate. We provide theoretical justifications of a long-held belief that SGDM with learning rate $\gamma$ and momentum $\beta$ performs comparably to SGD with learning rate $\eta = \frac{\gamma}{1-\beta}$ (Tugay and Tanik, 1989; Orr, 1996; Qian, 1999; Yuan et al., 2016; Smith et al., 2020). This finding offers negative evidence for the usefulness of noise reduction in momentum. Additionally, this also motivates us to reformulate SGDM in Definition 2.3 so SGDM and SGD perform comparably under the same learning rate $\eta$, which in turn simplifies our analysis.

More specifically, given a run of SGDM, we show that vanilla SGD can closely track its trajectory in the following two regimes with different time horizon:

**Regime I.** Training with SGD and SGDM for $O(1/\eta)$ steps where the scaling of gradient noise covariance can be as large as $O(1/\eta)$. Specifically, Theorem 3.5 shows that SGD and SGDM are $O(\sqrt{\eta/(1-\beta)})$-close to each other in the sense of weak approximation, where $\eta, \beta$ are the learning rate and momentum coefficient under the notation of Definition 2.3. Our analysis not only includes the classical result that both SGD and SGDM converge to Gradient Flow in $O(1/\eta)$ steps where the stochastic gradient is sampled from a bounded distribution independent of $\eta$, but also covers the regime of applying Linear Scaling Rule (Goyal et al., 2017), where one decreases the learning rate and batch size at the same rate, so the noise covariance increases inversely proportional to $\eta$, and in this case both SGD and SGDM converge to a Stochastic Differential Equation (SDE). Our results improve over previous analysis (Yuan et al., 2016; Liu et al., 2018) by avoiding underestimating the role of noise when scaling down the learning rate, and provide rigorous theoretical supports to the scaling claims in Smith et al. (2020); Cowsik et al. (2022). Technically we introduce an auxiliary dynamics $\boldsymbol{y}_k$ (Equation (15)) that bridges SGDM and SGD.

**Regime II.** Training with SGD and SGDM for $O(1/\eta^2)$ steps for overparametrized models where the minimizers of the loss connect as a manifold and after reaching such a manifold, the gradient noise propels the iterates to move slowly along it. Theorem 4.5 shows that SGD and SGDM follow the same dynamics along the manifold of minimizers and thus have the same implicit bias. The implicit bias result of SGD is due to Katzenberger (1991); Li et al. (2021b) whose analysis does not apply to SGDM because its dynamic depends non-homogeneously on $\eta$. Our proof of Theorem 4.5 is non-trivial in carefully decomposing the updates.

In Section 5, we further empirically verify that momentum indeed has limited benefits for both optimization and generalization in several practical training regimes, including small- to medium-batch training from scratch on ImageNet and fine-tuning RoBERTa-large on downstream tasks. For large-batch training, we observe that SGDM allows training with a large learning rate, in which regime vanilla SGD may exhibit instability that degrades the training speed and generalization. The observations are consistent with previous empirical studies on SGDM (Kidambi et al., 2018; Shallue et al., 2019; Smith et al., 2020). We argue that the use of a large learning rate makes the weak

approximation bound $O(\sqrt{\eta/(1-\beta)})$ loose: running SVAG (Li et al., 2021a), an SDE simulation method for both SGD and SGDM, shrinks or even eliminates the performance gain of momentum.

Finally, we highlight that our results can also have practical significance beyond just understanding the role of momentum. In recent years, the GPU memory capacity sometimes becomes a bottleneck in training large models. As the momentum buffer costs as expensive as storing the entire model, it has raised much interest in when it is safe to remove momentum (Shazeer and Stern, 2018). Our work sheds light on this question by formally proving that momentum only provides marginal values in small learning rate SGD. Furthermore, our results imply that within reasonable range of scales the final performance is insensitive to the momentum hyperparametrization, thereby provide support to save the effort in the extensive hyperparameter grid search.

## 2 PRELIMINARIES

Consider optimizing a loss function $\mathcal{L}(\boldsymbol{\theta}) = \frac{1}{\Xi}\sum_{i=1}^{\Xi}\mathcal{L}_i(\boldsymbol{\theta})$ where $\mathcal{L}_i : \mathbb{R}^d \to \mathbb{R}$ corresponds to the loss on the $i$-th sample. We use $\boldsymbol{\theta}$ to indicate parameters along a general trajectory. In each step, we sample a random minibatch $\mathcal{B} \subseteq [\Xi]$, and compute the gradient of the minibatch loss $\mathcal{L}_{\mathcal{B}}(\boldsymbol{\theta}) = \frac{1}{|\mathcal{B}|}\sum_{i\in\mathcal{B}}\mathcal{L}_i(\boldsymbol{\theta})$ to get the following noisy estimate of $\nabla\mathcal{L}(\boldsymbol{\theta})$, i.e., $\nabla\mathcal{L}_{\mathcal{B}}(\boldsymbol{\theta}) = \frac{1}{|\mathcal{B}|}\sum_{i\in\mathcal{B}}\nabla\mathcal{L}_i(\boldsymbol{\theta})$. It is easy to check that the noise covariance matrix of $\nabla\mathcal{L}_{\mathcal{B}}(\boldsymbol{\theta})$, namely $\mathbb{E}_{\mathcal{B}}(\nabla\mathcal{L}_{\mathcal{B}}(\boldsymbol{\theta}) - \nabla\mathcal{L}(\boldsymbol{\theta}))(\nabla\mathcal{L}_{\mathcal{B}}(\boldsymbol{\theta}) - \nabla\mathcal{L}(\boldsymbol{\theta}))^\top$, scales proportionally to $\frac{1}{|\mathcal{B}|}$. Motivated by this, Malladi et al. (2022) abstracts $\nabla\mathcal{L}_{\mathcal{B}}(\boldsymbol{\theta})$ as sampled from a noisy gradient oracle where the noise covariance only depends on a scale parameter.

**Definition 2.1** (NGOS, Malladi et al. (2022)). A *Noisy Gradient Oracle with Scale Parameter* (NGOS) is characterized by a tuple $\mathcal{G}_\sigma = (\mathcal{L}, \boldsymbol{\Sigma}, \mathcal{Z}_\sigma)$. For a scale parameter $\sigma > 0$, $\mathcal{G}_\sigma$ takes as input $\boldsymbol{\theta}$ and returns $\boldsymbol{g} = \nabla\mathcal{L}(\boldsymbol{\theta}) + \sigma\boldsymbol{v}$, where $\nabla\mathcal{L}(\boldsymbol{\theta})$ is the gradient of $\mathcal{L}$ at $\boldsymbol{\theta}$ and $\boldsymbol{v}$ is the gradient noise drawn from the probability distribution $\mathcal{Z}_\sigma(\boldsymbol{\theta})$ with mean zero and covariance matrix $\boldsymbol{\Sigma}(\boldsymbol{\theta})$. $\boldsymbol{\Sigma}(\boldsymbol{\theta})$ is independent of the noise scale $\sigma$. Slightly abusing the notation, we also use $\mathcal{G}_\sigma(\boldsymbol{\theta})$ to denote the distribution of $\boldsymbol{g}$ given $\sigma$ and $\boldsymbol{\theta}$.

In the above minibatch setting, we have noise scale $\sigma = \frac{1}{|\mathcal{B}|}$. Generally, we invoke NGOS with bigger $\sigma$ for smaller magnitudes of the learning rates. Such scaling is in compliance with the Linear Scaling Rule (Goyal et al., 2017) and is discussed further after Proposition 2.4. We now instantiate the SGD and SGDM trajectories under this noise oracle.

**Definition 2.2** (Vanilla SGD). Given a stochastic gradient oracle $\mathcal{G}_\sigma$, SGD with the learning rate schedule $\{\bar{\eta}_k\}$ updates the parameters $\boldsymbol{z}_k \in \mathbb{R}^d$ from initialization $\boldsymbol{z}_0$, as

$$\boldsymbol{z}_{k+1} = \boldsymbol{z}_k - \bar{\eta}_k\boldsymbol{g}_k, \qquad \boldsymbol{g}_k \sim \mathcal{G}_\sigma(\boldsymbol{z}_k). \tag{2}$$

**Definition 2.3** (SGD with Momentum/SGDM). Given oracle $\mathcal{G}_\sigma$, SGDM with the hyperparameter schedule $\{(\eta_k, \beta_k)\}$, where $\beta_k \in (0, 1)$, updates the parameters $\boldsymbol{x}_k \in \mathbb{R}^d$ from $(\boldsymbol{m}_0, \boldsymbol{x}_0)$, as

$$\boldsymbol{m}_{k+1} = \beta_k\boldsymbol{m}_k + (1 - \beta_k)\boldsymbol{g}_k, \qquad \boldsymbol{x}_{k+1} = \boldsymbol{x}_k - \eta_k\boldsymbol{m}_{k+1}, \qquad \boldsymbol{g}_k \sim \mathcal{G}_\sigma(\boldsymbol{x}_k). \tag{3}$$

Notice that the formulation of SGDM in Definition 2.3 is different from (1) that sometimes appear in previous literature. In our results, Definition 2.3 offers a more natural parameterization for the comparison between SGDM and SGD. An easy conversion is given by rewriting Equation (3) as:

$$\boldsymbol{x}_{k+1} = \boldsymbol{x}_k - \eta_k(1 - \beta_k)\boldsymbol{g}_k + \beta_k\frac{\eta_k}{\eta_{k-1}}(\boldsymbol{x}_k - \boldsymbol{x}_{k-1}).$$

Then setting $\eta_k = \frac{\gamma}{1-\beta}$ and $\beta_k = \beta$ recovers the form of (1).

Modeling the gradient noise as an NGOS gives us the flexibility to scale the noise in our theoretical setting to make the effect of noise non-vanishing in small learning rate training, as observed in Proposition 2.4, a variant of the standard gradient descent lemma for SGD.

**Proposition 2.4** (Descent Lemma for SGD). Given $\boldsymbol{z}_k$, the expected change of loss in the next step is

$$\mathbb{E}[\mathcal{L}(\boldsymbol{z}_{k+1})|\boldsymbol{z}_k] - \mathcal{L}(\boldsymbol{z}_k) =$$

$$\underbrace{-\eta\left\|\nabla\mathcal{L}(\boldsymbol{z}_k)\right\|^2}_{\text{descent force}} + \underbrace{\frac{1}{2}(\sigma\eta)^2\operatorname{tr}((\nabla^2\mathcal{L})\boldsymbol{\Sigma}(\boldsymbol{z}_k))}_{\text{noise-induced}} + \underbrace{\frac{1}{2}\eta^2(\nabla\mathcal{L}^\top(\nabla^2\mathcal{L})\nabla\mathcal{L}(\boldsymbol{z}_k))}_{\text{curvature-induced}} + o(\eta^2, (\sigma\eta)^2).$$

Proposition 2.4 highlights noise-induced and curvature-induced factors that prevent the loss to decrease. For regular loss functions and small learning rates, the following phenomenon are expected.

- In $O(\eta^{-1})$ steps, only for $\sigma = O(1/\sqrt{\eta})$, the loss is guaranteed to decrease for small $\eta$, during which the curvature-induced factor accumulates vanishing $o(1)$ impact as $\eta \to 0$. For $\sigma = \Theta(1/\sqrt{\eta})$, the noise-induced impact is on the same order as the descent force and will not be vanishing on the training curve, so noise affects the curve similarly across different learning rates. For $\sigma = o(1/\sqrt{\eta})$, the noise-induced impact will be vanishing as $\eta \to 0$.

- Assume $\text{tr}((\nabla^2 \mathcal{L})\boldsymbol{\Sigma}(\boldsymbol{z}_k))$ is non-vanishing as $\eta \to 0$. The loss plateaus at value $O(\eta\sigma^2)$ when the impacts balance each other, and the noise-induced impact is significant until $O((\sigma\eta)^{-2})$ steps of updates.

Inspired by these observations, we are interested in studying the behavior of SGDM in two regimes, for $O(\eta^{-1})$ steps of update with $\sigma \le 1/\sqrt{\eta}$, and for $O(\eta^{-2})$ steps of update with $\sigma \le 1$. The two regimes capture the common practices where people use noisy small-batch updates to train a model from scratch and then reduce the noise-induced impact after the training loss plateaus (usually by annealing the learning rate) in pursuit of a model that converges and generalizes better.

## 3 WEAK APPROXIMATION OF SGDM BY SGD IN $O(1/\eta)$ STEPS

Next, we will present our main theoretical results on SGDM with small learning rates. In this section, we show that in $O(1/\eta)$ steps, SGD approximates SGDM in the sense of Definition 3.1 for $\sigma \le 1/\sqrt{\eta}$. The next section studies SGDM over a longer training horizon ($O(1/\eta^2)$ steps) to characterize the coinciding implicit regularization effects of SGDM and SGD.

### 3.1 A WARM-UP EXAMPLE: THE VARIANCE REDUCTION EFFECT OF MOMENTUM

Intuitively, momentum makes the SGD update directions less noisy by averaging past stochastic gradients, which seems at first glance to contradict our result that the distribution of SGD and SGDM are approximately the same. However, the apparent discrepancy is a consequence that, by carrying the current gradient noise to subsequent steps, the updates of SGDM have long-range correlations .

For instance, we consider the case where the stochastic gradients are i.i.d. gaussian as $\boldsymbol{g}_k \sim \mathcal{N}(\boldsymbol{c}, \sigma^2 \boldsymbol{I})$ for a constant vector $\boldsymbol{c}$. We compare SGD and SGDM trajectories with hyperparameter $\eta_k = \eta$ and $\beta_k = \beta$, and initialization $\boldsymbol{z}_0 = \boldsymbol{x}_0$ and $\boldsymbol{m}_0 \sim \mathcal{N}(\boldsymbol{c}, \frac{1-\beta}{1+\beta}\sigma^2 \boldsymbol{I})$. The single-step updates are

$$\boldsymbol{z}_{k+1} - \boldsymbol{z}_k = -\eta \boldsymbol{g}_k \sim \mathcal{N}(-\eta\boldsymbol{c}, \eta^2\sigma^2 \boldsymbol{I}).$$

$$\boldsymbol{x}_{k+1} - \boldsymbol{x}_k = -\eta \boldsymbol{m}_{k+1} = -\eta(\beta^{k+1}\boldsymbol{m}_0 + \sum_{s=0}^{k} \beta^{k-s}(1-\beta)\boldsymbol{g}_s) \sim \mathcal{N}(-\eta\boldsymbol{c}, \frac{1-\beta}{1+\beta}\eta^2\sigma^2 \boldsymbol{I}).$$

Therefore, the variance of each single-step update is reduced by a factor of $\frac{1-\beta}{1+\beta}$, which implies larger momentum generates a smoother trajectory. However, we are usually more interested in tracking the final loss distributions induced by each trajectory. The distributions of after $k$ steps are

$$\boldsymbol{z}_k \sim \mathcal{N}(\boldsymbol{z}_0 - k\eta\boldsymbol{c}, k\eta^2\sigma^2 \boldsymbol{I});$$

$$\boldsymbol{x}_k = \boldsymbol{z}_0 - \eta\beta\frac{1-\beta^k}{1-\beta}\boldsymbol{m}_0 - \eta\sum_{s=0}^{k-1}(1-\beta^{k-s})\boldsymbol{g}_s \sim \mathcal{N}\left(\boldsymbol{z}_0 - k\eta\boldsymbol{c}, k\eta^2\sigma^2 \boldsymbol{I} - 2\beta\eta^2\sigma^2\frac{1-\beta^k}{1-\beta^2}\boldsymbol{I}\right).$$

Notice that the variance of the final endpoint is only different by $|2\beta\eta^2\sigma^2\frac{1-\beta^k}{1-\beta^2}| \le \frac{2\eta^2\sigma^2}{1-\beta^2}$, which is bounded regardless of $k$. The variance of $\boldsymbol{x}_k$ is increased at rate $\eta^2\sigma^2$ per step, which is significantly larger than the per step update variance $\frac{1-\beta}{1+\beta}\eta^2\sigma^2$. This is a consequence of the positive correlation of momentum updates that contributes to the variance of the trajectory in total.

Furthermore, we can observe that SGD and SGDM trajectories have different levels of turbulence induced by the different per step update variances. In some cases covered by our main result Theorem 3.5, the SGD and SGDM trajectories even exhibit different asymptotic behaviors in the limit

$\eta \to 0$. For instance, when $\sigma = \eta^{-1/3}$, $\beta = 1 - \eta^{3/4}$, if we track the trajectory at $k = t\eta^{-1}$ steps for constant $t > 0$, $z_k \sim \mathcal{N}(z_0 - tc, t\eta^{1/3}I)$ and $x_k \sim \mathcal{N}(z_0 - tc, t\eta^{1/3}I - 2\eta^{7/12}\frac{\beta(1-\beta^k)}{1+\beta}I)$ with vanishing variance as $\eta \to 0$. While both trajectories converge to the straight line $z_0 - tc|_{t\geq 0}$ in the limit, when we measure their total length,

$$\mathbb{E}\sum_k \|z_{k+1} - z_k\|_2 \geq t\eta^{-1/3}\mathbb{E}_{\xi \sim \mathcal{N}(0,I)}\|\xi\| \to \infty,$$

$$\mathbb{E}\sum_k \|x_{k+1} - x_k\|_2 \leq \frac{t}{\eta}\sqrt{\eta^2\|c\|^2 + \frac{\eta^{25/12}}{2 - \eta^{3/4}}d} \to t\|c\|.$$

We observe that as $\eta$ gets smaller, the SGD trajectory becomes more and more turbulent as its length goes unbounded, while the SGDM trajectory becomes more and more smooth, though they have the same limit. Consequently, the turbulence of the training curve may not faithfully reflect the true stochasticity of the iterates as a whole, and may not be indicative of the quality of the obtained model with different choices of the momentum.

## 3.2 MAIN RESULTS ON WEAK APPROXIMATIONS OF SGDM

The above warm-up example reminds us that SGD and SGDM trajectories may have different appearances that are irrelevant to the final distributions of the outcomes, which in our concern is mostly important. Therefore we need to talk about trajectory approximations in the correct mathematical metric. For our main results, we introduce the notion of weak approximations between two families of trajectories, inspired by (Li et al., 2019). We say a function $g(x) : \mathbb{R}^d \to \mathbb{R}^m$ has polynomial growth if there are constants $k_1, k_2 > 0$ such that $\|g(x)\|_2 \leq k_1(1 + \|x\|_2^{k_2})$, $\forall x \in \mathbb{R}^d$, and we say a function $g$ has all-order polynomial-growing derivatives if $g$ is $\mathcal{C}^\infty$ and $\nabla^\alpha g$ has polynomial growth for all $\alpha \geq 0$.

**Definition 3.1** (Order-$\gamma$ Weak Approximation). Two families of discrete trajectories $x_k^\eta$ and $y_k^\eta$ are weak approximations of each other, if there is $\eta_{\text{thr}} > 0$ that for any $T > 0$, any function $h$ of all-order polynomial-growing derivatives, and any $\eta \leq \eta_{\text{thr}}$, there is a constant $C_{h,T}$ independent of $\eta$ that

$$\max_{k=0,\dots,\lfloor T/\eta \rfloor} |\mathbb{E}h(x_k^\eta) - \mathbb{E}h(y_k^\eta)| \leq C_{h,T} \cdot \eta^\gamma.$$

Weak approximation implies that $x_k^\eta$ and $y_k^\eta$ have similar distributions at any step $k \leq T/\eta$ even when $k \to \infty$ as $\eta \to 0$, and specifically in the deep learning setting it implies that the two training (testing) curves are similar.

In the small learning rate cases, we use the big-$O$ notations to specify the order of magnitudes as the learning rate scale $\eta \to 0$. Consider a SGDM run with hyperparameters $\{(\eta_k, \beta_k)\}_{k\geq0}$. Let the magnitudes of the learning rates be controlled by a scalar $\eta$ as $\eta_k = O(\eta)$. Furthermore, to capture the asymptotic behaviour of the the momentum decay $\beta_k$, we set an index $\alpha \geq 0$ so that the decay rate of the momentum is controlled as $1 - \beta_k = O(\eta^\alpha)$. $\alpha = 0$ corresponds to a constant-scale decay schedule while $\alpha > 0$ corresponds to a schedule where $\beta_k$ is closer to $1$ for smaller learning rates. Formally, we introduce the following denotation.

**Definition 3.2.** A (family of) hyperparameter schedule $\{\eta_k, \beta_k\}_{k\geq1}$ is scaled by $\eta$ with index $\alpha$ if there are constants $\eta_{\max}, \lambda_{\min}$ and $\lambda_{\max}$, independent of $\eta$, such that for all $k$,

$$0 \leq \eta_k/\eta < \eta_{\max}, \quad 0 < \lambda_{\min} \leq (1 - \beta_k)/\eta^\alpha \leq \lambda_{\max} < 1.$$

We need the boundedness of the initial momentum for the SGDM trajectory to start safely.

**Assumption 3.3.** For each $m \geq 1$, there is constant $C_m \geq 0$ that $\mathbb{E}(\|m_0\|_2^m) \leq C_m$;

Following Malladi et al. (2022), we further assume that the NGOS satisfies the below conditions, which make the trajectory amenable to analysis.

**Assumption 3.4.** The NGOS $\mathcal{G}_\sigma = (\mathcal{L}, \Sigma, \mathcal{Z}_\sigma)$ satisfies the following conditions.

1. **Well-Behaved**: $\nabla\mathcal{L}$ is Lipschitz and $\mathcal{C}^\infty$-smooth; $\Sigma^{1/2}$ is bounded, Lipschitz, and $\mathcal{C}^\infty$-smooth; all partial derivatives of $\nabla\mathcal{L}$ and $\Sigma^{1/2}$ up to and including the third order have polynomial growth.

2. **Bounded Moments**: For all integers $m \geq 1$ and all noise scale parameters $\sigma$, there exists a constant $C_{2m}$ (independent of $\sigma$) such that $(\mathbb{E}_{\boldsymbol{v} \sim \mathcal{Z}_\sigma(\boldsymbol{\theta})}[\|\boldsymbol{v}\|_2^{2m}])^{\frac{1}{2m}} \leq C_{2m}(1 + \|\boldsymbol{\theta}\|_2), \forall \boldsymbol{\theta} \in \mathbb{R}^d$.

Given the above definitions, we are ready to establish our main result.

**Theorem 3.5** (Weak Approximation of SGDM by SGD). *Fix the initial point $\boldsymbol{x}_0$, $\alpha \in [0, 1)$, and an NGOS satisfying Assumption 3.4. Consider the SGDM update $\boldsymbol{x}_k^\eta$ with schedule $\{(\eta_k, \beta_k)\}_{k \geq 1}$ scaled by $\eta$ with index $\alpha$, noise scaling $\sigma \leq \eta^{-1/2}$ and initialization $(\boldsymbol{m}_0, \boldsymbol{x}_0)$ satisfying Assumption 3.3, then $\boldsymbol{x}_k^\eta$ is an order-$(1-\alpha)/2$ weak approximation (Definition 3.1) of the SGD trajectory $\boldsymbol{z}_k^\eta$ with initialization $\boldsymbol{z}_0^\eta = \boldsymbol{x}_0$, noise scaling $\sigma$ and learning rates $\bar{\eta}_k = \sum_{s=k}^\infty \eta_s \prod_{\tau=k+1}^s \beta_\tau (1 - \beta_k)$.*

*Specifically, for a constant schedule where $(\eta_k = \eta, \beta_k = \beta)$, $\bar{\eta}_k = \eta$. In this case, SGD and SGDM with the same learning rate weakly approximate each other at distance $O(\sqrt{\eta/(1 - \beta)})$.*

The theorem shows that when the learning rates has a small scale $\eta$, under reasonable momentum decay and reasonable gradient noise amplification, the outcomes obtained by SGDM and SGD are close in distribution over $O(1/\eta)$ steps. Specifically at the limit $\eta \to 0$, the outcomes will have the same distribution. Following Li et al. (2019), if $\sigma = 1/\sqrt{\eta}$, then the limiting distribution can be described by the law of the solution $\boldsymbol{X}_t$ to an stochastic differential equation (SDE):

$$d\boldsymbol{X}_t = -\lambda_t \nabla \mathcal{L}(\boldsymbol{X}_t)dt + \lambda_t \boldsymbol{\Sigma}^{1/2}(\boldsymbol{X}_t)d\boldsymbol{W}_t.$$

under brownian motion $\boldsymbol{W}_t$ and some rescaled learning rate schedule $\lambda_t$. If $\sigma \ll 1/\sqrt{\eta}$, however, the limit will in general be the solution to the gradient flow ODE $d\boldsymbol{X}_t = -\lambda_t \nabla \mathcal{L}(\boldsymbol{X}_t)dt$ and the impact of gradient noises will be vanishing.

The theorem is built upon an infinite learning rate schedule $k = 1, 2 \cdots \infty$. In the case where we wish to consider a finite schedule, we can apply the theorem after schedule extension by infinitely copying the hyperparameters at the last step. Besides, the theorem requires $\alpha \in [0, 1)$, and the approximation grows weaker as $\alpha$ approaches 1. At $\alpha = 1$, the two trajectories are no longer weak approximations of each other and have different limiting distributions. $\alpha > 1$ yields undesirable hyperparameter schedules where excessively heavy momentum usually slows down or even messes up optimization. Further details are discussed in Appendix C.2.

# 4 THE LIMIT OF SGDM AND SGD ARE IDENTICAL IN $O(1/\eta^2)$ STEPS

In this section, we follow the framework from Li et al. (2021b) to study the dynamics of SGDM when the iterates are close to some manifold of local minimizers of $\mathcal{L}$. Former analyses (e.g., Yan et al. (2018)) suggest that on regular functions, SGDM and SGD will get close to a local minimizer in $o(1/\eta^2)$ steps, at which point the loss function plateaus and the trajectory random walks near the local minimizer. If the local minimizers connect an manifold in the parameter space, then the updates accumulate into a drift inside the manifold over $O(1/\eta^2)$ steps. Li et al. (2021b) shows that under certain circumstances, the drift induces favorable generalization properties after the training loss reaches its minimum, by leading to minima with smaller local sharpness.

Therefore, by investigating this regime, we hope to detect the value of momentum in late-phase training, especially that concerning extra generalization benefits. Yet in this section, we show that when $\eta \to 0$, the limiting dynamic of SGDM admits the same form as that of SGD, suggesting that momentum provides no extra generalization benefits over at least $O(1/\eta^2)$ steps of updates.

## 4.1 PRELIMINARIES ON MANIFOLD OF LOCAL MINIMIZERS

We consider the case of optimizing an over-parameterized neural network, where usually the minimizers of the loss $\mathcal{L}$ form manifolds. Let $\Gamma$ be a region of local minimizers that SGD can reach, and we will work mathematically in $\Gamma$ to see whether adding momentum changes the dynamical behaviors.

**Assumption 4.1.** $\mathcal{L}$ is smooth. $\Gamma$ is a $(d - M)$-dimensional submanifold of $\mathbb{R}^d$ for some integer $0 \leq M \leq d$. Moreover, every $\boldsymbol{x} \in \Gamma$ is a local minimizer of $\mathcal{L}$ with $\nabla \mathcal{L}(\boldsymbol{x}) = 0$ and $\text{rank}(\nabla^2 \mathcal{L}(\boldsymbol{x})) = M$.

We consider a neighborhood $O_\Gamma$ of $\Gamma$ that $\Gamma$ is an attraction set of $O_\Gamma$ under $\nabla \mathcal{L}$. Specifically, we define the gradient flow under $\nabla \mathcal{L}$ by $\phi(\boldsymbol{x}, t) = \boldsymbol{x} - \int_0^t \nabla \mathcal{L}(\phi(\boldsymbol{x}, s))ds$ for any $\boldsymbol{x} \in \mathbb{R}^d$ and $t \geq 0$. We further define gradient projection map associated with $\nabla \mathcal{L}$ as $\Phi(\boldsymbol{x}) := \lim_{t \to \infty} \phi(\boldsymbol{x}, t)$.

**Assumption 4.2.** From any point $x \in O_\Gamma$, the gradient flow governed by $\nabla \mathcal{L}$ converges to some point in $\Gamma$, i.e., $\Phi(x)$ is well-defined and $\Phi(x) \in \Gamma$.

It can be shown that for every $x \in \Gamma$, $\partial\Phi(x)$ is the orthogonal projection onto the tangent space of $\Gamma$ at $x$. Moreover, Li et al. (2021b) proved that for any initialization $x_0 \in O_\Gamma$, a fixed learning rate schedule $\eta_k \equiv \eta$, and any $t > 0$, time-rescaled SGD iterates $z_{\lfloor t/\eta^2 \rfloor}$ converges in distribution to $Z_t$, the solution to the following SDE. We will refer to $Z_t$ as the slow SDE.

$$Z_t = \Phi(x_0) + \int_0^t \partial\Phi(Z_s)\Sigma^{1/2}(Z_s)\mathrm{d}W_s + \int_0^t \frac{1}{2}\partial^2\Phi(Z_s)[\Sigma(Z_s)]\mathrm{d}s. \tag{4}$$

Notice that $Z_t$ always stays in $\Gamma$ with $Z_0 = \Phi(x_0)$. Though $z_0 = x_0$ is not in $\Gamma$, for any $t > 0$ the limit of $z_{\lfloor t/\eta^2 \rfloor}$ will fall onto $\Gamma$.

## 4.2 ANALYSIS OF SGDM VIA THE SLOW SDE

As the limiting dynamics of SGD iterates is known as above, we are curious about whether adding momentum modifies this limit. It turns out that within a fairly free range of hyperparameters, SGDM also has the same limiting dynamics. Specifically, for a family of hyperparameters $\{(\eta_k^{(n)}, \beta_k^{(n)})\}_{k \geq 1}$ scaled by a series of scalars $\eta^{(n)}$ with index $\alpha$ (Definition 3.2, $\lim_{n \to \infty} \eta^{(n)} = 0$), we will show that if the hyperparameter schedules converge as $n \to \infty$, then SGDM iterates will also converge into the limiting dynamics of SGD with the limiting learning rates, irrelevant of the momentum decay factors.

Similar to the setting in Li et al. (2021b), we consider a fixed time rescaling $t = k(\eta^{(n)})^2$. We stipulate that the schedule $\eta_k^{(n)} \to \eta^{(n)} \cdot \lambda_t$ as $n \to \infty$ for a rescaled schedule in continuous time $\lambda : [0, T] \to \mathbb{R}^+$. In the special case $\eta_k^{(n)} \equiv \eta^{(n)}$, it is clear that $\lambda_t \equiv 1$, and the setting of Equation (4) is recovered. Formally we introduce

**Assumption 4.3.** $\lambda_t : [0, T] \to \mathbb{R}^+$ has finite variation, and

$$\lim_{n \to \infty} \eta^{(n)} \sum_{k=0}^{\lfloor T/(\eta^{(n)})^2 \rfloor} |\eta_k^{(n)} - \eta^{(n)} \cdot \lambda_{k(\eta^{(n)})^2}| = 0.$$

**Assumption 4.4** (Bounded variation). There is a constant $Q$ independent of $n$ such that for all $n$,

$$\sum_{k=1}^{\lfloor T/(\eta^{(n)})^2 \rfloor} |\eta_k^{(n)} - \eta_{k-1}^{(n)}| \leq Q\eta^{(n)}, \qquad \sum_{k=1}^{\lfloor T/(\eta^{(n)})^2 \rfloor} |\beta_k^{(n)} - \beta_{k-1}^{(n)}| \leq Q(\eta^{(n)})^\alpha$$

In this general regime, we define the slow SDE on $\Gamma$ to admit the following description:

$$X_t = \Phi(x_0) + \int_0^t \lambda_t \partial\Phi(X_s)\Sigma^{1/2}(X_s)\mathrm{d}W_s + \int_0^t \frac{\lambda_t^2}{2}\partial^2\Phi(X_s)[\Sigma(X_s)]\mathrm{d}s. \tag{5}$$

Both SGDM and SGD converge to the above slow SDE on $\Gamma$, as summarized in the following theorem.

**Theorem 4.5.** *Fix the initialization $x_0 = z_0 \in O_\Gamma$ and any $\alpha \in (0, 1)$, and suppose the initial momentum $m_0$ satisfies Assumption 3.3. For $n \geq 1$, let $\{(\eta_k^{(n)}, \beta_k^{(n)})\}_{k \geq 1}$ be any hyperparameter schedule scaled by $\eta^{(n)}$ with index $\alpha$, satisfing Assumptions 4.3 and 4.4. Fix the noise scale $\sigma^{(n)} \equiv 1$. Under Assumptions 4.1 and 4.2, consider the SGDM trajectory $\{x_k^{(n)}\}$ with schedule $\{(\eta_k^{(n)}, \beta_k^{(n)})\}$, initialization $(x_0, m_0)$, and the SGD trajectory $\{z_k^{(n)}\}$ with schedule $\{\eta_k^{(n)}\}$, initialization $z_0 = x_0$. Suppose the slow SDE defined in (5) has a global solution $\{X_t\}_{t \geq 0}$, then as $n \to \infty$ with $\eta^{(n)} \to 0$, both $x_{\lfloor t/(\eta^{(n)})^2 \rfloor}^{(n)}$ and $z_{\lfloor t/(\eta^{(n)})^2 \rfloor}^{(n)}$ converge in distribution to $X_t$.*

The proof of Theorem 4.5 is inspired by Calzolari and Marchetti (1997). Similarly in this regime, the momentum process $m_k^{(n)}$ behaves like an Uhlenbeck-Ornstein process with $O(\eta^\alpha)$ mixing variance, so the per-step variance will be significantly smaller than that of SGD as is in Section 3.1. To prove the result, a more careful expansion of the per-step change $\Phi(x_{k+1}) - \Phi(x_k)$ is needed. The proof is detailed in Appendix D.

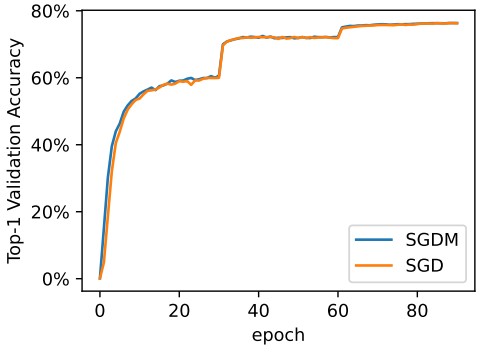 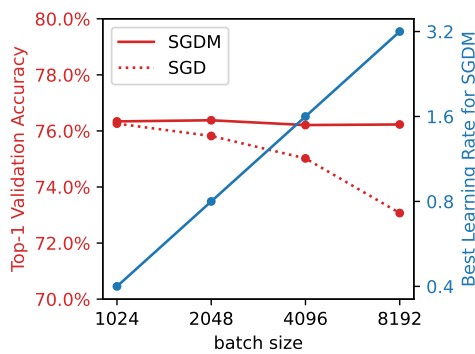

(a) SGDM and SGD with batch size 1024      (b) SGDM v.s. SGD across batch sizes

Figure 1: SGDM performs comparably to SGD in training ResNet-50 on ImageNet with smaller batch sizes (e.g., 1024), and outperforms SGD significantly at larger batch sizes.

## 5 EXPERIMENTS

In the previous sections, we conclude in theory that SGDM and SGD have similar performances within noisy short-horizon or general long-horizon training. While our theoretical results mostly work for learning rates that are asymptotically small, in this section, we verify that momentum indeed has limited benefits in practical training regimes where the optimal learning rate is finite but not very large. We defer some of the additional details of this section to the appendix.

### 5.1 MOMENTUM MAY INDEED HAVE MARGINAL VALUE IN PRACTICE

**ImageNet Experiments.** First, we train ResNet-50 on ImageNet across batch sizes. Following the experimental setup in Goyal et al. (2017), we use a learning rate schedule that starts with a 5-epoch linear warmup to the peak learning rate and decays it at epoch #30, #60, #80. For SGDM (1), we use the default value of $\beta = 0.9$, and grid search for the best learning rate $\gamma$ over $0.1 \times 2^k$ ($k \in \mathbb{Z}$). Then we check whether vanilla SGD with learning rate $\frac{\gamma}{1-\beta}$ can achieve the same performance as SGDM. Consistent with previous empirical studies (Shallue et al., 2019; Smith et al., 2020), we observed that for training with smaller batch sizes, the optimal learning rate of SGDM is small enough so that SGD can perform comparably, though SGDM can indeed outperform SGD at larger batch sizes.

**Language Model Experiments.** In fine-tuning a pre-trained model, a small learning rate is also preferable to retain the model's knowledge learned during pre-training. Indeed, we observe that SGD and SGDM behave similarly in this case. We fine-tune RoBERTa-large (Liu et al., 2019) on 5 diverse tasks (SST-2 (Socher et al., 2013), SST-5 (Socher et al., 2013), SNLI (Bowman et al., 2015), TREC (Voorhees and Tice, 2000), and MNLI (Williams et al., 2018)) using SGD and SGDM. We follow the few shot setting described in (Gao et al., 2021; Malladi et al., 2023), using a grid for SGD based on (Malladi et al., 2023) and sampling 512 examples per class (Table 1). Additional settings and trajectories are in Appendix E.

Table 1: SGD and SGDM for fine-tuning RoBERTa-large on 5 tasks using 512 examples from each class (Gao et al., 2021; Malladi et al., 2023). Results are averaged over 5 random subsets of the full dataset. These findings confirm that SGD and SGDM approximate each other in noisy settings.

| Task | **SST-2** | **SST-5** | **SNLI** | **TREC** | **MNLI** |
|------|-----------|-----------|----------|----------|----------|
| Zero-shot | 79.0 | 35.5 | 50.2 | 51.4 | 48.8 |
| SGD | 94.0 (0.4) | 55.2 (1.1) | 87.7 (0.3) | 97.2 (0.2) | 84.0 (0.3) |
| SGDM | 94.0 (0.5) | 55.0 (1.0) | 88.4 (0.6) | 97.2 (0.4) | 83.7 (0.8) |

### 5.2 INVESTIGATING THE BENEFIT OF MOMENTUM IN LARGE-BATCH TRAINING

The ImageNet experiments demonstrate that momentum indeed offers benefits in large-batch training when the optimal learning rate is relatively large. We now use large-batch experiments on CIFAR-10

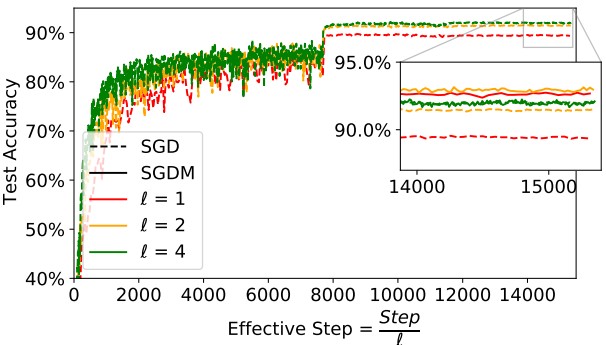

Figure 2: Standard SGDM achieves higher test performance than SGD (see $\ell = 1$), but the two trajectories get closer when reducing the curvature-induced term with SVAG (i.e., increasing the value of $\ell$, see Definition 5.1 and Lemma 2.4). These experiments confirm our theoretical findings that SGD and SGDM approximate each other when the gradient noise is the primary source of instability. We use batch size $B = 512$ with two learning rate decays by a factor of $0.1$ at epochs $80$ and $120$. We grid search to find the best learning rate for SGDM ($\eta = 0.2$) and then use it to run SGD and SGDM with SVAG. We use $\beta = 0.9$ for SGDM. Additional experimental details are in the appendix.

to provide empirical evidence that this benefit is not due to the noise reduction effect and is marginal when SGD is well-approximated by its SDE. To do this we apply SVAG (Li et al., 2021a) to control the noise scale in the gradient oracle in both SGDM and SGD updates.

**Definition 5.1** (SVAG). With any $\ell > 0$, SVAG transforms the NGOS $\mathcal{G}_\sigma = (f, \mathbf{\Sigma}, \mathcal{Z}_\sigma)$ (Definition 2.1) into another NGOS $\widehat{\mathcal{G}}_{\sqrt{\ell}\sigma} = (f, \mathbf{\Sigma}, \widehat{\mathcal{Z}}_{\sqrt{\ell}\sigma})$ with scale $\sqrt{\ell}\sigma$. For an input $\boldsymbol{\theta}$, $\widehat{\mathcal{G}}_{\ell\sigma}$ returns $\widehat{\boldsymbol{g}} = r_1(\ell)\boldsymbol{g}_1 + r_2(\ell)\boldsymbol{g}_2$ where $\boldsymbol{g}_1, \boldsymbol{g}_2 \sim \mathcal{G}_\sigma(\boldsymbol{\theta})$ and $r_i(\ell) = \frac{1}{2}(1 + (-1)^i \sqrt{2\ell - 1})$. $\widehat{\mathcal{Z}}_{\sqrt{\ell}\sigma}$ is defined to ensure $\widehat{\boldsymbol{g}}$ has the same distribution as $\nabla f(\boldsymbol{\theta}) + \sqrt{\ell}\sigma\boldsymbol{z}$ when $\boldsymbol{z} \sim \widehat{\mathcal{Z}}_{\sqrt{\ell}\sigma}(\boldsymbol{\theta})$.

In our experiments, given an original SGD run with learning rate $\eta$ for $K$ steps, we perform a new SGD run with learning rate $\eta/\ell$, SVAG($\ell$) as the gradient oracle, for $K\ell$ steps. The new SGD trajectory will be closer to its SDE approximation as $\ell$ gets larger, and converge to its SDE as $\ell \to +\infty$ (Li et al., 2021a). We also apply the same modifications to SGDM runs ($\beta$ the momentum decay factor is unmodified). In another view, applying this modification makes the total accumulated noise-induced impact and descent force (Proposition 2.4) qualitatively stay on the same scale, while the accumulated curvature-induced impact is reduced by a factor of $\ell$.

We train a ResNet-32 (He et al., 2016) on CIFAR-10 (Krizhevsky et al.) with batch size $B = 512$. We first grid search to find the best learning rate for the standard SGDM ($\ell = 1$), and then we perform SGD and SGDM with that same learning rate ($\bar{\eta}_k = \eta_k = \eta$ in the formulation Equations (2) and (3)). Then we modify both processes with different $\ell$ values. The results are summarized in Figure 2. We observe that while standard SGDM outperforms standard SGD, when we increase the value of $\ell$, the two trajectories become closer until SGDM has no evident edge over SGD. The finding corroborates that the benefit of adding momentum is mostly due to the alleviation of curvature-induced impacts, but will be marginal in general small-batch or small learning-rate settings when SGD is well-approximated by SDE.

## 6 CONCLUSIONS

This work provides theoretical characterizations of the role of momentum in stochastic gradient methods. We formally show that momentum does not introduce optimization and generalization benefits when the learning rates are small, and we further exhibit empirically that the value of momentum is marginal for gradient-noise-dominated learning settings with practical learning rate scales. Hence we conclude that momentum does not provide a significant performance boost in the above cases. Our results further suggest that model performance is agnostic to the choice of momentum parameters over a range of hyperparameter scales.

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

## A  RELATED WORKS

**The role of momentum in optimization.** The accelerating effect of some variants of momentum has been observed in convex optimization (Kidambi et al., 2018) and linear regression (Jain et al., 2018) with under specialized parametrizations. Smith (2018) pointed out that momentum can help stabilize training, but the optimal choice of momentum is closely related to the choice of learning rate. Plattner (2022) later empirically established that momentum enlarges the learning rate but does not boost performance. Arnold et al. (2019) argued using a quadratic example that momentum might not reduce variance as the gradient noise in each would actually be carried over to future iterates due to momentum. Tondji et al. (2021) showed that the application of a multi-momentum strategy can achieve variance reduction in deep learning.

Defazio (2020) proposed a stochastic primal averaging formulation for SGDM which facilitates a Lyapunov analysis for SGDM, and one particular insight from their analysis is that momentum may help reduce noise in the early stage of training but is no longer helpful when the iterates are close to local minima. Xie et al. (2021) showed that under SDE approximation, the posterior of SGDM is the same as that of SGD. Jelassi and Li (2022) proved the generalization benefit of momentum in GD in a specific setting of binary classification, by showing that GD+M is able to learn small margin data from the historical gradients in the momentum. A stronger implicit regularization effect of momentum in GD is also proved in Ghosh et al. (2023). Fu et al. (2023) showed that momentum is beneficial in deferring an "abrupt sharpening" phenomenon that slows down optimization when the learning rate is large.

**Convergence of momentum methods.** Momentum-based methods do not tend to yield faster convergence rates in theory. Yu et al. (2019) showed that distributed SGDM can achieve the same linear speedup as ditributed SGD in the non-convex setting. Also in the non-convex setting, Yan et al. (2018) showed that the gradient norm converges at the same rate for SGD, SGDM and stochastic Nestrov's accelerated gradient descent, and they used stability analysis to argue that momentum helps generalization when the loss function is Lipschitz. Under the formulation of quasi-hyperbolic momentum (Ma and Yarats, 2019), Gitman et al. (2019) proposed another unified analysis for momentum methods. Liu et al. (2020) proved that SGDM converges as fast as SGD for strongly convex and non-convex objectives even without a bounded gradient assumption. Using a iterate-averaging formulation, Sebbouh et al. (2021) proved last-iterate convergence of SGDM in both convex and non-convex settings. Later, Li et al. (2022a) showed that constant momentum can lead to suboptimal last-iterate convergence rate and increasing momentum resolves the issue. Smith (2018); Liu et al. (2018) provided evidence that momentum helps escape saddle points.

**Characterizing implicit bias near manifold of local minimizers** A recent line of work has studied the implicit bias induced by gradient noise in SGD-type algorithms, when iterates are close to some manifold of local minimizers (Blanc et al., 2020; Damian et al., 2021; Li et al., 2021b). In particular, Li et al. (2021b) developed a framework for describing the dynamics of SGD via a slow SDE on the manifold of local minimizers in the regime of small learning rate (see Appendix D for an introduction). Similar methodology has become a powerful tool for analyzing algorithmic implicit bias and has been extended to many other settings, including SGD/GD for models with normalization layers (Lyu et al., 2022; Li et al., 2022b), GD in the edge of stability regime (Arora et al., 2022), Local SGD (Gu et al., 2023), sharpness-aware minimization (Wen et al., 2022), and pre-training for language models (Liu et al., 2022). Notably, Cowsik et al. (2022) utilized the similar idea to study the slow SDE of SGDM study the optimal scale of the momentum parameter with respect to the learning rate, which has a focus different from our paper.

## B  ADDITIONAL PRELIMINARIES

### B.1  SGDM FORMULATIONS

Recall Definition 2.3 where we defined SGDM as following:

$$\boldsymbol{m}_{k+1} = \beta_k \boldsymbol{m}_k + (1 - \beta_k)\boldsymbol{g}_k, \qquad \boldsymbol{x}_{k+1} = \boldsymbol{x}_k - \eta_k \boldsymbol{m}_{k+1}. \tag{6}$$

Our formulation is different from various SGDM implementations (e.g. pytorch (Paszke et al., 2019)) which admit the following form:

**Definition B.1** (Standard formulation of SGD with momentum). Given a stochastic gradient $\bar{\boldsymbol{g}}_k \sim \mathcal{G}_\sigma(\bar{\boldsymbol{x}}_k)$, SGD-Standard with the hyperparameter schedule $(\gamma_k, \mu_k, \tau_k)$ momentum updates the parameters $\bar{\boldsymbol{x}}_k \in \mathbb{R}^d$ from $(\bar{\boldsymbol{m}}_0, \bar{\boldsymbol{x}}_0)$ as

$$\bar{\boldsymbol{m}}_{k+1} = \mu_k \bar{\boldsymbol{m}}_k + (1 - \tau_k)\bar{\boldsymbol{g}}_k \tag{7}$$
$$\bar{\boldsymbol{x}}_{k+1} = \bar{\boldsymbol{x}}_k - \gamma_k \bar{\boldsymbol{m}}_{k+1} \tag{8}$$

where $\tau_k \in [0, 1)$.

Notice that with $\tau_k = 0$, $\mu_k = \beta$ and $\gamma_k = \gamma$, the standard formulation recovers Equation (1) that has commonly appeared in the literature. The standard formulation also recovers Equation (6) with $\mu_k = \tau_k = \beta_k$ and $\eta_k = \gamma_k$. Actually, as we will show in the following lemma, these formulations are equivalent up to hyperparameter transformations.

**Lemma B.2** (Equivalence of SGDM and SGD-Standard). *Let $\alpha_k$ be the sequence that $\alpha_0 = 1$ and*

$$\alpha_{k+1} = \frac{\alpha_k}{\alpha_k(1 - \tau_k) + \mu_k}.$$

*Then $0 < \alpha_{k+1} \leq \frac{1}{1-\tau_k}$. For parameters schedules and initialization*

$$\beta_k = \frac{\alpha_{k+1}}{\alpha_k}\mu_k, \eta_k = \frac{\gamma_k}{\alpha_{k+1}}, \boldsymbol{m}_0 = \bar{\boldsymbol{m}}_0, \boldsymbol{x}_0 = \bar{\boldsymbol{x}}_0$$

*then $\bar{\boldsymbol{x}}_k$ and $\boldsymbol{x}_k$ follow the same distribution. $\boldsymbol{x}_k$ is by Definition 2.3 and $\bar{\boldsymbol{x}}_k$ by Definition B.1.*

*Proof.* We shall prove that given $\boldsymbol{x}_k = \bar{\boldsymbol{x}}_k$, $\boldsymbol{g}_k = \bar{\boldsymbol{g}}_k$ and $\boldsymbol{m}_k = \alpha_k \bar{\boldsymbol{m}}_k$, there is and $\boldsymbol{m}_{k+1} = \alpha_{k+1}\bar{\boldsymbol{m}}_{k+1}$ and $\bar{\boldsymbol{x}}_{k+1} = \boldsymbol{x}_{k+1}$. This directly follows that $\alpha_{k+1}(1 - \tau_k) = 1 - \beta_k$, and

$$\begin{aligned}
\bar{\boldsymbol{m}}_{k+1} &= \mu_k \bar{\boldsymbol{m}}_k + (1 - \tau_k)\bar{\boldsymbol{g}}_k \\
&= \frac{\mu_k}{\alpha_k}\boldsymbol{m}_k + (1 - \tau_k)\boldsymbol{g}_k \\
&= \frac{1}{\alpha_{k+1}}(\beta_k \boldsymbol{m}_k + \alpha_{k+1}(1 - \tau_k)\boldsymbol{g}_k) \\
&= \frac{1}{\alpha_{k+1}}\boldsymbol{m}_{k+1}. \\
\bar{\boldsymbol{x}}_{k+1} &= \bar{\boldsymbol{x}}_k - \gamma_k \bar{\boldsymbol{m}}_{k+1} \\
&= \boldsymbol{x}_k - \eta_k \boldsymbol{m}_{k+1}.
\end{aligned}$$

Then we can prove the claim by induction that $\boldsymbol{x}_k$ and $\bar{\boldsymbol{x}}_k$ are identical in distribution. More specifically, we shall prove that the CDF for $(\boldsymbol{x}_k, \boldsymbol{m}_k, \boldsymbol{g}_k)$ at $(\boldsymbol{x}, \boldsymbol{m}, \boldsymbol{g})$ is the same as the CDF for $(\bar{\boldsymbol{x}}_k, \bar{\boldsymbol{m}}_k, \bar{\boldsymbol{g}}_k)$ at $(\boldsymbol{x}, \frac{\boldsymbol{m}}{\alpha_k}, \boldsymbol{g})$. For $k = 0$, the claim is trivial. If the induction premise holds for $k$, then let $(\boldsymbol{x}_{k+1}, \boldsymbol{m}_{k+1}, \boldsymbol{g}_{k+1})$ be the one-step iterate in Equation (6) from $(\boldsymbol{x}_k, \boldsymbol{m}_k, \boldsymbol{g}_k) = (\boldsymbol{x}, \boldsymbol{m}, \boldsymbol{g})$ and $(\bar{\boldsymbol{x}}_{k+1}, \bar{\boldsymbol{m}}_{k+1}, \bar{\boldsymbol{g}}_{k+1})$ be the one-step iterate in Definition B.1 from $(\bar{\boldsymbol{x}}_k, \bar{\boldsymbol{m}}_k, \bar{\boldsymbol{g}}_k) = (\boldsymbol{x}, \frac{\boldsymbol{m}}{\alpha_k}, \boldsymbol{g})$, then we know from above that $\alpha_{k+1}\bar{\boldsymbol{m}}_{k+1} = \boldsymbol{m}_{k+1}$ and $\bar{\boldsymbol{x}}_{k+1} = \boldsymbol{x}_{k+1}$, and therefore the conditional distribution for $(\boldsymbol{x}_{k+1}, \boldsymbol{m}_{k+1}, \boldsymbol{g}_{k+1})$ is the same as that for $(\bar{\boldsymbol{x}}_{k+1}, \alpha_{k+1}\bar{\boldsymbol{m}}_{k+1}, \bar{\boldsymbol{g}}_{k+1})$. Integration proves the claim for $k + 1$, and thereby the induction is complete. $\qquad\square$

Therefore, in the rest part of the appendix, we will stick to the formulation of SGDM as Equation (6) unless specified otherwise.

## B.2    DESCENT LEMMA FOR SGD AND SGDM

In Proposition 2.4, given the SGD updates $\{z_k\}$ with constant learning rate $\eta$, we have the gradient descent lemma

$$\mathbb{E}[\mathcal{L}(\boldsymbol{z}_{k+1})|\boldsymbol{z}_k] - \mathcal{L}(\boldsymbol{z}_k) = \tag{9}$$

$$\underbrace{-\eta \|\nabla\mathcal{L}(\boldsymbol{z}_k)\|^2}_{\text{descent force}} + \underbrace{\frac{1}{2}(\sigma\eta)^2 \operatorname{tr}((\nabla^2\mathcal{L})\boldsymbol{\Sigma}(\boldsymbol{z}_k))}_{\text{noise-induced}} + \underbrace{\frac{1}{2}\eta^2(\nabla\mathcal{L}^\top(\nabla^2\mathcal{L})\nabla\mathcal{L}(\boldsymbol{z}_k))}_{\text{curvature-induced}} + o(\eta^2, (\sigma\eta)^2).$$

$$\tag{10}$$

We can get a similar result for the SGDM updates $\{x_k\}$, namely, when $1 - \beta \ll \eta$,

$$\mathbb{E}\mathcal{L}(\boldsymbol{x}_{k+1}) = \mathbb{E}\mathcal{L}(\boldsymbol{x}_k) - \eta\mathbb{E}\|\nabla\mathcal{L}(\boldsymbol{x}_k)\|^2 \tag{11}$$

$$+ \frac{1}{2}\eta^2\mathbb{E}(\nabla\mathcal{L}^\top(\nabla^2\mathcal{L})\nabla\mathcal{L}(\boldsymbol{x}_k) + \frac{1}{2}(\sigma\eta)^2 \operatorname{tr}((\nabla^2\mathcal{L})\boldsymbol{\Sigma}(\boldsymbol{x}_k))) \tag{12}$$

$$+ \frac{\eta\beta}{1-\beta}(S_{k+1} - S_k) + \eta^2\mathbb{E}[\boldsymbol{m}_k^\top\nabla^2\mathcal{L}\nabla\mathcal{L}(\boldsymbol{x}_k)] + o(\eta^2/(1-\beta), (\sigma\eta)^2/(1-\beta)). \tag{13}$$

where $S_k = \mathbb{E}[\nabla\mathcal{L}(\boldsymbol{x}_k)^\top\boldsymbol{m}_k - \frac{\eta}{2}\boldsymbol{m}_k^\top\nabla^2\mathcal{L}(\boldsymbol{x}_k)\boldsymbol{m}_k]$. Notice that beside the terms in the SGD updates, additions are an $O(\eta/(1-\beta))$-order telescoping term and an $O(\eta^2)$ extra impact term. The impact of the telescoping term is large for per-step updates but is limited across the whole trajectory, so the difference between SGDM and SGD trajectory is actually depicted by the extra term $\eta^2\mathbb{E}[\boldsymbol{m}_k^\top\nabla^2\mathcal{L}\nabla\mathcal{L}(\boldsymbol{x}_k)]$. In the settings where the extra term is bounded, we can bound the difference between training curves of SGDM and SGD.

*Brief Proof for proposition 2.4.* We assume $\|\nabla^n\mathcal{L}\|$ is bounded ($\|\nabla^n\mathcal{L}\|_\infty < C_n$) for any order of derivatives $n = 0, 1, 2, 3 \cdots$ and the trajectories are bounded ($\mathbb{E}\|\boldsymbol{z}_k\|^m, \mathbb{E}\|\boldsymbol{x}_k\|^m \leq C_m$) to simplify the reasoning. In the SGD case, $z_{k+1} - z_k = -\eta g_k$ has moments

$$\mathbb{E}\|z_{k+1} - z_k\|^{2m} = \eta^{2m}\mathbb{E}\|g_k\|^{2m}$$
$$\leq \eta^{2m}2^{2m}\mathbb{E}(\|\nabla\mathcal{L}(z_k)\|^{2m} + \sigma^{2m}\|\boldsymbol{v}_k\|^{2m})$$
$$= O(\eta^{2m} + (\sigma\eta)^{2m}).$$

Similarly,

$$\mathbb{E}\|x_{k+1} - x_k\|^{2m} = \eta^{2m}\mathbb{E}\|m_{k+1}\|^{2m}$$
$$= \eta^{2m}\mathbb{E}\left\|\beta^{k+1}m_0 + \sum_{s=0}^{k}\beta^{k-s}(1-\beta)g_s\right\|^{2m}$$
$$\leq \eta^{2m}2^{2m}\mathbb{E}\left(\beta^{2m(k+1)}\|m_0\|^{2m} + \left(\sum_{s=0}^{k}\beta^{k-s}\right)^{2m-1}\left(\sum_{s=0}^{k}\beta^{k-s}(1-\beta)^{2m}\|g_s\|^{2m}\right)\right)$$
$$= O(\eta^{2m} + (\sigma\eta)^{2m}).$$

Therefore $\mathbb{E}\|z_{k+1} - z_k\|^3 \leq \sqrt{\mathbb{E}\|z_{k+1} - z_k\|^6} = o(\eta^2, (\sigma\eta)^2)$ and $\mathbb{E}\|x_{k+1} - x_k\|^3 \leq \sqrt{\mathbb{E}\|x_{k+1} - x_k\|^6} = o(\eta^2, (\sigma\eta)^2)$. Taylor expansion gives

$$\mathbb{E}\mathcal{L}(z_{k+1})|z_k = \mathcal{L}(z_k) + \mathbb{E}\nabla\mathcal{L}(z_k)^\top(z_{k+1} - z_k) + \frac{1}{2}\mathbb{E}(z_{k+1} - z_k)^\top\nabla^2\mathcal{L}(z_k)(z_{k+1} - z_k) + o(\eta^2, (\sigma\eta)^2).$$

Expansion yields Equation (9). Similarly,

$$\mathbb{E}\mathcal{L}(x_{k+1}) = \mathbb{E}[\mathcal{L}(x_k) + \nabla\mathcal{L}(x_k)^\top(x_{k+1} - x_k) + \frac{1}{2}(x_{k+1} - x_k)^\top\nabla^2\mathcal{L}(x_k)(x_{k+1} - x_k)] + o(\eta^2, (\sigma\eta)^2)$$

$$= \mathbb{E}[\mathcal{L}(x_k) - \eta\nabla\mathcal{L}(x_k)^\top m_{k+1} + \frac{\eta^2}{2}m_{k+1}^\top\nabla^2\mathcal{L}(x_k)m_{k+1}] + o(\eta^2, (\sigma\eta)^2)$$

Since $m_{k+1} = g_k + \frac{\beta}{1-\beta}(m_k - m_{k+1})$,

$$
\begin{aligned}
\mathbb{E}\mathcal{L}(x_{k+1}) &= \mathbb{E}[\mathcal{L}(x_k) - \eta\nabla\mathcal{L}(x_k)^\top g_k - \eta\nabla\mathcal{L}(x_k)^\top \frac{\beta}{1-\beta}(m_k - m_{k+1}) \\
&\quad + \frac{\eta^2}{2}m_{k+1}^\top \nabla^2\mathcal{L}(x_k)(g_k + \frac{\beta}{1-\beta}(m_k - m_{k+1}))] + o(\eta^2, (\sigma\eta)^2) \\
&= \mathbb{E}[\mathcal{L}(x_k) - \eta\nabla\mathcal{L}(x_k)^\top g_k - \frac{\eta\beta}{1-\beta}(\nabla\mathcal{L}(x_k)^\top m_k - \nabla\mathcal{L}(x_{k+1})^\top m_{k+1}) \\
&\quad + \frac{\eta\beta}{1-\beta}(m_{k+1}^\top \nabla^2\mathcal{L}(x_k)(x_k - x_{k+1}) + O(\eta^2, \sigma^3\eta^2)) \\
&\quad + \frac{\eta^2}{2}m_{k+1}^\top \nabla^2\mathcal{L}(x_k)(g_k + \frac{\beta}{1-\beta}(m_k - m_{k+1}))] + o(\eta^2, (\sigma\eta)^2) \\
&= \mathbb{E}[\mathcal{L}(x_k) - \eta\nabla\mathcal{L}(x_k)^\top g_k - \frac{\eta\beta}{1-\beta}(\nabla\mathcal{L}(x_k)^\top m_k - \nabla\mathcal{L}(x_{k+1})^\top m_{k+1}) \\
&\quad + \frac{\eta^2}{2}(g_k + \frac{\beta}{1-\beta}(m_k - m_{k+1}))^\top \nabla^2\mathcal{L}(x_k)(g_k + \frac{\beta}{1-\beta}(m_k + m_{k+1}))] + \frac{o(\eta^2, (\sigma\eta)^2)}{1-\beta}.
\end{aligned}
$$

Expansion yields Equation (11). $\qquad\square$

## C  THE DYNAMICS OF SGDM IN $O(1/\eta)$ TIME

In this section we will prove Theorem 3.5 as the first part of our main result. First we shall take a closer examination at our update rule in Definition 2.3. Let $\beta_{s:t} = \prod_{r=s}^t \beta_r$ be the product of consecutive momentum hyperparameters for $s \le t$, and define $\beta_{s:t} = 1$ for $s > t$, we get the following expansion forms.

**Lemma C.1.**

$$
\boldsymbol{m}_k = \beta_{0:k-1}\boldsymbol{m}_0 + \sum_{s=0}^{k-1} \beta_{s+1:k-1}(1 - \beta_s)\boldsymbol{g}_s
$$

$$
\boldsymbol{x}_k = \boldsymbol{x}_0 - \sum_{s=0}^{k-1} \eta_s\beta_{0:s}\boldsymbol{m}_0 - \sum_{s=0}^{k-1}\sum_{r=s}^{k-1} \eta_r\beta_{s+1:r}(1 - \beta_s)\boldsymbol{g}_s
$$

*Proof.* By expansion of Equation (3). $\qquad\square$

Let the coefficients $c_{s,k} = \sum_{r=s}^{k-1} \eta_r\beta_{s+1:r}(1-\beta_s)$ so that $\boldsymbol{x}_k = \boldsymbol{x}_0 - \frac{\beta_0 c_{0:k}}{1-\beta_0}\boldsymbol{m}_0 - \sum_{s=0}^{k-1} c_{s,k}\boldsymbol{g}_s$. Notice that $c_{s,k}$ is increasing in $k$ but it is always upper-bounded as $c_{s,k} \le c_{s,\infty} = \sum_{r=s}^{\infty} \eta_r\beta_{s+1:r}(1-\beta_s) \le (\sup_t \eta_t)\sum_{r=s}^{\infty}(\sup_t \beta_t)^{r-s} = \frac{\sup_r \eta_r}{1-\sup_t \beta_t}$. Notice that when we have a finite schedule, we can make any extension to infinity steps (e.g. infinitely copy the hyperparameters on the last step), and the reasonings will work for the extended schedule. In this case, though the trajectory $\boldsymbol{x}_k$ are not affected by future hyperparameters, the coefficients $c_{s,\infty}$ will depend on our extension. Furthermore, if for any $t$, $\beta_t$ and $\eta_t$ are about the same scale, then we should expect that the difference $c_{s,\infty} - c_{s,k}$ is exponentially small in $k$. Then we can hypothesize that the trajectory $\boldsymbol{x}_k$ should be close to a trajectory of the following form:

$$
\widetilde{\boldsymbol{x}}_k = \widetilde{\boldsymbol{x}}_0 - \frac{\beta_0 c_{0:\infty}}{1-\beta_0}\boldsymbol{m}_0 - \sum_{s=0}^{k-1} c_{s,\infty}\widetilde{\boldsymbol{g}}_s
$$

which is exactly a SGD trajectory given the learning rate schedule $c_{s,\infty}$.

To formalize the above thought, we define the averaged learning rate schedule

$$
\bar{\eta}_k := c_{k,\infty} = \sum_{s=k}^{\infty} \eta_s\beta_{k+1:s}(1 - \beta_k) \tag{14}
$$

And the coupled trajectory

$$\boldsymbol{y}_k := \boldsymbol{x}_k - \frac{\bar{\eta}_k \beta_k}{1 - \beta_k} \boldsymbol{m}_k \tag{15}$$

$$= \boldsymbol{x}_0 - \frac{\beta_0 \bar{\eta}_0}{1 - \beta_0} \boldsymbol{m}_0 - \sum_{s=0}^{k-1} \bar{\eta}_s \boldsymbol{g}_s. \tag{16}$$

Then there is the following transition:

$$\boldsymbol{y}_k = \boldsymbol{y}_{k-1} - \bar{\eta}_{k-1} \boldsymbol{g}_{k-1} \tag{17}$$

$\boldsymbol{y}_k$ has an interesting geometrical interpretation as the endpoint of SGDM if we cut-off all the gradient signal from step $k$. E.g., $\boldsymbol{x}_\infty$ if we set $\boldsymbol{g}_{k+1} = \boldsymbol{g}_{k+2} = \cdots = 0$ for and update the SGDM from $(\boldsymbol{x}_k, \boldsymbol{m}_k)$.

## C.1 $\alpha \in [0, 1)$

In this section we provide the proof for Theorem 3.5. Specifically, when $\alpha \in [0, 1)$ is the index for the control of the learning rate schedule (Definition 3.2), we hope to show that $\boldsymbol{y}_k$ is close to a SGD trajectory $\boldsymbol{z}_k$, defined by the following for $k \geq 0$:

$$\boldsymbol{z}_0 = \boldsymbol{x}_0 \tag{18}$$
$$\boldsymbol{h}_k \sim \mathcal{G}_\sigma(\boldsymbol{z}_k) \tag{19}$$
$$\boldsymbol{z}_{k+1} = \boldsymbol{z}_k - \bar{\eta}_k \boldsymbol{h}_k \tag{20}$$

Specifically, we use $\boldsymbol{y}_k$ as a bridge for connecting $\boldsymbol{x}_k$ and $\boldsymbol{z}_k$. Our proof consists of two steps.

1. We show that $\boldsymbol{x}_k$ and $\boldsymbol{y}_k$ are order-$(1 - \alpha)/2$ weak approximations of each other.
2. We show that $\boldsymbol{z}_k$ and $\boldsymbol{y}_k$ are order-$(1 - \alpha)/2$ weak approximations of each other.

Then we can conclude that $\boldsymbol{x}_k$ and $\boldsymbol{z}_k$ are order-$(1 - \alpha)/2$ weak approximations of each other, which states our main theorem Theorem 3.5.

First, we give a control of the averaged learning rate $\bar{\eta}_k$ (Equation (14)) with the following lemma.

**Lemma C.2.** *Let $(\eta_k, \beta_k)$ be a learning rate schedule scaled by $\eta$ (Definition 3.2) with index $\alpha$, and $\bar{\eta}_k$ be the averaged learning rate (Equation (14)), there is*

$$\bar{\eta}_k \leq \frac{\lambda_{\max} \eta_{\max}}{\lambda_{\min}} \eta$$

$$\frac{\bar{\eta}_k \beta_k}{1 - \beta_k} \leq \frac{\lambda_{\max} \eta_{\max}}{\lambda_{\min}^2} \eta^{1-\alpha}.$$

*Proof.* By Definition 3.2 we know $\beta_k \in [1 - \lambda_{\max} \eta^\alpha, 1 - \lambda_{\min} \eta^\alpha]$ and $\eta_k \leq \eta_{\max} \eta$. Therefore

$$\bar{\eta}_k = \sum_{s=k}^{\infty} \eta_s \beta_{k+1:s} (1 - \beta_k)$$

$$\leq \sum_{s=k}^{\infty} \eta_{\max} \eta (1 - \lambda_{\min} \eta^\alpha)^{s-k} \lambda_{\max} \eta^\alpha$$

$$= \frac{\lambda_{\max} \eta_{\max}}{\lambda_{\min}} \eta.$$

$$\frac{\bar{\eta}_k \beta_k}{1 - \beta_k} \leq \frac{\lambda_{\max} \eta_{\max}}{\lambda_{\min}} \eta \cdot \frac{1}{1 - \beta_k}$$

$$\leq \frac{\lambda_{\max} \eta_{\max}}{\lambda_{\min}^2} \eta^{1-\alpha}.$$

$\square$

### C.1.1   STEP 1

For the first step we are going to proof the following Theorem C.3. For notation simplicity we omit the superscript $\eta$ for the scaling when there is no ambiguity.

**Theorem C.3** (Weak Approximation of Coupled Trajectory). *With the learning rate schedule $(\eta_k, \beta_k)$ scaled by $\eta$ with index $\alpha \in [0, 1)$. Assume that the NGOS $\mathcal{G}_\sigma$ satisfy Assumption 3.4 and the initialization $\boldsymbol{m}_0$ satisfy Assumption 3.3. For any noise scale $\sigma \leq \eta^{-1/2}$, let $(\boldsymbol{x}_k, \boldsymbol{m}_k)$ be the SGDM trajectory and $\boldsymbol{y}_k$ be the corresponding coupled trajectory (Equation (15)). Then, the coupled trajectory $\boldsymbol{y}_k$ is an order-$\gamma$ weak approximation (Definition 3.1) to $\boldsymbol{x}_k$ with $\gamma = (1 - \alpha)/2$.*

For notations, let $\nabla_k = \nabla\mathcal{L}(\boldsymbol{x}_k)$ so $\boldsymbol{g}_k = \nabla_k + \sigma\boldsymbol{v}_k$ is the stochastic gradient sampled from the NGOS. Also by Assumption 3.4, as $\boldsymbol{\Sigma}^{1/2}$ is bounded, let $C_{\mathrm{tr}} > 0$ be the constant that $\mathrm{tr}\,\boldsymbol{\Sigma}(\boldsymbol{x}) \leq C_{\mathrm{tr}}$ for all $\boldsymbol{x} \in \mathbb{R}^d$. As $\nabla\mathcal{L}$ is Lipschitz, let $L > 0$ be the constant that $\|\nabla\mathcal{L}(\boldsymbol{x}) - \nabla\mathcal{L}(\boldsymbol{y})\| \leq L\|\boldsymbol{x} - \boldsymbol{y}\|$ for all $\boldsymbol{x}, \boldsymbol{y} \in \mathbb{R}^d$. As $\boldsymbol{\Sigma}^{1/2}$ is Lipschitz and bounded, let $L_\Sigma > 0$ be the constant that $|\mathrm{tr}(\boldsymbol{\Sigma}(\boldsymbol{x}) - \boldsymbol{\Sigma}(\boldsymbol{y}))| \leq L_\Sigma\|\boldsymbol{x} - \boldsymbol{y}\|$ for all $\boldsymbol{x}, \boldsymbol{y} \in \mathbb{R}^d$.

To show the result, let $h$ be a test function for the weak approximation, then for some $\boldsymbol{z}$ between $\boldsymbol{x}_k$ and $\boldsymbol{y}_k$, $|\mathbb{E}h(\boldsymbol{x}_k) - \mathbb{E}h(\boldsymbol{y}_k)| = \mathbb{E}[|\langle \nabla h(\boldsymbol{z}), \boldsymbol{x}_k - \boldsymbol{y}_k \rangle|] \leq \mathbb{E}[\|\nabla h(\boldsymbol{z})\|\|\boldsymbol{x}_k - \boldsymbol{y}_k\|] = \mathbb{E}[\frac{\eta_k\beta_k}{1-\beta_k}\|\nabla h(\boldsymbol{z})\|\|\boldsymbol{m}_k\|]$. Thus it suffices to bound $\mathbb{E}\|\boldsymbol{x}_k\|^{2m}$, $\mathbb{E}\|\boldsymbol{y}_k\|^{2m}$ and $\mathbb{E}\|\boldsymbol{m}_k\|$ for any $m \geq 0$ and $k = O(\eta^{-1})$. As the gradient noises are scaled with variance $O(\sigma^2) = O(\eta^{-1})$, there will be $\mathbb{E}\|\boldsymbol{m}_k\| \gg 1$, and we will show that $\mathbb{E}\|\boldsymbol{m}_k\| = O(\eta^{(\alpha-1)/2})$ is the correct scale so we still have $\mathbb{E}\|\boldsymbol{x}_k\|^{2m} = O(1)$ and $|\mathbb{E}h(\boldsymbol{x}_k) - \mathbb{E}h(\boldsymbol{y}_k)| = O(\eta^{(1-\alpha)/2})$. An useful inequality we will use often in our proof is the Grönwall's inequality.

**Lemma C.4** (Grönwall's Inequality (Gronwall, 1919)). *For non-negative sequences $\{f_i, g_i, k_i \in \mathbb{R}\}_{i \geq 0}$, if for all $t > 0$*

$$f_t \leq g_t + \sum_{s=0}^{t-1} k_s f_s$$

*then $f_t \leq g_t + \sum_{s=0}^{t-1} k_s g_s \exp(\sum_{r=s+1}^{t-1} k_r)$.*

Next follows the proofs.

**Lemma C.5.** *We can bound the expected squared norm*

$$\mathbb{E}\|\boldsymbol{m}_k\|^2 \leq 12\frac{\lambda_{\max}^2}{\lambda_{\min}^2} \sup_{\tau=1,\dots,k} \mathbb{E}(\|\nabla_\tau\|)^2 + (4 + 8\frac{L\eta_{\max}}{\lambda_{\min}} + \eta^{\alpha-1})K_0 \tag{21}$$

*with constant $K_0 = \frac{2\lambda_{\max}^2}{\lambda_{\min}}C_{\mathrm{tr}} + 3\mathbb{E}\|\boldsymbol{m}_0\|^2 \leq \frac{2\lambda_{\max}^2}{\lambda_{\min}}C_{\mathrm{tr}} + 3C_2$ ($C_2$ by Assumption 3.3), for a small enough learning rate $\eta < \left(\frac{\lambda_{\min}}{4L\eta_{\max}}\right)^{\frac{1}{1-\alpha}}$.*

*Proof.* To bound $\mathbb{E}\|\boldsymbol{m}_k\|^2$, we unroll the momentum by Lemma C.1 and write

$$\boldsymbol{m}_k = \beta_{0:k-1}\boldsymbol{m}_0 + \sum_{s=0}^{k-1} \beta_{s+1:k-1}(1 - \beta_s)\boldsymbol{g}_s \tag{22}$$

Define

$$\widetilde{\boldsymbol{m}}_k = \beta_{0:k-1}\boldsymbol{m}_0 + \sum_{s=0}^{k-1} \beta_{s+1:k-1}(1 - \beta_s)\nabla_s.$$

Then $\mathbb{E}\boldsymbol{m}_k = \mathbb{E}\widetilde{\boldsymbol{m}}_k$, and

$$\boldsymbol{m}_k - \widetilde{\boldsymbol{m}}_k = \sum_{s=0}^{k-1} \beta_{s+1:k-1}(1 - \beta_s)\sigma\boldsymbol{v}_s$$

Let constant $K_0 = 2C_{\mathrm{tr}} + 3C_2$, we can write

$$\mathbb{E}\left\|\boldsymbol{m}_k\right\|^2 \le \mathbb{E}(\|\widetilde{\boldsymbol{m}}_k\| + \|\boldsymbol{m}_k - \widetilde{\boldsymbol{m}}_k\|)^2 \tag{23}$$

$$\le 2\mathbb{E}\left\|\widetilde{\boldsymbol{m}}_k\right\|^2 + 2\mathbb{E}\left\|\boldsymbol{m}_k - \widetilde{\boldsymbol{m}}_k\right\|^2 \tag{24}$$

$$= 2\mathbb{E}\left\|\widetilde{\boldsymbol{m}}_k\right\|^2 + 2\sum_{s=0}^{k-1}(\beta_{s+1:k-1})^2(1-\beta_s)^2\sigma^2\mathbb{E}\operatorname{tr}\boldsymbol{\Sigma}(\boldsymbol{x}_s) \tag{25}$$

$$\le 2\mathbb{E}\left\|\widetilde{\boldsymbol{m}}_k\right\|^2 + 2\sum_{s=0}^{k-1}(1-\lambda_{\min}\eta^\alpha)^{2(k-s-1)}\lambda_{\max}^2\eta^{2\alpha}\sigma^2\mathbb{E}\operatorname{tr}\boldsymbol{\Sigma}(\boldsymbol{x}_s) \tag{26}$$

$$\le 2\mathbb{E}\left\|\widetilde{\boldsymbol{m}}_k\right\|^2 + \eta^{\alpha-1}\frac{2\lambda_{\max}^2}{\lambda_{\min}}C_{\mathrm{tr}} \tag{27}$$

$$\le 2\mathbb{E}\left\|\widetilde{\boldsymbol{m}}_k\right\|^2 + \eta^{\alpha-1}K_0. \tag{28}$$

On the other hand, we know

$$\widetilde{\boldsymbol{m}}_k = \beta_{0:k-1}\boldsymbol{m}_0 + \sum_{s=0}^{k-1}\beta_{s+1:k-1}(1-\beta_s)\nabla_s$$

We use the Lipschitzness of $\nabla$ to write

$$\|\nabla_s - \nabla_k\| \le L\|\boldsymbol{x}_s - \boldsymbol{x}_k\| \le L\sum_{\tau=s}^{k-1}\eta_\tau\left\|\boldsymbol{m}_{\tau+1}\right\|,$$

then we can write

$$\mathbb{E}\left\|\widetilde{\boldsymbol{m}}_k\right\|^2 \le \mathbb{E}\left(\beta_{0:k-1}\left\|\boldsymbol{m}_0\right\| + \sum_{s=0}^{k-1}\beta_{s+1:k-1}(1-\beta_s)(\|\nabla_k\| + L\sum_{\tau=s}^{k-1}\eta_\tau\left\|\boldsymbol{m}_{\tau+1}\right\|)\right)^2$$

$$\le \mathbb{E}\left(\left\|\boldsymbol{m}_0\right\| + \frac{\lambda_{\max}}{\lambda_{\min}}\|\nabla_k\| + L\eta\eta_{\max}\sum_{\tau=0}^{k-1}(1-\lambda_{\min}\eta^\alpha)^{k-1-\tau}\left\|\boldsymbol{m}_{\tau+1}\right\|\right)^2$$

$$\le 3\mathbb{E}(\left\|\boldsymbol{m}_0\right\|)^2 + 3\frac{\lambda_{\max}^2}{\lambda_{\min}^2}\mathbb{E}(\|\nabla_k\|)^2 + 3L^2\eta^2\eta_{\max}^2\left(\sum_{\tau=0}^{k-1}(1-\lambda_{\min}\eta^\alpha)^{k-1-\tau}\right)\cdot$$

$$\left(\sum_{\tau=0}^{k-1}(1-\lambda_{\min}\eta^\alpha)^{k-1-\tau}\mathbb{E}\|\boldsymbol{m}_{\tau+1}\|^2\right)$$

$$\le K_0 + 3\frac{\lambda_{\max}^2}{\lambda_{\min}^2}\mathbb{E}(\|\nabla_k\|)^2 + \frac{3L^2\eta_{\max}^2\eta^{2-2\alpha}}{\lambda_{\min}^2}\sup_{\tau=1\cdots k}\mathbb{E}\|\boldsymbol{m}_\tau\|^2.$$

Then by Equation (28), we know

$$\sup_{\tau=1,\ldots,k}\mathbb{E}\|\boldsymbol{m}_\tau\|^2 \le 2\sup_{\tau=1,\ldots,k}\mathbb{E}\left\|\widetilde{\boldsymbol{m}}_\tau\right\|^2 + \eta^{\alpha-1}K_0,$$

so

$$\sup_{\tau=1,\ldots,k}\mathbb{E}\left\|\widetilde{\boldsymbol{m}}_\tau\right\|^2 \le K_0 + 3\frac{\lambda_{\max}^2}{\lambda_{\min}^2}\sup_{\tau=1,\ldots,k}\mathbb{E}(\|\nabla_\tau\|)^2 + \frac{6L^2\eta_{\max}^2\eta^{1-\alpha}}{\lambda_{\min}^2}K_0 \tag{29}$$

$$+ \frac{6L^2\eta_{\max}^2\eta^{2-2\alpha}}{\lambda_{\min}^2}\sup_{\tau=1,\cdots,k}\mathbb{E}\|\widetilde{\boldsymbol{m}}_\tau\|^2 \tag{30}$$

Choose $\eta$ small enough so that $\frac{6L^2\eta_{\max}^2\eta^{2-2\alpha}}{\lambda_{\min}^2} < \frac{1}{2}$, we know

$$\sup_{\tau=1,\ldots,k}\mathbb{E}\left\|\widetilde{\boldsymbol{m}}_\tau\right\|^2 \le 6\frac{\lambda_{\max}^2}{\lambda_{\min}^2}\sup_{\tau=1,\ldots,k}\mathbb{E}(\|\nabla_\tau\|)^2 + (2 + 2\frac{\sqrt{3}L\eta_{\max}}{\lambda_{\min}})K_0. \tag{31}$$

So

$$\mathbb{E}\left\|\boldsymbol{m}_k\right\|^2 \le 2\mathbb{E}\left\|\widetilde{\boldsymbol{m}}_k\right\|^2 + \eta^{\alpha-1}K_0 \tag{32}$$

$$\le 12\frac{\lambda_{\max}^2}{\lambda_{\min}^2}\sup_{\tau=1,\dots,k}\mathbb{E}(\|\nabla_\tau\|)^2 + (4 + 8\frac{L\eta_{\max}}{\lambda_{\min}} + \eta^{\alpha-1})K_0. \tag{33}$$

$\square$

**Lemma C.6.** *There is a constant $f$ that depends on $T$, irrelevant to $\eta$, such that*

$$\sup_{k=0,\dots,\lfloor T/\eta\rfloor}\mathbb{E}\left\|\nabla_k\right\|^2 \le f(T)$$

*when $\eta < \min((\frac{\lambda_{\min}^3}{12L\lambda_{\max}^2\eta_{\max}})^{\frac{1}{1-\alpha}}, 1)$*

*Proof.* By the Lipschitzness of $\nabla\mathcal{L}$, we know

$$\|\nabla_k\| \le \|\nabla_0\| + L\left\|\boldsymbol{x}_k - \boldsymbol{x}_0\right\| \tag{34}$$

$$\le \|\nabla_0\| + L\left\|\boldsymbol{y}_k - \boldsymbol{y}_0\right\| + L\left\|\boldsymbol{x}_0 - \boldsymbol{y}_0\right\| + L\left\|\boldsymbol{x}_k - \boldsymbol{y}_k\right\| \tag{35}$$

Observe that $\|\boldsymbol{x}_k - \boldsymbol{y}_k\| = \frac{\bar{\eta}_k\beta_k}{1-\beta_k}\|\boldsymbol{m}_k\| \le \frac{\lambda_{\max}\eta_{\max}}{\lambda_{\min}^2}\eta^{1-\alpha}\|\boldsymbol{m}_k\|$. Let $d_k = \mathbb{E}\|\boldsymbol{y}_k - \boldsymbol{y}_0\|^2$. As a result, we can write

$$\mathbb{E}\|\nabla_k\|^2 \le 3\mathbb{E}\left(\|\nabla_0\| + \frac{L\lambda_{\max}\eta_{\max}}{\lambda_{\min}^2}\eta^{1-\alpha}\|\boldsymbol{m}_0\|\right)^2 + 3\left(\frac{L\lambda_{\max}\eta_{\max}}{\lambda_{\min}^2}\eta^{1-\alpha}\right)^2\mathbb{E}(\|\boldsymbol{m}_k\|^2) + 3L^2 d_k. \tag{36}$$

Choose $\eta$ such that $\eta^{1-\alpha} < \frac{\lambda_{\min}^3}{12L\lambda_{\max}^2\eta_{\max}}$ and let the constant

$$K_1 = 3\mathbb{E}\left(\|\nabla\mathcal{L}(\boldsymbol{x}_0)\| + \frac{\lambda_{\min}}{12\lambda_{\max}}\|\boldsymbol{m}_0\|\right)^2 \tag{37}$$

$$+ \left(\frac{\lambda_{\min}}{6\lambda_{\max}}\right)^2(3 + 6\frac{L\eta_{\max}}{\lambda_{\min}})K_0 + \left(\frac{L\eta_{\max}}{4\lambda_{\min}}\right)K_0 \tag{38}$$

where recall $K_0$ is the constant from Lemma C.5. Plug Equation (21) into Equation (36) gives

$$\mathbb{E}\|\nabla_k\|^2 \le K_1 + \frac{1}{2}\sup_{\tau=1,\dots,k}\mathbb{E}(\|\nabla_\tau\|)^2 + 3L^2 d_k. \tag{39}$$

therefore

$$\mathbb{E}\|\nabla_k\|^2 \le 2K_1 + 6L^2\sup_{\tau=1,\dots,k}d_\tau. \tag{40}$$

Now we need to bound $d_k$ by iteration.

$$d_{k+1} = d_k + \mathbb{E}\|\boldsymbol{y}_{k+1} - \boldsymbol{y}_k\|^2 + 2\mathbb{E}(\boldsymbol{y}_{k+1} - \boldsymbol{y}_k)^\top(\boldsymbol{y}_k - \boldsymbol{y}_0) \tag{41}$$

$$= d_k + \bar{\eta}_k^2\mathbb{E}\|\nabla_k\|^2 + \bar{\eta}_k^2\sigma^2\mathbb{E}\operatorname{tr}(\boldsymbol{\Sigma}(\boldsymbol{x}_k)) + 2\mathbb{E}(\nabla_k)^\top(\boldsymbol{y}_k - \boldsymbol{y}_0) \tag{42}$$

$$\le d_k + \frac{\lambda_{\max}^2}{\lambda_{\min}^2}\eta_{\max}^2\eta^2\mathbb{E}\|\nabla_k\|^2 + \frac{\lambda_{\max}^2}{\lambda_{\min}^2}\eta_{\max}^2\eta C_{\operatorname{tr}} + 2\frac{\lambda_{\max}}{\lambda_{\min}}\eta_{\max}\eta\sqrt{d_k\mathbb{E}\|\nabla_k\|^2} \tag{43}$$

Let function $K_2(\boldsymbol{m}_0, \boldsymbol{x}_0) = \frac{\lambda_{\max}^2}{\lambda_{\min}^2}\eta_{\max}^2 C_{\operatorname{tr}} + 2\frac{\lambda_{\max}^2}{\lambda_{\min}^2}\eta_{\max}^2 K_1(\boldsymbol{m}_0, \boldsymbol{x}_0) + 2\frac{\lambda_{\max}}{\lambda_{\min}}\eta_{\max}\frac{K_1(\boldsymbol{m}_0,\boldsymbol{x}_0)}{\sqrt{6L}}$, and $\bar{d}_k = \sup_{\tau=1,\dots,k}d_\tau$. We know

$$\sqrt{d_k\mathbb{E}\|\nabla_k\|^2} \le \sqrt{6L^2}d_k + \frac{K_1}{\sqrt{6L^2}}, \tag{44}$$

$$\bar{d}_{k+1} \le \bar{d}_k + K_2\eta + 6L^2\frac{\lambda_{\max}^2}{\lambda_{\min}^2}\eta_{\max}^2\eta^2\bar{d}_k + 2\frac{\lambda_{\max}}{\lambda_{\min}}\eta_{\max}\eta(\sqrt{6L}\bar{d}_k) \tag{45}$$

$$\le (1 + 6L\frac{\lambda_{\max}\eta_{\max}}{\lambda_{\min}}\eta + 6L^2\frac{\lambda_{\max}^2\eta_{\max}^2}{\lambda_{\min}^2}\eta^2)\bar{d}_k + \eta K_2 \tag{46}$$

Let $\kappa = 1 + 6L\frac{\lambda_{\max}\eta_{\max}}{\lambda_{\min}}\eta + 6L^2\frac{\lambda_{\max}^2\eta_{\max}^2}{\lambda_{\min}^2}\eta^2$, then by the Grönwall's inequality (Lemma D.10),

$$\bar{d}_k \le \eta K_2(1 + \sum_{s=0}^{k-1}\kappa\exp^{\kappa s}) \tag{47}$$

$$\le K_2 + K_2 T \exp^{1 + 6L\frac{\lambda_{\max}\eta_{\max}}{\lambda_{\min}}T + 6L^2\frac{\lambda_{\max}^2\eta_{\max}^2}{\lambda_{\min}^2}T}. \tag{48}$$

Plugging into Equation (40) finished the proof. □

**Lemma C.7.** *There is a constant $f$ that depends on $T$, irrelevant to $\eta$, such that*

$$\mathbb{E}\|\boldsymbol{m}_k\|^2 \le \eta^{\alpha-1}f(T)$$

*for all $k \le \frac{T}{\eta}$.*

*Proof.* The result directly follows from Lemma C.5 and Lemma C.6. □

**Lemma C.8.** *There exists function $g(m,T)$ that, for all $k \le \frac{T}{\eta}$,*

$$\mathbb{E}[(1 + \|\boldsymbol{x}_k\|^2)^m] \le g(m,T),$$
$$\mathbb{E}[(1 + \|\boldsymbol{y}_k\|^2)^m] \le g(m,T),$$
$$\mathbb{E}[\eta^m\|\boldsymbol{g}_k\|^{2m}] \le g(m,T),$$
$$\mathbb{E}[(1 + \|\boldsymbol{y}_k\|^2)^m\eta^{1-\alpha}\|\boldsymbol{m}_k\|^2] \le g(m,T)$$

*and that $g$ is irrelevant to $k$ and $\eta$.*

*Proof.* We use the fact that $(a+b)^m \le 2^{m-1}(a^m + b^m)$ from the Jensen inequality for $a,b > 0$ and $m \ge 1$. Furthermore by Young's inequality $a^\alpha b^\beta \le \frac{\alpha a^{\alpha+\beta} + \beta b^{\alpha+\beta}}{\alpha+\beta} \le a^{\alpha+\beta} + b^{\alpha+\beta}$ for $\alpha, \beta > 0$.

$$\|\boldsymbol{g}_k\|^{2m} \le (\|\nabla_k\| + \sigma\|\boldsymbol{v}_k\|)^{2m}$$
$$\le 2^{2m-1}(\|\nabla_k\|^{2m} + \eta^{-m}\|\boldsymbol{v}_k\|^{2m})$$
$$\|\nabla_k\|^{2m} \le (\|\nabla_0\| + L\|\boldsymbol{x}_k - \boldsymbol{x}_0\|)^{2m}$$
$$\le 2^{2m-1}((\|\nabla_0\| + L\|\boldsymbol{x}_0\|)^{2m} + L^{2m}\|\boldsymbol{x}_k\|^{2m})$$

As $\mathbb{E}\|\boldsymbol{v}_k\|^{2m}$ is bounded by Assumption 3.4, for some constant $R$ there is

$$\mathbb{E}\|\boldsymbol{g}_k\|^{2m} \le R(1 + \eta^{-m})\mathbb{E}(1 + \|\boldsymbol{x}_k\|^{2m}). \tag{49}$$

Now we need to show the bounds on $\mathbb{E}[(1 + \|\boldsymbol{x}_k\|^2)^m]$ and $\mathbb{E}[(1 + \|\boldsymbol{y}_k\|^2)^m]$. Specifically we shall prove $\mathbb{E}[(1 + \|\boldsymbol{y}_k\|^2 + \eta^{2-2\alpha}\|\boldsymbol{m}_k\|^2)^m] \le g(m,T)$ and $\mathbb{E}[(1 + \|\boldsymbol{y}_k\|^2 + \eta^{2-2\alpha}\|\boldsymbol{m}_k\|^2)^m\eta^{1-\alpha}\|\boldsymbol{m}_k\|^2] \le g(m,T)$ for some function $g$.

Let

$$\delta_k = \bar{\eta}_k^2\|\boldsymbol{g}_k\|^2 - 2\bar{\eta}_k\langle\boldsymbol{g}_k,\boldsymbol{y}_k\rangle + \eta^{2-2\alpha}(\|\beta_k\boldsymbol{m}_k + (1-\beta_k)\boldsymbol{g}_k\|^2 - \|\boldsymbol{m}_k\|^2),$$

then there is

$$\mathbb{E}[(1 + \|\boldsymbol{y}_{k+1}\|^2 + \eta^{2-2\alpha}\|\boldsymbol{m}_{k+1}\|^2)^m|\boldsymbol{y}_k,\boldsymbol{m}_k]$$
$$= \mathbb{E}[(1 + \|\boldsymbol{y}_k - \bar{\eta}_k\boldsymbol{g}_k\|^2 + \eta^{2-2\alpha}\|\beta_k\boldsymbol{m}_k + (1-\beta_k)\boldsymbol{g}_k\|^2)^m|\boldsymbol{y}_k,\boldsymbol{m}_k]$$
$$= \mathbb{E}[(1 + \|\boldsymbol{y}_k\|^2 + \eta^{2-2\alpha}\|\boldsymbol{m}_k\|^2 + \delta_k)^m|\boldsymbol{y}_k,\boldsymbol{m}_k]$$
$$\le (1 + \|\boldsymbol{y}_k\|^2 + \eta^{2-2\alpha}\|\boldsymbol{m}_k\|^2)^m + m(1 + \|\boldsymbol{y}_k\|^2 + \eta^{2-2\alpha}\|\boldsymbol{m}_k\|^2)^{m-1}\mathbb{E}[\delta_k|\boldsymbol{y}_k,\boldsymbol{m}_k]$$
$$+ \mathbb{E}[\delta_k^2\sum_{i=0}^{m-2}\binom{m}{i+2}\delta_k^i(1 + \|\boldsymbol{y}_k\|^2 + \eta^{2-2\alpha}\|\boldsymbol{m}_k\|^2)^{m-2-i}|\boldsymbol{y}_k,\boldsymbol{m}_k]$$
$$\le (1 + \|\boldsymbol{y}_k\|^2 + \eta^{2-2\alpha}\|\boldsymbol{m}_k\|^2)^m + m(1 + \|\boldsymbol{y}_k\|^2 + \eta^{2-2\alpha}\|\boldsymbol{m}_k\|^2)^{m-1}\mathbb{E}[\delta_k|\boldsymbol{y}_k,\boldsymbol{m}_k]$$
$$+ 2^m\mathbb{E}[\delta_k^2(|\delta_k|^{m-2} + (1 + \|\boldsymbol{y}_k\|^2 + \eta^{2-2\alpha}\|\boldsymbol{m}_k\|^2)^{m-2})|\boldsymbol{y}_k,\boldsymbol{m}_k].$$

Then there exists constant $K_3$ independent of $\eta$ such that

$$
\begin{aligned}
\mathbb{E}[\delta_k|\boldsymbol{y}_k,\boldsymbol{m}_k] &= \bar{\eta}_k^2\left\|\nabla_k\right\|^2 + \bar{\eta}_k^2\sigma^2\mathbb{E}_{|\boldsymbol{y}_k,\boldsymbol{m}_k}\operatorname{tr}\boldsymbol{\Sigma}(\boldsymbol{x}_k) - 2\bar{\eta}_k\langle\nabla_k,\boldsymbol{y}_k\rangle + \eta^{2-2\alpha}(\|\beta_k\boldsymbol{m}_k + (1-\beta_k)\boldsymbol{g}_k\|^2 - \|\boldsymbol{m}_k\|^2) \\
&\leq \mathbb{E}_{|\boldsymbol{y}_k,\boldsymbol{m}_k}[\frac{\lambda_{\max}^2\eta_{\max}^2}{\lambda_{\min}^2}(2\eta^2(\|\nabla_0\| + L\|\boldsymbol{x}_0\|)^2 + 2\eta^2L^2\|\boldsymbol{x}_k\|^2 + \eta C_{\mathrm{tr}}) \\
&\quad + 2\frac{\lambda_{\max}\eta_{\max}}{\lambda_{\min}}\eta(\|\nabla_0\| + L\|\boldsymbol{x}_k\| + L\|\boldsymbol{x}_0\|)\|\boldsymbol{y}_k\| \\
&\quad + \eta^{2-2\alpha}(\frac{1-\beta_k}{1+\beta_k}\|\nabla_k\|^2 + (1-\beta_k)^2\sigma^2\mathbb{E}\operatorname{tr}\boldsymbol{\Sigma}(\boldsymbol{x}_k))] \\
&\leq (1 + \mathbb{E}[\|\boldsymbol{x}_k\|^2\,|\boldsymbol{y}_k,\boldsymbol{m}_k] + \|\boldsymbol{y}_k\|^2)\eta K_3
\end{aligned}
$$

as $(1-\beta_k)\eta^{2-2\alpha} = O(\eta^{2-\alpha}) = O(\eta)$ and $(1-\beta_k)^2\sigma^2\eta^{2-2\alpha} = O(\eta^{2\alpha-1+2-2\alpha}) = O(\eta)$.

Note that

$$
\begin{aligned}
\|\boldsymbol{x}_k\|^2 &= \left\|\boldsymbol{y}_k + \bar{\eta}_k\beta_k(1-\beta_k)^{-1}\boldsymbol{m}_k\right\|^2 \\
&\leq 2\|\boldsymbol{y}_k\|^2 + \frac{2\eta_{\max}^2\lambda_{\max}^2}{\lambda_{\max}^4}\eta^{2-2\alpha}\|\boldsymbol{m}_k\|^2
\end{aligned}
$$

Then we can write $\mathbb{E}[\delta_k|\boldsymbol{y}_k,\boldsymbol{m}_k] \leq K_4\eta(1 + \|\boldsymbol{y}_k\|^2 + \eta^{2-2\alpha}\|\boldsymbol{m}_k\|^2)$ for some constant $K_4$ independent of $\eta$. Futhermore, as $\mathbb{E}_{|\boldsymbol{y}_k}\langle\boldsymbol{v}_k,\boldsymbol{y}_k\rangle = \mathbb{E}_{|\boldsymbol{y}_k}\langle\boldsymbol{v}_k,\boldsymbol{m}_k\rangle = \mathbb{E}_{|\boldsymbol{y}_k}\langle\boldsymbol{v}_k,\boldsymbol{x}_k\rangle = 0$, expansion gives for some constant $K_4, K_5, K_6, K_7$,

$$
\begin{aligned}
\mathbb{E}[\delta_k^2|\boldsymbol{y}_k] &\leq K_4\mathbb{E}(\eta\|\boldsymbol{x}_k\|^2 + \eta\|\boldsymbol{v}_k\|^2 + \eta\|\boldsymbol{y}_k\|^2 + \eta^{1/2}\langle\boldsymbol{v}_k,\boldsymbol{y}_k\rangle + \eta^{3/2-\alpha}\langle\boldsymbol{m}_k,\boldsymbol{v}_k\rangle + \eta^{3/2-\alpha}\langle\boldsymbol{x}_k,\boldsymbol{v}_k\rangle)^2 \\
&\leq K_5\eta(1 + \|\boldsymbol{y}_k\|^2 + \eta^{2-2\alpha}\|\boldsymbol{m}_k\|^2). \\
\mathbb{E}[|\delta_k|^m|\boldsymbol{y}_k] &= K_6\mathbb{E}(\eta\|\boldsymbol{x}_k\|^2 + \eta\|\boldsymbol{v}_k\|^2 + \eta\|\boldsymbol{y}_k\|^2 + \eta^{1/2}\langle\boldsymbol{v}_k,\boldsymbol{y}_k\rangle + \eta^{3/2-\alpha}\langle\boldsymbol{m}_k,\boldsymbol{v}_k\rangle + \eta^{3/2-\alpha}\langle\boldsymbol{x}_k,\boldsymbol{v}_k\rangle)^m \\
&\leq K_7\eta^{m/2}(1 + \|\boldsymbol{y}_k\|^2 + \eta^{2-2\alpha}\|\boldsymbol{m}_k\|^2).
\end{aligned}
$$

Therefore taking expecation with respect to all, we know

$$
\mathbb{E}[(1+\|\boldsymbol{y}_{k+1}\|^2+\eta^{2-2\alpha}\|\boldsymbol{m}_{k+1}\|^2)^m] \leq (1+mK_4\eta+2^m(K_5+K_7)\eta)\mathbb{E}(1+\|\boldsymbol{y}_k\|^2+\eta^{2-2\alpha}\|\boldsymbol{m}_k\|^2)^m
$$

Therefore by the Grönwall's inequality (Lemma D.10),

$$
\begin{aligned}
\mathbb{E}(1 + \|\boldsymbol{y}_k\|^2 + \eta^{2-2\alpha}\|\boldsymbol{m}_k\|^2)^m &\leq e^{(mK_4\eta+2^m(K_5+K_7)\eta)k}\mathbb{E}(1 + \|\boldsymbol{y}_0\|^2 + \eta^{2-2\alpha}\|\boldsymbol{m}_0\|^2)^m \\
&\leq e^{mK_4T+2^m(K_5+K_7)T}3^m(1 + \mathbb{E}\|\boldsymbol{y}_0\|^{2m} + \eta^{2-2\alpha}\mathbb{E}\|\boldsymbol{m}_0\|^{2m}).
\end{aligned}
$$

And clearly $\mathbb{E}\|\boldsymbol{y}_0\|^{2m} \leq 2^m\mathbb{E}\|\boldsymbol{x}_0\|^{2m} + (\frac{2\lambda_{\max}\eta_{\max}}{\lambda_{\min}^2})^m\eta^{m-m\alpha}\mathbb{E}\|\boldsymbol{m}_0\|^{2m}$. Therefore we finished bounding the moments of $\boldsymbol{y}_k$. Similar induction arguments on $(1 + \|\boldsymbol{y}_k\|^2 + \eta^{2-2\alpha}\|\boldsymbol{m}_k\|^2)^m\eta^{1-\alpha}\|\boldsymbol{m}_k\|^2$ offer the last equation in the lemma. Then for $\boldsymbol{x}_k$, we can write

$$
\|\boldsymbol{x}_k\|^{2m} \leq 2^{2m}\left(\|\boldsymbol{y}_k\|^{2m} + (\frac{2\lambda_{\max}\eta_{\max}}{\lambda_{\min}^2})^m\eta^{m-m\alpha}\|\boldsymbol{m}_k\|^{2m}\right)
$$

And for $\boldsymbol{g}_k$ with Equation (49) we are able to finish the proof. $\qquad\square$

*Proof for Theorem C.3.* We expand the weak approximation error for a single $k$. There is $\theta \in [0,1]$ and $\boldsymbol{z} = \theta\boldsymbol{x}_k + (1-\theta)\boldsymbol{y}_k$ such that

$$
\begin{aligned}
|\mathbb{E}h(\boldsymbol{x}_k) - \mathbb{E}h(\boldsymbol{y}_k)| &= \mathbb{E}[|\langle\nabla h(\boldsymbol{z}),\boldsymbol{x}_k - \boldsymbol{y}_k\rangle|] \\
&\leq \mathbb{E}[\|\nabla h(\boldsymbol{z})\|\|\boldsymbol{x}_k - \boldsymbol{y}_k\|] \\
&\leq \sqrt{\mathbb{E}[\|\nabla h(\boldsymbol{z})\|^2]\mathbb{E}[\|\boldsymbol{x}_k - \boldsymbol{y}_k\|^2]}.
\end{aligned}
$$

As $\nabla h(\boldsymbol{z})$ have polynomial growth, there is $k_1, k_2$ such that

$$
\|\nabla h(\boldsymbol{z})\|^2 \leq k_1(1 + \|\boldsymbol{z}\|^{k_2}) \leq k_1(1 + \|\boldsymbol{x}_k\|^{k_2} + \|\boldsymbol{y}_k\|^{k_2}).
$$

Combined with Lemma C.8, we see $\sqrt{\mathbb{E}[\|\nabla h(\boldsymbol{z})\|^2}$ is bounded by some constant $2k_1 g(k_2, T)$. From the definition of the coupled trajectory, we can write

$$\mathbb{E}\|\boldsymbol{x}_k - \boldsymbol{y}_k\|^2 \leq \frac{\lambda_{\max}^2 \eta_{\max}^2}{\lambda_{\min}^4} \eta^{2-2\alpha} \mathbb{E}\|\boldsymbol{m}_k\|^2$$

Then as Lemma C.7, there is constant $f$ that $\mathbb{E}\|\boldsymbol{m}_k\|^2 \leq \eta^{\alpha-1} f(T)$, therefore

$$|\mathbb{E}h(\boldsymbol{x}_k) - \mathbb{E}h(\boldsymbol{y}_k)| \leq 2k_1 \eta^{(1-\alpha)/2} \frac{\lambda_{\max}\eta_{\max}}{\lambda_{\min}^2} f(T)g(k_2, T).$$

Thus by definition, $\boldsymbol{x}_k$ and $\boldsymbol{y}_k$ are order-$(1-\alpha)/2$ weak approximations of each other. $\qquad\square$

### C.1.2 STEP 2

In this section we are comparing the trajectory $\boldsymbol{y}_k$ with a SGD trajectory $\boldsymbol{z}_k$. To avoid notation ambiguity, denote $\boldsymbol{g}(\boldsymbol{x}) \sim \mathcal{G}_\sigma(\boldsymbol{x})$ to be the stochastic gradient sampled at $\boldsymbol{x}$. Recall that (with $\boldsymbol{y}_0 = \boldsymbol{x}_0 - \frac{\bar{\eta}_0\beta_0}{1-\beta_0}\boldsymbol{m}_0$ and $\boldsymbol{z}_0 = \boldsymbol{x}_0$)

$$\boldsymbol{y}_k = \boldsymbol{y}_{k-1} - \bar{\eta}_{k-1}\boldsymbol{g}(\boldsymbol{x}_{k-1})$$
$$\boldsymbol{z}_k = \boldsymbol{z}_{k-1} - \bar{\eta}_{k-1}\boldsymbol{g}(\boldsymbol{z}_{k-1})$$

The only difference in the iterate is that the stochastic gradients are taken at close but different locations of the trajectory. Therefore to study the trajectory difference, we adopt the method of moments proposed in Li et al. (2019).

We start by defining the one-step updates for the coupled trajectory and for SGD.

**Definition C.9** (One-Step Update of Coupled Trajectory). The one-step update for the coupled trajectory $\Delta$ can be written as

$$\Delta(\boldsymbol{y}, \boldsymbol{m}, C) = -\eta\boldsymbol{g}\left(\boldsymbol{y} + C\eta^{1-\alpha}\boldsymbol{m}\right)$$

**Definition C.10** (One-Step Update of SGD). The one-step update for SGD $\widetilde{\Delta}$ can be written as

$$\widetilde{\Delta}(\boldsymbol{y}) = -\eta\boldsymbol{g}(\boldsymbol{y})$$

**Lemma C.11.** *Let $\Delta$ be the one-step update for the coupled trajectory and $\widetilde{\Delta}$ be the one-step update for SGD. Let $\boldsymbol{x}_k, \boldsymbol{m}_k$ be the $k$-th step of an SGDM run and $\boldsymbol{y}_k = \boldsymbol{x}_k - \frac{\bar{\eta}_k\beta_k}{1-\beta_k}\boldsymbol{m}_k$ be the coupled trajectory in our consideration. Then for any $C \in [0, \frac{\lambda_{\max}\eta_{\max}}{\lambda_{\min}^2}]$, there is function $J(T)$ independent of $\eta$, such that for all vector entries $(i, j)$,*

$$\left|\mathbb{E}[\widetilde{\Delta}_{(i)}(\boldsymbol{y}_k) - \Delta_{(i)}(\boldsymbol{y}_k, \boldsymbol{m}_k, C)|\boldsymbol{y}_k]\right| \leq \eta^{2-\alpha} J(T)\mathbb{E}[\|\boldsymbol{m}_k\| \, |\boldsymbol{y}_k]$$

$$\left|\mathbb{E}[\widetilde{\Delta}_{(i)}\widetilde{\Delta}_{(j)}(\boldsymbol{y}_k) - \Delta_{(i)}\Delta_{(j)}(\boldsymbol{y}_k, \boldsymbol{m}_k, C)|\boldsymbol{y}_k]\right|$$
$$\leq \eta^{2-\alpha} J(T)(\eta + \mathbb{E}[\|\boldsymbol{m}_k\| \, |\boldsymbol{y}_k](1 + \eta\|\boldsymbol{y}_k\|) + \eta^{2-\alpha}\mathbb{E}[\|\boldsymbol{m}_k\|^2 \, |\boldsymbol{y}_k])$$

*and*

$$\left|\mathbb{E}[\widetilde{\Delta}_{(i)}(\boldsymbol{y}_k) - \Delta_{(i)}(\boldsymbol{y}_k, \boldsymbol{m}_k, C)]\right| \leq \eta^{\frac{3-\alpha}{2}} J(T)$$
$$\left|\mathbb{E}[\widetilde{\Delta}_{(i)}\widetilde{\Delta}_{(j)}(\boldsymbol{y}_k) - \Delta_{(i)}\Delta_{(j)}(\boldsymbol{y}_k, \boldsymbol{m}_k, C)]\right| \leq \eta^{\frac{3-\alpha}{2}} J(T)$$

*for all $k \in [0, T/\eta]$.*

*Proof.* Recall that we use $L$ to denote the Lipschitz constant of $\nabla\mathcal{L}$ and $L_\Sigma$ to denote the Lipschitz constant of the $\Sigma$ in the Frobenius norm. We can write

$$\left|\mathbb{E}[\widetilde{\Delta}_{(i)}(\boldsymbol{y}_k) - \Delta_{(i)}(\boldsymbol{y}_k, \boldsymbol{m}_k, C)|\boldsymbol{y}_k]\right| = \eta\left|\mathbb{E}[\partial_{(i)}\mathcal{L}(\boldsymbol{y}_k) - \partial_{(i)}\mathcal{L}\left(\boldsymbol{y}_k + C\eta^{1-\alpha}\boldsymbol{m}_k\right)|\boldsymbol{y}_k]\right|$$
$$\leq LC\eta^{2-\alpha}\mathbb{E}[\|\boldsymbol{m}_k\| \, |\boldsymbol{y}_k]$$

where the second step uses the Lipschitzness of the loss gradient. The proof can be completed by noting $\mathbb{E}\|\boldsymbol{m}_k\| \leq \sqrt{\mathbb{E}\|\boldsymbol{m}_k\|^2} = O(\eta^{(\alpha-1)/2})$ (Lemma C.7).

$$\left|\mathbb{E}[\Delta_{(i)}\Delta_{(j)}(\boldsymbol{y}_k) - \widetilde{\Delta}_{(i)}\widetilde{\Delta}_{(j)}(\boldsymbol{y}_k, \boldsymbol{m}_k, C)|\boldsymbol{y}_k]\right|$$

$$=\mathbb{E}[\eta^2\partial_i\mathcal{L}\partial_j\mathcal{L}(\boldsymbol{y}_k) + \eta^2\sigma^2\Sigma_{ij}(\boldsymbol{y}_k)|\boldsymbol{y}_k] - \mathbb{E}\left[\eta^2\partial_i\mathcal{L}\partial_j\mathcal{L}\left(\boldsymbol{y}_k + C\eta^{1-\alpha}\boldsymbol{m}_k\right) + \eta^2\sigma^2\Sigma_{ij}(\boldsymbol{y}_k + C\eta^{1-\alpha}\boldsymbol{m}_k)|\boldsymbol{y}_k\right]$$

$$\leq\eta^{2-\alpha}C(L_\Sigma\mathbb{E}[\|\boldsymbol{m}_k\||\boldsymbol{y}_k] + \eta L(\|\nabla\mathcal{L}(0)\| + L\|\boldsymbol{y}_k\|\mathbb{E}[\|\boldsymbol{m}_k\||\boldsymbol{y}_k] + LC\eta^{1-\alpha}\mathbb{E}[\|\boldsymbol{m}_k\|^2|\boldsymbol{y}_k])).$$

Again by Lemma C.7, $\mathbb{E}\|\boldsymbol{m}_k\| = O(\eta^{(\alpha-1)/2})$, so we obtain the desired result. $\qquad\square$

**Lemma C.12.** *For any $m \geq 1$, $C \in [0, \frac{\lambda_{\max}\eta_{\max}}{\lambda_{\min}^2}]$, there is function $g$ independent of $\eta$ and $k$, such that*

$$\mathbb{E}[\|\Delta(\boldsymbol{y}_k, \boldsymbol{m}_k, C)\|^{2m}] \leq \eta^m g(m, T)$$
$$\mathbb{E}[\|\widetilde{\Delta}(\boldsymbol{y}_k)\|^{2m}] \leq \eta^m g(m, T)$$

*for all $k \leq T/\eta$.*

*Proof.* The second line directly follows from Lemma C.8. For the first line,

$$\mathbb{E}[\|\Delta(\boldsymbol{y}_k, \boldsymbol{m}_k, C)\|^{2m}] = \eta^{2m}\mathbb{E}\left\|\nabla\mathcal{L}(\boldsymbol{y}_k + C\eta^{1-\alpha}\boldsymbol{m}_k) + \sigma\boldsymbol{v}_k\right\|^{2m}$$
$$\leq \eta^{2m}2^{2m}(\mathbb{E}[\|\nabla\mathcal{L}(0)\| + L\|\boldsymbol{y}_k\| + LC\eta^{1-\alpha}\|\boldsymbol{m}_k\|])^{2m} + 2^{2m}\eta^m\mathbb{E}\|\boldsymbol{v}_k\|^{2m}$$
$$= O(\eta^m)$$

by Lemma C.8. $\qquad\square$

### C.1.3 MAIN RESULT

Below we state an intermediate theorem for proving the weak approximation.

**Theorem C.13.** *Let $T > 0$ and $N = \lfloor T/\eta \rfloor$. Let $(\boldsymbol{x}_k, \boldsymbol{m}_k)$ be the states of an SGDM run with the coupled trajectory $\boldsymbol{y}_k$ defined in Equation (15), and let $\widehat{\boldsymbol{z}}_k$ be an SGD run with start $\widehat{\boldsymbol{z}}_0 = \boldsymbol{y}_0$. Then $\boldsymbol{y}_k$ is a weak approximation of $\widehat{\boldsymbol{z}}_k$.*

Notice that the only difference between Theorem C.13 and our final result Theorem 3.5 is that $\boldsymbol{z}_k$ is an SGD process starting from $\boldsymbol{x}_0$ while $\widehat{\boldsymbol{z}}_k$ is an SGD process starting from $\boldsymbol{y}_0$.

Before proving Theorem C.13, we introduce the following lemma, which is an analog to Lemma 27 of Li et al. (2019), Lemma C.2 of Li et al. (2021a), and Lemma B.6 of Malladi et al. (2022). It shows that if the update rules for the two trajectories are close in all of their moments, then the test function value will also not change much after a single update from the same initial condition. Let $G^k$ be the set of functions that are $k$-times continuously differentiable and all its derivatives up to and including order $k$ has polynomial growth, and let $G = G^0$ be the set of functions with polynomial growth.

**Lemma C.14.** *Suppose $u \in G^3$. Then, there exists a constant $K_u(T)$ independent of $\eta$, such that for all $k \leq T/\eta$,*

$$\left|\mathbb{E}[u(\boldsymbol{y}_k + \Delta(\boldsymbol{y}_k, \boldsymbol{m}_k, \frac{\bar{\eta}_k\beta_k}{1-\beta_k}))] - \mathbb{E}[u(\boldsymbol{y}_k + \widetilde{\Delta}(\boldsymbol{y}_k))]\right| \leq K_u(T)\eta^{(3-\alpha)/2}$$

*Proof.* Since $u \in G^3$, we can find $K_0(\boldsymbol{x}) = k_1(1 + \|\boldsymbol{x}\|^{k_2}) \in G$ such that $u(\boldsymbol{x})$ is bounded by $K_0(\boldsymbol{x})$ and so are all the partial derivatives of $u$ up to order 3. For notation simplicity let

$\Delta = \Delta(\boldsymbol{y}_k, \boldsymbol{m}_k, \frac{\bar{\eta}_k \beta_k}{1 - \beta_k})$ and $\widetilde{\Delta} = \widetilde{\Delta}(\boldsymbol{y}_k)$. By Taylor's Theorem with Lagrange Remainder, we have

$$u(\boldsymbol{y}_k + \Delta) - u(\boldsymbol{y}_k + \widetilde{\Delta}) = \underbrace{\sum_{1 \le i \le d} \frac{\partial u(\boldsymbol{y}_k)}{\partial x_i} \left( \Delta_i - \widetilde{\Delta}_i \right)}_{B_1}$$

$$+ \underbrace{\frac{1}{2} \sum_{1 \le i_1, i_2 \le d} \frac{\partial^2 u(\boldsymbol{y}_k)}{\partial x_{i_1} \partial x_{i_2}} \left( \Delta_{i_1} \Delta_{i_2} - \widetilde{\Delta}_{i_1} \widetilde{\Delta}_{i_2} \right)}_{B_2}$$

$$+ R_3 - \widetilde{R}_3$$

The remainders $R_3, \widetilde{R}_3$ are

$$R_3 := \frac{1}{6} \sum_{1 \le i_1, i_2, i_3 \le d} \frac{\partial^3 u(\boldsymbol{y}_k + a\Delta)}{\partial x_{i_1} \partial x_{i_2} \partial x_{i_3}} \Delta_{i_1} \Delta_{i_2} \Delta_{i_3},$$

$$\widetilde{R}_3 := \frac{1}{6} \sum_{1 \le i_1, i_2, i_3 \le d} \frac{\partial^3 u(\boldsymbol{y}_k + \widetilde{a}\widetilde{\Delta})}{\partial x_{i_1} \partial x_{i_2} \partial x_{i_3}} \widetilde{\Delta}_{i_1} \widetilde{\Delta}_{i_2} \widetilde{\Delta}_{i_3}.$$

for some $a, \widetilde{a} \in [0, 1]$. We know from Lemma C.11 that

$$\mathbb{E}B_1 = \mathbb{E}\left[ \sum_{1 \le i \le d} \frac{\partial u(\boldsymbol{y}_k)}{\partial x_i} \mathbb{E}[\Delta_i - \widetilde{\Delta}_i | \boldsymbol{y}_k] \right]$$

$$\le \mathbb{E}\left[ \sum_{1 \le i \le d} \frac{\partial u(\boldsymbol{y}_k)}{\partial x_i} \eta^{2-\alpha} J(T) \mathbb{E}[\|\boldsymbol{m}_k\| \, |\boldsymbol{y}_k] \right]$$

$$\le \mathbb{E}(\eta^{2-\alpha} d k_1 J(T)(\|\boldsymbol{m}_k\| + \|\boldsymbol{m}_k\| \|\boldsymbol{y}_k\|^{k_2}))$$

$$= O(\eta^{\frac{3-\alpha}{2}}).$$

The last step is due to Lemma C.8. Also,

$$\mathbb{E}B_2 = \mathbb{E}\left[ \frac{1}{2} \sum_{1 \le i_1, i_2 \le d} \frac{\partial^2 u(\boldsymbol{y}_k)}{\partial x_{i_1} \partial x_{i_2}} \left( \mathbb{E}[\Delta_{i_1} \Delta_{i_2} - \widetilde{\Delta}_{i_1} \widetilde{\Delta}_{i_2} | \boldsymbol{y}_k] \right) \right]$$

$$\le \mathbb{E}(d^2 k_1 J(T) \eta^{2-\alpha} (1 + \|\boldsymbol{y}_k\|^{k_2})(\eta + \|\boldsymbol{m}_k\| (1 + \eta \|\boldsymbol{y}_k\|) + \eta^{2-\alpha} \|\boldsymbol{m}_k\|^2)))$$

$$= O(\eta^{\frac{3-\alpha}{2}}).$$

For $R_3$, by Cauchy-Schwarz inequality we have

$$\mathbb{E}[R_3] \le \frac{1}{6} \left( \sum_{i_j} \mathbb{E} \left| \frac{\partial^3 u(\boldsymbol{y}_k + a\Delta)}{\partial x_{i_1} \partial x_{i_2} \partial x_{i_3}} \right|^2 \right)^{1/2} \cdot \left( \mathbb{E}\|\Delta\|^6 \right)^{1/2}$$

$$\le \left( d^3 \mathbb{E} K_0^2(\boldsymbol{y}_k + a\Delta) \right)^{1/2} \cdot \eta^{3/2} g^{1/2}(m, T)$$

by Lemma C.12. For $K_0^2(\boldsymbol{y}_k + a\Delta)$, we can bound its expectation by

$$\mathbb{E}[K_0^2(\boldsymbol{y}_k + a\Delta)] \le k_1^2 \mathbb{E}(1 + \|\boldsymbol{y}_k + a\Delta\|^{k_2})^2$$

$$\le 2k_1^2 \left( 1 + 2^{2k_2 - 1} \mathbb{E}[\|\boldsymbol{y}_k\|^{2k_2} + \mathbb{E}\|\Delta\|^{2k_2}] \right)$$

$$= O(1).$$

Therefore $\mathbb{E}[R_3] = O(\eta^{3/2})$. An analogous argument bounds $\mathbb{E}[\widetilde{R}_3] = O(\eta^{3/2})$ as well. Thus, the entire Taylor expansion and remainders are bounded as desired.

$\square$

*Proof for Theorem C.13.* Let $\widehat{\boldsymbol{y}}_{j,k}$ be the trajectory defined by following the coupled trajectory for $j$ steps and then do $k - j$ steps SGD updates. So, $\widehat{\boldsymbol{y}}_{j,j+1} = \boldsymbol{y}_j + \widetilde{\Delta}(\boldsymbol{y}_j)$ and $\widehat{\boldsymbol{y}}_{j+1,j+1} = \boldsymbol{y}_j + \Delta(\boldsymbol{y}_j, \boldsymbol{m}_j, \frac{\bar{\eta}_j \beta_j}{1 - \beta_j})$. Let $h$ be the test function with at most polynomial growth. Then, we can write

$$|\mathbb{E}[h(\boldsymbol{z}_k) - \mathbb{E}[h(\boldsymbol{y}_k)]| = \sum_{j=0}^{k-1} (\mathbb{E}[h(\widehat{\boldsymbol{y}}_{j+1,k})] - \mathbb{E}[h(\widehat{\boldsymbol{y}}_{j,k})]) \tag{50}$$

Define $u(\boldsymbol{y}, s, t) = \mathbb{E}_{\widehat{\boldsymbol{y}} \sim \mathcal{P}(\boldsymbol{y}, s, t)}[h(\widehat{\boldsymbol{y}}_t)]$, where $\mathcal{P}(\boldsymbol{y}, s, t)$ is the distribution induced by starting from $\boldsymbol{y}$ at time $s$ and following the SGD updates until time $t$. Then,

$$|\mathbb{E}[h(\boldsymbol{z}_k) - \mathbb{E}[h(\boldsymbol{y}_k)]| \leq \sum_{j=0}^{k-1} |\mathbb{E}[u(\widehat{\boldsymbol{y}}_{j+1,j+1}, j+1, k)] - \mathbb{E}[u(\widehat{\boldsymbol{y}}_{j,j+1}, j+1, k)]| \tag{51}$$

Define $u_{j+1} = u(\boldsymbol{y}, j+1, k)$. Then,

$$|\mathbb{E}[h(\boldsymbol{z}_k) - \mathbb{E}[h(\boldsymbol{y}_k)]| \leq \sum_{j=0}^{k-1} |\mathbb{E}[u_{j+1}(\boldsymbol{y}_j + \widetilde{\Delta}(\boldsymbol{y}_j))] - \mathbb{E}[u_{j+1}(\boldsymbol{y}_j + \Delta(\boldsymbol{y}_j, \boldsymbol{m}_j, \frac{\bar{\eta}_j \beta_j}{1 - \beta_j}))]| \tag{52}$$

We will show that $u_{j+1} \in G^3$, so by Lemma C.14,

$$|\mathbb{E}[u_{j+1}(\boldsymbol{y}_j + \widetilde{\Delta}(\boldsymbol{y}_j))] - \mathbb{E}[u_{j+1}(\boldsymbol{y}_j + \Delta(\boldsymbol{y}_j, \boldsymbol{m}_j, \frac{\bar{\eta}_j \beta_j}{1 - \beta_j}))]| \leq K_{u_{j+1}}(T) \eta^{(3-\alpha)/2} \tag{53}$$

Then,

$$\begin{aligned}
|\mathbb{E}[h(\boldsymbol{z}_k) - \mathbb{E}[h(\boldsymbol{y}_k)]| &\leq \sum_{j=0}^{k-1} K_{u_{j+1}}(T) \eta^{(3-\alpha)/2} \\
&\leq k \sup_j K_{u_j}(T) \eta^{(3-\alpha)/2} \\
&\leq T \sup_j K_{u_j}(T) \eta^{(1-\alpha)/2}.
\end{aligned}$$

We will show that we can choose $K_{u_j}$ so that $\sup_j K_{u_j}(T)$ is bounded by some universal constant $K(T)$, and by definition $\boldsymbol{z}_k$ and $\boldsymbol{y}_k$ are order-$\gamma$ weak approximations of each other.

Finally, we show that $u_{j+1} \in G^3$, and there is an universal function $K_u \in G$ that $\|\partial^m u_{j+1}(\boldsymbol{x})\| \leq K_u(\boldsymbol{x})$ for all $j \leq T/\eta$ and $m = 0, 1, 2, 3$. Then from the proof of Lemma C.14 we know that there is an universal constant $K(T)$ that upper bounds all $K_{u_j}(T)$. The proof follows the same steps as that in Li et al. (2019) for Proposition 25, except that $u_j$ is defined for discrete SGD updates instead of continuous SDE updates.

Notice that $u_{j+1}(\boldsymbol{x}) = \mathbb{E}_{\boldsymbol{y} \sim \mathcal{P}(\boldsymbol{x}, j+1, k)} h(\boldsymbol{y})$ is the expected $h$ value after running SGD from $\boldsymbol{x}$ for $k - j - 1$ steps. Let $\{\boldsymbol{s}_i\}$ be such a process with $\boldsymbol{s}_{j+1} = \boldsymbol{x}$.

First we show that $\boldsymbol{s}_i$ have bounded moments, and the bound is universal for all $i \leq T/\eta$ with polynomial growth with respect to the initial point $\boldsymbol{x}$. For any $m$, we know

$$\begin{aligned}
\mathbb{E}(1 + \|\boldsymbol{s}_{i+1}\|^2)^m | \boldsymbol{s}_i &= \mathbb{E}(1 + \|\boldsymbol{s}_i - \bar{\eta}_i (\nabla \mathcal{L}(\boldsymbol{s}_i) + \sigma \boldsymbol{v}_i)\|^2)^m | \boldsymbol{s}_i \\
&= \mathbb{E}(1 + \|\boldsymbol{s}_i\|^2 + \bar{\eta}_i^2 \|\nabla \mathcal{L}(\boldsymbol{s}_i) + \sigma \boldsymbol{v}_i\|^2 - 2\bar{\eta}_i \boldsymbol{s}_i^\top (\nabla \mathcal{L}(\boldsymbol{s}_i) + \sigma \boldsymbol{v}_i))^m | \boldsymbol{s}_i
\end{aligned}$$

From the Lipschitzness we know $\|\nabla \mathcal{L}(\boldsymbol{s}_i)\| \leq \|\nabla \mathcal{L}(0)\| + L \|\boldsymbol{s}_i\|$ and $\|\Sigma(\boldsymbol{s}_i)\|_F \leq \|\Sigma(0)\|_F + L_\Sigma \|\boldsymbol{s}_i\|$. Let $\delta_i = \bar{\eta}_i^2 \|\nabla \mathcal{L}(\boldsymbol{s}_i) + \sigma \boldsymbol{v}_i\|^2 - 2\bar{\eta}_i \boldsymbol{s}_i^\top (\nabla \mathcal{L}(\boldsymbol{s}_i) + \sigma \boldsymbol{v}_i)$, so there is constant $C$ such that $\mathbb{E}\delta_i | \boldsymbol{s}_i \leq \eta C (1 + \|\boldsymbol{s}_i\|^2)$, $\mathbb{E}\delta_i^2 | \boldsymbol{s}_i \leq \eta C (1 + \|\boldsymbol{s}_i\|^2)^2$ and $\mathbb{E}\delta_i^m \leq \eta C (1 + \|\boldsymbol{s}_i\|^2)^m$. Then, similar to the SGDM case,

$$\begin{aligned}
\mathbb{E}(1 + \|\boldsymbol{s}_{i+1}\|^2)^m | \boldsymbol{s}_i &= \mathbb{E}(1 + \|\boldsymbol{s}_i\|^2 + \delta_i)^m | \boldsymbol{s}_i \\
&\leq (1 + \|\boldsymbol{s}_i\|^2)^m + m \mathbb{E}\delta_i (1 + \|\boldsymbol{s}_i\|^2)^{m-1} | \boldsymbol{s}_i \\
&\quad + 2^{m-1} \mathbb{E}\delta_i^2 ((1 + \|\boldsymbol{s}_i\|^2)^{m-2} + |\delta_i|^{m-2}) | \boldsymbol{s}_i \\
&\leq (1 + \|\boldsymbol{s}_i\|^2)^m (1 + \eta C (m + 2^m))
\end{aligned}$$

So $\mathbb{E}(1 + \|s_{i+1}\|^2)^m \leq \mathbb{E}(1 + \|s_{j+1}\|^2)^m (1 + \eta C(m + 2^m))^{i-j} \leq (1 + \|x\|^2)^m e^{TC(m+2^m)}$ for all $i, j \leq T/\eta$. Thus we proved that the moments of $s_i$ are bounded by a universal polynomial of the initial point $x$. Hence, as $h$ is bounded by a polynomial, $u_{j+1}(x) = \mathbb{E}_{s_k|s_{j+1}=x} h(s_k)$ is also uniformly bounded by a polynomial of $x$ independent of $\eta$.

Now we consider the derivatives of $u_{j+1}$. We use the notion of derivatives of a random variable in the $\mathcal{L}^2$ sense as in Li et al. (2019), e.g. $\partial_x v_i$ is defined in the sense that $\mathbb{E}\|\partial_x v_i\|^2 = \partial_x \operatorname{tr} \Sigma(s_i)$. Taking derivatives,

$$\mathbb{E}(1 + \|\partial_x s_{i+1}\|^2)^m|s_i = \mathbb{E}(1 + \|\partial_x s_i - \bar{\eta}_i(\nabla^2 \mathcal{L}(s_i)\partial_x s_i + \sigma \partial_x v_i)\|^2)^m|s_i$$
$$= \mathbb{E}(1 + \|\partial_x s_i\|^2 + \xi_i)^m|s_i$$

where $\xi_i = \bar{\eta}_i^2 \|\nabla^2 \mathcal{L}(s_i)\partial_x s_i + \sigma \partial_x v_i\|^2 - 2\bar{\eta}_i s_i^\top (\nabla^2 \mathcal{L}(s_i)\partial_x s_i + \sigma \partial_x v_i)$. Lipschitzness dictates that $\|\nabla^2 \mathcal{L}(s_i)\| \leq L$ and $\|\nabla \Sigma(s_i)\| \leq L_\Sigma$, so again there is constant $D$ such that $\mathbb{E}\xi_i|s_i \leq \eta D(1 + \|\partial_x s_i\|^2)$, $\mathbb{E}\xi_i^2|s_i \leq \eta D(1 + \|\partial_x s_i\|^2)^2$ and $\mathbb{E}\xi_i^m \leq \eta D(1 + \|\partial_x s_i\|^2)^m$. Similarly,

$$\mathbb{E}(1 + \|\partial_x s_{i+1}\|^2)^m = \mathbb{E}(1 + \|\partial_x s_i\|^2 + \xi_i)^m$$
$$\leq \mathbb{E}(1 + \|\partial_x s_i\|^2)^m + m\xi_i(1 + \|\partial_x s_i\|^2)^{m-1}$$
$$+ 2^{m-1}\xi_i^2((1 + \|\partial_x s_i\|^2)^{m-2} + |\xi_i|^{m-2})$$
$$\leq (1 + \|\partial_x s_i\|^2)^m (1 + \eta D(m + 2^m))$$

So $\mathbb{E}(1 + \|\partial_x s_{i+1}\|^2)^m \leq \mathbb{E}(1 + \|\partial_x s_{j+1}\|^2)^m (1 + \eta D(m + 2^m))^{i-j} \leq (1 + \|\partial_x x\|^2)^m e^{TD(m+2^m)} = (1 + d)^m e^{TD(m+2^m)}$ for all $i, j \leq T/\eta$. Thus we proved that the moments of $\partial_x s_i$ are also bounded by a universal polynomial of the initial point $x$. Hence, as $h$ has polynomial-growing derivatives, $\|\partial_x u_{j+1}(x)\| = \mathbb{E}_{s_k|s_{j+1}=x} \nabla h(s_k)^\top \partial_x s_k \leq ([\mathbb{E}\|\nabla h(s_k)\|^2][\mathbb{E}\|\partial_x s_k\|^2])^{1/2}$ is also uniformly bounded by a polynomial of $x$ independent of $\eta$.

Notice that the higher order derivatives of $s_i$ have similar forms of iterates, e.g.

$$\partial^b s_{i+1} = \partial_x^b s_i + \phi_i^b - \bar{\eta}_i(\nabla^2 \mathcal{L}(s_i)\partial_x^b s_i + \sigma \partial_x^b v_i)$$

where $\phi_i^b$ only relates with $\nabla^\beta \mathcal{L}$, $\nabla^\beta \Sigma$ and $\partial_x^\beta s_i$ for $\beta < b$. Notice that by assuming $\mathcal{L}$ and $\Sigma$ have polynomial-growing derivates up to and including the third order (Assumption 3.4), by induction we can prove that the process $\phi_i^b$ is universally bounded as a polynomial of derivates of order $< b$, so similarly by the Grönwall inequality we can obtain an universal bound for $\partial^\beta u_{j+1}$ for $\beta = 0, 1, 2, 3$. As the argument here is identical to that in the proof for Proposition 25, Li et al. (2019), we omit the fine details. $\square$

*Proof for Theorem 3.5.* From Theorem C.3, we know $x_k$ and $y_k$ are order-$(1 - \alpha)/2$-weak approximations of each other, and from Theorem C.13 we know $\widehat{z}_k$ and $y_k$ are order-$(1 - \alpha)/2$-weak approximations of each other. Furthermore, in the above proof we can see that $|\mathbb{E}[h(z_k) - h(\widehat{z}_k)]| = |\mathbb{E}[u_0(y_0) - u_0(x_0)]| = |\mathbb{E}[u_0(x_0 - \frac{\bar{\eta}_0 \beta_0}{1-\beta_0} m_0) - u_0(x_0)]|$, so by the similar Taylor expansion as in Lemma C.14, as $m_0$ has bounded moments (Assumption 3.3), we can see that $|\mathbb{E}[h(z_k) - h(\widehat{z}_k)]| = O(\eta^{1-\alpha})$. This shows that $z_k$ and $\widehat{z}_k$ are order-$(1 - \alpha)$-weak approximations of each other, and naturally they are order-$(1 - \alpha)/2$-weak approximations.

Then we conclude that $x_k$ and $z_k$ are order-$(1 - \alpha)/2$-weak approximations of each other. $\square$

## C.2 $\alpha \geq 1$

Following the idea of Stochastic Modified Equations (Li et al., 2019), the limiting distribution can be described by the law of the solution $X_t$ to an SDE under brownian motion $W_t$

$$\mathrm{d}X_t = -\lambda_t \nabla \mathcal{L}(X_t)\mathrm{d}t + \lambda_t \Sigma^{1/2}(X_t)\mathrm{d}W_t \tag{54}$$

for some rescaled learning rate schedule $\lambda_t$ that $\bar{\eta}_k \to \lambda_{k\eta}$ in the limit.

When $\alpha = 1$, the limit distribution of SGDM becomes

$$\mathrm{d}X_t = \lambda_t/\gamma_t \mathrm{d}M_t - \lambda_t \nabla \mathcal{L}(X_t)\mathrm{d}t + \lambda_t \Sigma^{1/2}(X_t)\mathrm{d}W_t \tag{55}$$

$$\mathrm{d}M_t = -\gamma_t M_t \mathrm{d}t + \gamma_t \nabla \mathcal{L}(X_t)\mathrm{d}t - \gamma_t \Sigma^{1/2}(X_t)\mathrm{d}W_t \tag{56}$$

with similarly $\gamma_t$ being the limit $(1 - \beta_k)/\eta \to \gamma_{k\eta}$. Here $M_t$ is the rescaled momentum process that induces a non-negligible impact on the original trajectory $X_t$.

Furthermore, when $\alpha > 1$, if we still stick to following $O(1/\eta)$ steps for any $\eta$, then the dynamics of trajectory will become trivial. In the limit $\eta \to 0$, as $m_k - m_0 = O(k\eta^\alpha) := O(T\eta^{\alpha-1}) \to 0$, the trajectory has limit $x_k = x_0 - \sum_{i=0}^{k-1} \eta_i m_0$ for all $k = O(1/\eta)$. This is different from the case $\alpha \le 1$ where there is always a non-trivial dynamics in $x_k$ for the same time scale $k = O(1/\eta)$, regardless of the $\alpha$ index. We can think of the phenomenon by considering the trajectory of SGDM on a quadratic loss landscape, and in this case the SGDM behaves like a Uhlenbeck-Ornstein process. When $\alpha < 1$, the direction of $x$ has a mixing time of $O(1/\eta)$ while the direction of $m$ has a shorter mixing time of $O(1/\eta^\alpha)$, while when $\alpha > 1$, both mixing time of $x$ and $m$ mixes at a time scale of $O(1/\eta^\alpha)$, so in $O(1/\eta)$ steps the trajectory is far from any stationary states.

Therefore in this regime we should only consider the case where $m_0 = 0$ to avoid the trajectory moving too far. By rescaling $m$ and considering $O(\eta^{-\frac{1+\alpha}{2}})$ steps, we would spot non-trivial behaviours of the SGDM trajectory. In this case the SGDM have a different tolerance on the noise scale $\sigma$.

# D    THE DYNAMICS OF SGDM IN $O(1/\eta^2)$ TIME

In this section we will present results that characterizes the behaviour of SGD and SGDM in $O(1/\eta^2)$ time. We call this setting the slow SDE regime in accordance with previous works (Gu et al., 2023).

## D.1    SLOW SDE INTRODUCTION

There are a line of works that discusses the slow SDE regime that emerges when SGD is trapped in the manifold of local minimizers. The phenomenon was introduced in Blanc et al. (2020) and studied more generally in Li et al. (2021b). In these works, the behavior of the trajectory near the manifold, found to be a sharpness minimization process for SGD, is thought to be responsible for the generalization behavior of the trajectory.

The observations in these works is that SGD should mimic a Uhlenbeck-Ornstein process along the *normal* direction of the manifold of minimizers. Each stochastic step in the normal direction contributes a very small movement in the tangent space. Over a long range of time, these small contributions accumulate into a drift.

To overcome the theoretical difficulties in analyzing these small contributions, Li et al. (2021b) analyzed a projection $\Phi$ applied to the iterate that maps a point near the manifold to a point on the manifold. $\Phi(X)$ is chosen to be the limit of gradient flow when starting from $X$. Then it is observed that when the learning rate is small enough, $\Phi(X) \approx X$, and the dynamics of $\Phi(X)$ provides an SDE on the manifold that marks the behaviour of SGD in this regime.

## D.2    SLOW SDE PRELIMINARIES

### D.2.1    THE PROJECTION MAP

We consider the case where the local minimizers of the loss $\mathcal{L}$ form a manifold $\Gamma$ that satisfy certain regularity conditions as Assumption 4.1. In this section, we fix a neighborhood $O_\Gamma$ of $\Gamma$ that is an attraction set under $\nabla \mathcal{L}$, and define $\phi(x, t) = x - \int_0^t \nabla \mathcal{L}(\phi(x, s)) \mathrm{d}s$ and $\Phi(x) := \lim_{t \to \infty} \phi(x, t)$. $\Phi(x)$ is well-defined for all $x \in O_\Gamma$ as indicated by Assumption 4.2. We call $\Phi$ the gradient projection map.

The most important property of the projection map is that its gradient is always orthogonal to the direction of gradient. We ultilize the following lemma from previous works.

**Lemma D.1** (Li et al. (2021b) Lemma C.2). *For all $x \in O_\Gamma$ there is $\partial \Phi(x) \nabla \mathcal{L}(x) = 0$.*

For technical simplicity, we choose compact set $K \subset O_\Gamma$ and only consider the dynamics of $x_k$ within the set $K$. Formally, for any dynamics $x_k$ with $x_0 \in \mathring{K}$, define the exiting stopping time $\tau = \min_k \{k > 0 : x_{k+1} \notin K\}$, and the truncated process $\hat{x}_k = x_{\min(k,\tau)}$; for any dynamics $X_t$ in

continuous time with $\boldsymbol{X}_0 \in \mathring{K}$, define the exiting stopping time $\tau = \inf\{t > 0 : \boldsymbol{X}_t \notin \mathring{K}\}$, and the truncated process $\widehat{\boldsymbol{X}_t} = \boldsymbol{X}_{\min(t,\tau)}$.

There are a few regularity conditions Katzenberger proved in the paper Katzenberger (1991):

**Lemma D.2.** *There are the following facts.*

1. *If the loss $\mathcal{L}$ is smooth, then $\Phi$ is third-order continuously differentiable on $O_\Gamma$.*

2. *For the distance function $d(\boldsymbol{x}, \Gamma) = \min_{\boldsymbol{y} \in \Gamma \cap K} \|\boldsymbol{y} - \boldsymbol{x}\|$, there exists a positive constant $C_K$ that*
$$\|\Phi(\boldsymbol{X}) - \boldsymbol{X}\| \le C_K d(\boldsymbol{X}, \Gamma)$$
   *for any $\boldsymbol{X} \in K$.*

3. *There exists a Lyaponuv function $h(\boldsymbol{X})$ on $K$ that*
   - *$h : K \to [0, \infty)$ is third-order continuously differentiable and $h(\boldsymbol{X}) = 0$ iff $\boldsymbol{X} \in \Gamma$.*
   - *For all $\boldsymbol{X} \in K$, $h(\boldsymbol{X}) \le c\langle \nabla h(\boldsymbol{X}), \nabla \mathcal{L}(\boldsymbol{X}) \rangle$ for some constant $c > 0$.*
   - *$d^2(\boldsymbol{X}, \Gamma) \le c' h(\boldsymbol{X})$ for some constant $c' > 0$.*

### D.2.2 THE KATZENBERGER PROCESS

We recap the notion of *Katzenberger processes* in Li et al. (2021b) and the characterization of the corresponding limiting diffusion based on Katzenberger's theorems (Katzenberger, 1991).

**Definition D.3** (Uniform metric). *The uniform metric between two functions $f, g : [0, \infty) \to \mathbb{R}^D$ is defined to be $d_U(f, g) = \sum_{T=1}^{\infty} 2^{-T} \min\{1, \sup_{t \in [0,T)} \|f(t) - g(t)\|_2\}$.*

For each $n \in \mathbb{N}$, let $A_n : \mathbb{R}_+ \to \mathbb{R}_+$ be a non-decreasing functions with $A_n(0) = 0$, and $\{Z_n(t)\}_{t \ge 0}$ be a $\mathbb{R}^\Xi$-valued stochastic process. In our context of SGD, given loss function $\mathcal{L} : \mathbb{R}^D \to \mathbb{R}$, noise function $\sigma : \mathbb{R}^D \to \mathbb{R}^{D \times \Xi}$ and initialization $x_{\text{init}} \in U$, we call the following stochastic process (57) a *Katzenberger process*

$$X_n(t) = x_{\text{init}} + \int_0^t \sigma(X_n(s)) \mathrm{d}Z_n(s) - \int_0^t \nabla \mathcal{L}(X_n(s)) \mathrm{d}A_n(s) \tag{57}$$

if as $n \to \infty$ the following conditions are satisfied:

1. $A_n$ increases infinitely fast, i.e., $\forall \epsilon > 0, \inf_{t \ge 0}(A_n(t + \epsilon) - A_n(t)) \to \infty$;

2. $Z_n$ converges in distribution to $Z$ in uniform metric.

**Theorem D.4** (Adapted from Theorem B.7 in Li et al. (2021b)). *Given a Katzenberger process $\{X_n(\cdot)\}_{n \in \mathbb{N}}$, if SDE (58) has a global solution $Y$ in $U$ with $Y(0) = \Phi(x_{\text{init}})$, then for any $t > 0$, $X_n(t)$ converges in distribution to $Y(t)$ as $n \to \infty$.*

$$Y(t) = \Phi(x_{\text{init}}) + \int_0^t \partial\Phi(Y(s))\sigma(Y(s))\mathrm{d}Z(s)$$
$$+ \int_0^t \frac{1}{2} \sum_{i,j \in [D]} \sum_{k,l \in [\Xi]} \partial_{ij}^2 \Phi(Y(s))\sigma_{ik}(Y(s))\sigma_{jl}(Y(s))]\mathrm{d}[Z]_{kl}(s). \tag{58}$$

We note that the global solution always exists if the manifold $\Gamma$ is compact. For the case where $\Gamma$ is not compact, we introduce a compact neighbourhood of $\Gamma$ and a stopping time later for our result. Our formulation is under the original framework of Katzenberger (1991) and the proof in Li et al. (2021b) can be easily adapted to Theorem D.4.

### D.2.3 THE CÀDLÀG PROCESS

A càdlàg process is a right continuous process that has a left limit everwhere. For real-value càdlàg semimartingale processes $X_t$ and $Y_t$, define $X_{t-} = \lim_{s \to t-} X_s$, and $\int_a^b X_s dY_s$ to be the process $\int_{a+}^{b+} X_{s-} dY_s$ for interval $a < b$. That is in the integral we do not count the jump of process $Y_s$ at $s = a$ but we count the jump at $s = b$. Then it's easy to see that

- $\int_a^b X_s \mathrm{d}Y_s + \int_b^c X_s \mathrm{d}Y_s = \int_a^c X_s \mathrm{d}Y_s$ for $a < b < c$.
- $Z_t = \int_0^t X_s \mathrm{d}Y_s$ is a càdlàg semimartingale if both $X_s$ and $Y_s$ are càdlàg semimartingales.

By a extension to higher dimensions, let

$$\mathrm{d}[X]_t = \mathrm{d}(X_t X_t^\top) - X_t(\mathrm{d}X_t)^\top - (\mathrm{d}X_t)X_t^\top$$

and

$$\mathrm{d}[X,Y]_t = \mathrm{d}(X_t Y_t^\top) - X_t(\mathrm{d}Y_t)^\top - (\mathrm{d}X_t)Y_t^\top$$

,

we know $[X]_t$ and $[X,Y]_t$ are actually matrix-valued processes with $\Delta[X]_t = (\Delta X_t)(\Delta X_t)^\top$ and $\Delta[X,Y]_t = (\Delta X_t)(\Delta Y_t)^\top$.

The generalized Ito's formula applies to a càdlàg semimartingale process (let $\partial^2 f(X)[M] = \sum_{i,j} M_{ij}\partial_{ij}f(X)$ for any matrix $M$) is given as

$$\mathrm{d}f(X_t) = \langle \partial f(X_t), \mathrm{d}X_t \rangle + \frac{1}{2}\partial^2 f(X_t)[d[X]_t] + \Delta f(X)_t - \langle \partial f(X_{t-}), \Delta X_t \rangle - \frac{1}{2}\partial^2 f(X_{t-})[\Delta X_t \Delta X_t^\top].$$

And integration by part

$$\mathrm{d}(XY)_t = X_t(dY_t) + (dX_t)Y_t + d[X,Y]_t.$$

These formulas will be useful in our proof of the main theorem.

### D.2.4   WEAK LIMIT FOR CÀDLÀG PROCESSES

As we are showing the weak limit for a càdlàg process as the solution of an SDE, the following theorem is useful. We use $\mathcal{C}([0,T],\mathbb{R}^d)$ to denote the set of càdlàg functions $X : [0,T] \to \mathbb{R}^d$.

**Theorem D.5** (Theorem 2.2 in Kurtz and Protter (1991)). *For each $n$, let $\boldsymbol{X}_t^{(n)}$ be a processes with path in $\mathcal{C}([0,T],\mathbb{R}^{d\times m})$ and let $\boldsymbol{Y}_t^{(n)}$ be a semi-martingale with sample path in $\mathcal{C}([0,T],\mathbb{R}^m)$ respectively. Define function $h_\delta(r) = (1 - \delta/r)^+$ and $\widetilde{Y}_t^{(n)} = Y_t^{(n)} - \sum_{s\leq t} h_\delta(|\Delta Y_s^{(n)}|)\Delta Y_s^{(n)}$ be the process with reduced jumps. Then $\widetilde{Y}_t^{(n)}$ is also a semi-martingale. If the expected quadratic variation of $\widetilde{Y}_t^{(n)}$ is bounded uniformly in $n$, and $(X_n, Y_n) \to (X, Y)$ in distribution under the uniform metric (Definition D.3) of $\mathcal{C}([0,T],\mathbb{R}^{d\times m} \times \mathbb{R}^m)$, then*

$$\left(X_n, Y_n, \int X_n \mathrm{d}Y_n\right) \to \left(X, Y, \int X \mathrm{d}Y\right)$$

*in distribution under the uniform metric of $\mathcal{C}([0,T],\mathbb{R}^{d\times m} \times \mathbb{R}^m \times \mathbb{R}^d)$.*

Therefore if $X^{(n)}$ is the solution of an SDE $X_t^{(n)} = X_0 + Z_t^{(n)} + \int_0^t F^{(n)}(X_s^{(n)})\mathrm{d}Y_s^{(n)}$, and if the tuple $(F^{(n)}(X_s), Y_s^{(n)}, Z_t^{(n)})$ converges for all $X_s$ to the process $(F(X_s), Y_s, 0)$, then by the above theorem we know the processes $(X^{(n)}, Y^{(n)}, Z^{(n)})$ are relative compact, and the solution to the SDE $X_t = X_0 + \int_0^t F(X_s)\mathrm{d}Y_s$ is the limit of $X_t^{(n)}$, as stated rigorously in Theorem 5.4, Kurtz and Protter (1991). This will be the main tool in finding the limiting dynamics of SGDM.

### D.3   THE MAIN RESULTS

We provide a more formal version of Theorem 4.5.

**Theorem D.6.** *Fix a compact set $k \subset O_\Gamma$, an initialization $\boldsymbol{x}_0 \in K$ and $\alpha \in (0,1)$. Consider the SGDM trajectory $(\boldsymbol{x}_k^{(n)})$ with hyperparameter schedule $(\eta_k^{(n)}, \beta_k^{(n)})$ scaled by $\eta^{(n)}$, noise scaling $\sigma^{(n)} = 1$ and initialization $(\boldsymbol{x}_0, \boldsymbol{m}_0)$ satisfy Assumption 3.3; SGD trajectory $(\boldsymbol{z}_k^{(n)})$ with learning rate schedule $(\eta_k^{(n)})$, noise scaling $1$ and initialization $\boldsymbol{z}_0 = \boldsymbol{x}_0$. Furthermore the hyperparameter schedules satisfy Assumptions 4.3 and 4.4. Define the process $\boldsymbol{X}_t^{(n)} = \boldsymbol{x}_{\lfloor t/(\eta^{(n)})^2 \rfloor} - \phi(\boldsymbol{x}_0, t/\eta^{(n)}) +$*

$\Phi(\boldsymbol{x}_0)$ and $\boldsymbol{Z}_t^{(n)} = \boldsymbol{z}_{\lfloor t/(\eta^{(n)})^2 \rfloor} - \phi(\boldsymbol{z}_0, t/\eta^{(n)}) + \Phi(\boldsymbol{z}_0)$, and stopping time $\tau_n = \inf\{t > 0 : \boldsymbol{X}_t^{(n)} \notin K\}, \psi_n = \inf\{t > 0 : \boldsymbol{Z}_t^{(n)} \notin K\}$. Then the processes $(\boldsymbol{X}_{t \wedge \tau_n}^{(n)}, \tau_n)$ and $(\boldsymbol{Z}_{t \wedge \psi_n}^{(n)}, \psi_n)$ are relative compact in $\mathcal{C}([0,T], \mathbb{R}^d) \times [0,T]$ with the uniform metric and they have the same unique limit point $(\boldsymbol{X}_{t \wedge \tau}, \tau)$ such that $\boldsymbol{X}_{t \wedge \tau} \in \Gamma$ almost surely for every $t > 0$, $\tau = \inf\{t > 0 : \boldsymbol{X}_t \notin K\}$, and

$$\boldsymbol{X}_t = \Phi(\boldsymbol{x}_0) + \int_0^t \lambda_t \partial \Phi(X_s) \boldsymbol{\Sigma}^{1/2}(X_s) \mathrm{d}W_s + \int_0^t \frac{\lambda_t^2}{2} \partial^2 \Phi(X_s)[\boldsymbol{\Sigma}(X_s)] \mathrm{d}s.$$

Note that for a sequence $\boldsymbol{x}_k$ we are considering the sequence $\boldsymbol{X}_t = \boldsymbol{x}_{\lfloor t/(\eta^{(n)})^2 \rfloor} - \phi(\boldsymbol{x}_0, t/\eta^{(n)}) + \Phi(\boldsymbol{x}_0)$ after rescaling time $k = \lfloor t/(\eta^{(n)})^2 \rfloor$. This adaptation is due to technical difficulty that the limit of $\boldsymbol{x}_k^{(n)}$ will lie on the manifold $\Gamma$ for any $t > 0$, but $\boldsymbol{x}_0^{(n)} = \boldsymbol{x}_0 \notin \Gamma$, so the limiting process will be continuous anywhere but zero. For the process $\boldsymbol{X}_t^{(n)} = \boldsymbol{x}_{\lfloor t/(\eta^{(n)})^2 \rfloor} - \phi(\boldsymbol{x}_0, t/\eta^{(n)}) + \Phi(\boldsymbol{x}_0)$ however, we know $\boldsymbol{X}_t^{(n)} \to \boldsymbol{x}_{\lfloor t/(\eta^{(n)})^2 \rfloor}$ for any $t > 0$ and $\boldsymbol{X}_0^{(n)} = \Phi(\boldsymbol{x}_0) \in \Gamma$, thereby we make the limit a continuous process while preseving the same limit for all $t > 0$.

### D.3.1 THE LIMIT OF SGD

Let us reparametrize the SGD process $\boldsymbol{z}_k$. Define the learning rate $\lambda_t^{(n)} = \frac{\eta_{\lfloor t/(\eta^{(n)})^2 \rfloor}^{(n)}}{\eta^{(n)}}$, and the SGD iterates is given by

$$\boldsymbol{g}_k = \nabla \mathcal{L}(\boldsymbol{x}_k) + \boldsymbol{\Sigma}^{1/2}(\boldsymbol{x}_k) \boldsymbol{\xi}_k, \tag{59}$$

$$\boldsymbol{z}_{k+1} = \boldsymbol{z}_k - \eta_k^{(n)} \boldsymbol{g}_k. \tag{60}$$

Here we reparameterize the noise $\boldsymbol{v}_k = \boldsymbol{\Sigma}^{1/2}(\boldsymbol{x}_k) \boldsymbol{\xi}_k$ so that $\boldsymbol{\xi}_k$ are independent, $\mathbb{E}\boldsymbol{\xi}_k = 0$ and $\mathbb{E}\boldsymbol{\xi}_k \boldsymbol{\xi}_k^\top = \boldsymbol{I}$. We also assume that the constants $C_m$ in Assumption 3.4 are defined here as $\mathbb{E} \|\boldsymbol{\xi}_k\|^m \leq C_m$.

Now define the stochastic process $A_n(t) = \sum_{k=0}^{\lfloor t/(\eta^{(n)})^2 \rfloor - 1} \eta_k^{(n)}$ and $Z_n(t) = \sum_{k=0}^{\lfloor t/(\eta^{(n)})^2 \rfloor - 1} \eta_k^{(n)} \boldsymbol{\xi}_k$, then Equation (60) can be written as a stochastic integral as

$$\boldsymbol{Z}_t^{(n)} = \boldsymbol{z}_0 - \int_0^t \nabla \mathcal{L}(\boldsymbol{Z}_t^{(n)}) \mathrm{d}A_n(t) + \boldsymbol{\Sigma}^{1/2}(\boldsymbol{Z}_t^{(n)}) \mathrm{d}Z_n(t). \tag{61}$$

with $\boldsymbol{Z}_t^{(n)} = \boldsymbol{z}_{\lfloor t/(\eta^{(n)})^2 \rfloor}$. Then we can characterize its limiting dynamics.

**Theorem D.7.** *The process $\boldsymbol{Z}_t^{(n)}$ is a Katzenberger process, and for any $t > 0$, $\boldsymbol{Z}_t^{(n)}$ converges in distribution to $\boldsymbol{Z}_t$ as $n \to \infty$ that*

$$\boldsymbol{Z}_t = \Phi(\boldsymbol{z}_0) + \int_0^t \lambda_t \partial \Phi(\boldsymbol{Z}_s) \boldsymbol{\Sigma}^{1/2}(\boldsymbol{Z}_s) \mathrm{d}W_s + \int_0^t \frac{\lambda_t^2}{2} \partial^2 \Phi(\boldsymbol{Z}_s)[\boldsymbol{\Sigma}(\boldsymbol{Z}_s)] \mathrm{d}s.$$

*Proof.* First we show that $\boldsymbol{Z}_t^{(n)}$ is a Katzenberger process. Note that

- $A_n$ increases infinitely fast: by Assumption 4.3, for all $s < t$,

$$A_n(t) - A_n(s) = \sum_{k=\lfloor s/(\eta^{(n)})^2 \rfloor}^{\lfloor t/(\eta^{(n)})^2 \rfloor - 1} \eta_k^{(n)} \to \sum_{k=\lfloor s/(\eta^{(n)})^2 \rfloor}^{\lfloor t/(\eta^{(n)})^2 \rfloor - 1} \eta^{(n)} \lambda_{k(\eta^{(n)})^2},$$

$$\sum_{k=\lfloor s/(\eta^{(n)})^2 \rfloor}^{\lfloor t/(\eta^{(n)})^2 \rfloor - 1} \eta^{(n)} \lambda_{k(\eta^{(n)})^2} \geq \left(\frac{s-t}{\eta^{(n)}} - \eta^{(n)}\right)\left(\inf_{t \in [0,T]} \lambda_t\right) \to \infty$$

as $\eta^{(n)} \to 0$.

- $Z_n(t)$ converges to $Z(t)$ that there is a brownian motion $W_t$ and

$$Z(t) = \int_0^t \lambda_s \mathrm{d}W_s.$$

This is shown with the standard central limit theorem. Let $W_n(t) = \int_0^t \frac{\mathrm{d}Z(t)}{\lambda_s}$ be the normalized martingale. By the standard central limit theorem (for instance Theorem 4.3.2 Whitt (2002)), $W_n(t) - W_n(s)$ has a limit as a gaussian distribution with variance $\sum_{k=\lfloor s/(\eta^{(n)})^2 \rfloor}^{\lfloor t/(\eta^{(n)})^2 \rfloor - 1} (\frac{\eta_k^{(n)}}{\lambda_{k(\eta^{(n)})^2}})^2 \to (t-s)$ by Assumption 4.3. Then $W_n(t)$ converges to a bronwian motion $W_t$ by Levy's characterization.

Therefore $\boldsymbol{Z}_t^{(n)}$ is a Katzenberger process, and by Theorem D.4 its limit is given by

$$\boldsymbol{Z}_t = \Phi(\boldsymbol{z}_0) + \int_0^t \lambda_t \partial\Phi(\boldsymbol{Z}_s)\boldsymbol{\Sigma}^{1/2}(\boldsymbol{Z}_s)\mathrm{d}W_s + \int_0^t \frac{\lambda_t^2}{2}\partial^2\Phi(\boldsymbol{Z}_s)[\boldsymbol{\Sigma}(\boldsymbol{Z}_s)]\mathrm{d}s.$$

$\square$

### D.3.2 SGDM when $\alpha < 1$

For the SGDM setting, we wish to extract the scale from the hyperparameters to make notations clear. Therefore we define

$$\lambda_t^n = \frac{\eta_k^{(n)}}{\eta^{(n)}}$$

$$\gamma_t^n = \frac{1 - \beta_k^{(n)}}{(\eta^{(n)})^\alpha},$$

$$k = \lfloor t/(\eta^{(n)})^2 \rfloor.$$

Then the original process

$$\boldsymbol{m}_{k+1}^{(n)} = \beta_k^{(n)}\boldsymbol{m}_k^{(n)} + (1 - \beta_k^{(n)})\boldsymbol{g}_k^{(n)} \tag{62}$$

$$\boldsymbol{x}_{k+1}^{(n)} = \boldsymbol{x}_k^{(n)} - \eta_k^{(n)}\boldsymbol{m}_{k+1}^{(n)} \tag{63}$$

can be rewritten into a SDE formulation. Similarly, we reparameterize the noise $\boldsymbol{v}_k = \boldsymbol{\Sigma}^{1/2}(\boldsymbol{x}_k)\boldsymbol{\xi}_k$ so that $\boldsymbol{\xi}_k$ are independent, $\mathbb{E}\boldsymbol{\xi}_k = 0$ and $\mathbb{E}\boldsymbol{\xi}_k\boldsymbol{\xi}_k^\top = \boldsymbol{I}$. Let the processes $Z_t^n = \eta^2\lfloor t/\eta^2 \rfloor$, and $Y_t^n = \eta \sum_{i=1}^{\lfloor t/\eta^2 \rfloor} \boldsymbol{\xi}_i$. By the previous convention, the process can be rewritten as (let $\eta = \eta^{(n)}$)

$$\begin{cases} \mathrm{d}X_t^n &= -\frac{\lambda_t}{\eta}(M_t^n\mathrm{d}Z_t^n + \eta^2\mathrm{d}M_t^n) \\ &= -\frac{\lambda_t}{\eta}((1 - \gamma_t\eta^\alpha)M_t^n\mathrm{d}Z_t^n + \gamma_t\eta^\alpha\nabla\mathcal{L}(X_t^n))\mathrm{d}Z_t^n - \gamma_t\lambda_t\eta^\alpha\boldsymbol{\Sigma}^{1/2}(X_t^n)\mathrm{d}Y_t^n \\ \mathrm{d}M_t^n &= -\frac{\gamma_t}{\eta^{2-\alpha}}(M_t^n - \nabla\mathcal{L}(X_t^n))\mathrm{d}Z_t^n + \frac{\gamma_t}{\eta^{1-\alpha}}\boldsymbol{\Sigma}^{1/2}(X_t^n)\mathrm{d}Y_t^n \end{cases} \tag{64}$$

Then $X_t^n = \boldsymbol{x}_{\lfloor t/\eta^2 \rfloor}^{(n)}$ and $M_t^n = \boldsymbol{m}_{\lfloor t/\eta^2 \rfloor}^{(n)}$.

Rewriting the second line gives

$$M_t^n\mathrm{d}Z_t^n = -\gamma_t^{-1}\eta^{2-\alpha}\mathrm{d}M_t^n + \nabla\mathcal{L}(X_t^n)\mathrm{d}Z_t^n + \eta\boldsymbol{\Sigma}^{1/2}(X_t^n)\mathrm{d}Y_t^n \tag{65}$$

So

$$dX_t^n = -\lambda_t\eta\mathrm{d}M_t^n + (\lambda_t/\gamma_t)\eta^{1-\alpha}\mathrm{d}M_t^n - \lambda_t\eta^{-1}\nabla\mathcal{L}(X_t^n)\mathrm{d}Z_t^n - \lambda_t\boldsymbol{\Sigma}^{1/2}(X_t^n)\mathrm{d}Y_t^n \tag{66}$$

Now consider the Ito's formula applied to the gradient projected process $\Phi(X_t^n)$. Fix $K \subset O_\Gamma$ be a compact neighbourhood of the manifold $\Gamma$. Since $\Phi(X)$ is only defined for $X \in O_\Gamma$, we take an arbitrary regular extension of $\Phi$ to the whole space. Fix time horizon $T > 0$, and let $\tau_n = \min(\inf\{t > 0 : X_t^n \notin \mathring{K}\}, T)$ to be the exiting time for the compact set $K \subset O_\Gamma$ we have chosen earlier. Since $X_0^n \in \mathring{K}$, we know $\tau_n \geq \eta^2$. We use $\chi_t^n = \mathbf{1}[t < \tau_n]$ to denote the

indicator process of the stopping time $\tau_n$. For any càdlàg semi-martingale $X_t$, $X_{t\wedge\tau_n}$ is a càdlàg semi-martingale that $\mathrm{d}X_{t\wedge\tau_n} = \chi_t^n \mathrm{d}X_t$.

For simplicity we omit the superscript $n$ unless necessary. Ito's formula on $\Phi(X_t)$ gives

$$
\begin{aligned}
\mathrm{d}\Phi(X_t) &= \partial\Phi(X_t)\mathrm{d}X_t + \frac{1}{2}\partial^2\Phi(X_t)[\mathrm{d}[X]_t] + \mathrm{d}\delta \\
&= \partial\Phi(X_t)((-\lambda_t\eta + \lambda_t\eta^{1-\alpha}/\gamma_t)\mathrm{d}M_t - \lambda_t\eta^{-1}\nabla\mathcal{L}(X_t)dZ_t - \lambda_t\boldsymbol{\Sigma}^{1/2}(X_t)dY_t) \\
&\quad + \frac{1}{2}\partial^2\Phi(X_t)[\mathrm{d}[X]_t] + \mathrm{d}\delta
\end{aligned}
$$

where $\delta$ are error terms as $\delta(0) = 0$ and

$$
\mathrm{d}\delta = \Delta\Phi(X_t) - \partial\Phi(X_{t-})\Delta X_t - \frac{1}{2}\partial^2\Phi(X_{t-})[\Delta X_t(\Delta X_t)^\top]
$$

For $X_t \in K$, there is always $\partial\Phi(X_t)\nabla\mathcal{L}(X_t) = 0$ by Lemma D.1, so consider the following process $\Phi_t$: $\Phi_0 = \Phi(X_0)$ and

$$
\mathrm{d}\Phi_t = \partial\Phi(X_t)((-\lambda_t\eta + \lambda_t\eta^{1-\alpha}/\gamma_t)\mathrm{d}M_t - \lambda_t\boldsymbol{\Sigma}^{1/2}(X_t)\mathrm{d}Y_t) + \frac{1}{2}\partial^2\Phi(X_t)[\mathrm{d}[X]_t] + \mathrm{d}\delta \quad (67)
$$

Then $\Phi_{t\wedge\tau_n} = \Phi(X_{t\wedge\tau_n})$.

In addition, direct calculations gives

$$
\begin{aligned}
[Z]_t &= \eta^2 Z_t, \\
[Y]_t &= \eta^2 \sum_{i=1}^{\lfloor t/\eta^2\rfloor} \xi_i\xi_i^\top, \mathbb{E}[Y]_t = Z_t \boldsymbol{I} \\
[Y,Z]_t &= \eta^2 Y_t, \\
\mathrm{d}[M]_t &= \gamma_t^2\eta^{2\alpha-2}(M_t - \nabla\mathcal{L}(X_t))(M_t - \nabla\mathcal{L}(X_t))^\top \mathrm{d}Z_t + \gamma_t^2\eta^{2\alpha-2}\boldsymbol{\Sigma}^{1/2}(X_t)\mathrm{d}[Y]_t\boldsymbol{\Sigma}^{1/2}(X_t) \\
&\quad - \gamma_t^2\eta^{2\alpha-1}(M_t - \nabla\mathcal{L}(X_t))dY_t^\top\boldsymbol{\Sigma}^{1/2}(X_t) - \gamma_t^2\eta^{2\alpha-1}\boldsymbol{\Sigma}^{1/2}(X_t)dY_t(M_t - \nabla\mathcal{L}(X_t))^\top, \\
\mathrm{d}[M,Z]_t &= -\gamma_t\eta^\alpha(M_t - \nabla\mathcal{L}(X_t))\mathrm{d}Z_t + \gamma_t\eta^{\alpha+1}\boldsymbol{\Sigma}^{1/2}(X_t)\mathrm{d}Y_t \\
d[X]_t &= \lambda_t^2(M_tM_t^\top dZ_t + \eta^2 d[M]_t + M_t d[Z,M]_t + d[M,Z]_t M_t^\top) \\
d[M,X]_t &= -\lambda_t(-\gamma_t\eta^{\alpha-1}(M_t - \nabla\mathcal{L}(X_t))M_t^\top \mathrm{d}Z_t + \gamma_t\eta^\alpha\boldsymbol{\Sigma}^{1/2}(X_t)\mathrm{d}Y_t M_t^\top) + \eta d[M]_t)
\end{aligned}
$$

### D.3.3 Control of the velocity processes

Notice that our process $X_t^n \in K$ is bounded for $t < \tau_n$. Therefore following regularies, for any continuous function $f: K \to \mathbb{R}$, $f(X_t^n)$ is bounded $t < \tau_n$. Also, notice that $M_{t\wedge\tau_n}^n$ has bounded moments:

**Lemma D.8.** *There exists constants $C_m^n$ such that $\mathbb{E}\left\|M_{t\wedge\tau_n}^n\right\|^m \le C_m^n$.*

*Proof.* This follows trivially from the iterate $\boldsymbol{m}_{k+1}^{(n)} = \beta_k^{(n)}\boldsymbol{m}_k^{(n)} + (1 - \beta_k^{(n)})\boldsymbol{g}_k^{(n)}$,

$$
\begin{aligned}
\mathbb{E}\left\|\boldsymbol{m}_{k+1}^{(n)}\right\|^m &\le 2^m\left\|\boldsymbol{m}_k^{(n)}\right\|^m + 2^m\left\|\boldsymbol{g}_k^{(n)}\right\|^m \\
&\le 2^m\left\|\boldsymbol{m}_k^{(n)}\right\|^m + 4^m\left\|\nabla\mathcal{L}(\boldsymbol{x}_k^{(n)})\right\|^m + 4^m\left\|\boldsymbol{v}_k^{(n)}\right\|^m.
\end{aligned}
$$

The term $\left\|\nabla\mathcal{L}(\boldsymbol{x}_k^{(n)})\right\|^m \le \sup_{\boldsymbol{x}\in K}\|\nabla\mathcal{L}(\boldsymbol{x})\|^m$ is bounded and $\left\|\boldsymbol{v}_k^{(n)}\right\|^m$ is bounded by Assumption 3.4. Therefore the lemma follows from the Grönwall inequality Lemma C.4. $\qquad\square$

**Lemma D.9.** *For all stopping time $t$ and function $f > 0$, $\int_0^t f(M_s, X_s)\mathrm{d}Z_s \le \int_0^t f(M_s, X_s)\mathrm{d}s$, and $\int_0^t f(M_s, X_s)\mathrm{d}Z_s \ge \int_0^t f(M_s, X_s)\mathrm{d}s - \eta^2 f(M_t, X_t)$.*

*Proof.* Note that the processes $M_s$ and $X_s$ only changes at jumps at $s = k\eta^2$, the result followed directly from the definition of $Z_s$. □

Next follows some facts are are useful in proving the theorems.

**Lemma D.10** ([Katzenberger](#) (1991) lemma 2.1 ). *Let $f, g : \mathbf{R} \to [0, \infty)$ be functions that $g$ is non-decreasing and $g(0) = 0$. Assume for constant $C > 0$ and all $t > 0$*

$$0 \le f(t) \le C + \int_0^t f(s-)\mathrm{d}g(s).$$

*Then $f(t) \le Ce^{g(t)}$ for all $t \ge 0$.*

*Proof.* In this case $g(t) \ge 0$. Expansion gives

$$
\begin{aligned}
f(t) &\le C(1 + \sum_{n=1}^{\infty} \int_0^t \int_0^{s_1-} \int_0^{s_2-} \cdots \mathrm{d}g(s_3)\mathrm{d}g(s_2)\mathrm{d}g(s_1)) \\
&\le C(1 + \sum_{n=1}^{\infty} \frac{1}{n!}g^n(t)) \\
&= Ce^{g(t)}.
\end{aligned}
$$

□

We may also encounter the case where $g$ is negative. Another form of Grönwall inequality is useful here.

**Lemma D.11** (Grönwall). *Let $f, g : \mathbf{R} \to [0, \infty)$ be functions that $g(0) = 0$. Assume for constant $C > 0$ and all $0 < s < t < T$*

$$f(t) \le f(s) - C \int_s^t f(r-)\mathrm{d}r + g(t) - g(s).$$

*Then $f(t) \le e^{-Ct}(f(0) + g(t)) + C \int_0^t e^{-C(t-s)}(g(t) - g(s))\mathrm{d}s$ for all $0 \le s \le T$.*

We need a form of Grönwall's inequality with our uncountinuous process $Z_t$

**Lemma D.12.** *Let $f : \mathbb{R} \to [0, \infty)$ be a non-decreasing function and $g : \mathbb{R} \times \mathbb{R} \to [0, \infty)$ be non-negative. Assume for constant $C > 0$ and all $0 < s < t < T$*

$$f(t) \le f(s) - C \int_s^t f(r-)\mathrm{d}Z_r^n + g(t, s).$$

*Then $f(t) \le e^{-C'Z_t^n}(f(0) + g(t, 0)) + C \int_0^t e^{-C'(Z_t^n - Z_s^n)}g(t, s)\mathrm{d}Z_s^n$ for all $0 \le s \le T$ with $C' = \eta^{-2}\log(1 + C\eta^2)$.*

**Lemma D.13.** $\int_0^t e^{CZ_s^n}dZ_s^n = \frac{\eta^2}{e^{C\eta^2}-1}(e^{CZ_t^n} - 1)$.

*Proof.* Directly calculation that $\sum_{i=0}^n c^i = \frac{1-c^{n+1}}{1-c}$ for $c = e^{C\eta^2}$. □

*Proof of Lemma D.12.* multiplying both sides with $e^{-C'(Z_t^n - Z_s^n)}$ and integration yields the result. □

Specifically for $g(t, s) = Z_t^n - Z_s^n$, the bound can be further simlified.

**Lemma D.14.** $\int_0^t e^{-C(Z_t^n - Z_s^n)}(Z_t^n - Z_s^n)\mathrm{d}Z_s^n \le C^{-2}$.

*Proof.*

$$\int_0^t e^{-C(Z_t^n - Z_s^n)}(Z_t^n - Z_s^n)\mathrm{d}Z_s^n \le \int_0^{\infty} e^{-Cz}z\mathrm{d}z = C^{-2}.$$

□

Another useful theorem is the Doob's martingale inequality.

**Lemma D.15.** *Let $X_t$ be a martingale for $t \in [0, T]$ whose sample path is almost surely right-continuous. Then for any $C > 0$, and $p \geq 1$,*

$$\Pr[\sup_{t \in [0,T]} |X_t| \geq C] \leq \frac{\mathbb{E}|X_T|^p}{C^p}.$$

*Furthermore, integration with $p = 2$ gives*

$$\mathbb{E} \sup_{t \in [0,T]} |X_t| \leq 2\sqrt{\mathbb{E}|X_T|^2}.$$

**Lemma D.16.** *For all $t > 0$ and $\alpha \in [0, 1)$,*

- $\lim_{n \to \infty} \eta^\beta \mathbb{E} \sup_{t \in [0,T]} \left\| M_{t \wedge \tau_n}^n \right\|^2 = 0$ *for any $\beta > 1 - \alpha$.*
  *moreover,* $\lim_{n \to \infty} \eta^{\beta/2} \mathbb{E} \sup_{t \in [0,T]} \left\| M_{t \wedge \tau_n}^n \right\| = 0$

- $\lim_{n \to \infty} \eta^\beta \mathbb{E} \int_0^{t \wedge \tau_n} \left\| M_{t \wedge \tau_n}^n \right\|^2 \mathrm{d}Z_s^n = 0$ *for any $\beta > 0$.*
  *moreover,* $\sup_n \mathbb{E} \int_0^{t \wedge \tau_n} \left\| M_{t \wedge \tau_n}^n \right\|^2 \mathrm{d}Z_s^n < \infty$.

*For $\alpha \in (0, 1)$,*

- $\lim_{n \to \infty} \eta^\beta \mathbb{E} \int_0^{t \wedge \tau_n} \| M_s^n \|^3 \mathrm{d}Z_s^n = 0$ *for any $\beta > 0$.*

*Proof.* Ito's formula on $\| M_t^n \|^2$ gives

$$
\begin{aligned}
d \| M_t^n \|^2 &= 2\langle M_t^n, dM_t^n \rangle + \operatorname{tr} d[M^n]_t \\
&= -2\gamma_t \eta^{\alpha-2}(\| M_t^n \|^2 - (M_t^n)^\top \nabla \mathcal{L}(X_t^n))\mathrm{d}Z_t^n + 2\gamma_t \eta^{\alpha-1}(M_t^n)^\top \boldsymbol{\Sigma}^{1/2}(X_t^n)\mathrm{d}Y_t^n \\
&\quad + \gamma_t^2 \eta^{2\alpha-2} \| M_t^n - \nabla \mathcal{L}(X_t^n) \|^2 \mathrm{d}Z_t^n + \gamma_t^2 \eta^{2\alpha-2} \operatorname{tr}(\boldsymbol{\Sigma}(X_t^n)\mathrm{d}[Y^n]_t) \\
&\quad - 2\gamma_t^2 \eta^{2\alpha-1} \operatorname{tr}((M_t^n - \nabla \mathcal{L}(X_t^n))^\top \boldsymbol{\Sigma}^{1/2}(X_t^n)\mathrm{d}Y_t^n) \\
&= \eta^{2\alpha-2}\mathrm{d}W_t^n - 2\gamma_t \eta^{\alpha-2}(\| M_t^n \|^2 - (M_t^n)^\top \nabla \mathcal{L}(X_t^n))\mathrm{d}Z_t^n \\
&\quad + \gamma_t^2 \eta^{2\alpha-2} \| M_t^n - \nabla \mathcal{L}(X_t^n) \|^2 \mathrm{d}Z_t^n + \gamma_t^2 \eta^{2\alpha-2} \operatorname{tr}(\boldsymbol{\Sigma}(X_t^n))\mathrm{d}Z_t^n.
\end{aligned}
$$

Let $W_t^n$ be the martingale that $W_0^n = 0$ and

$$
\begin{aligned}
\mathrm{d}W_t^n &= 2\gamma_t \eta^{1-\alpha}(M_t^n)^\top \boldsymbol{\Sigma}^{1/2}(X_t^n)\mathrm{d}Y_t^n + \gamma_t^2 \operatorname{tr}(\boldsymbol{\Sigma}(X_t^n)(\mathrm{d}[Y^n]_t - \boldsymbol{I}\mathrm{d}Z_t^n)) \\
&\quad - 2\gamma_t^2 \eta(M_t^n - \nabla \mathcal{L}(X_t^n))^\top \boldsymbol{\Sigma}^{1/2}(X_t^n)\mathrm{d}Y_t^n.
\end{aligned}
$$

When $\alpha > 0$, take the constant $C_1 = 2\sup_{X \in K} \|\nabla \mathcal{L}(X)\| (2 + \|\nabla \mathcal{L}(X)\|)$ and $C_2 = \sup_{X \in K} \operatorname{tr} \boldsymbol{\Sigma}(X) + C_1$. Since $\eta^{-\alpha} \| M_t^n \| \leq (\eta^{-1} + \eta^{1-2\alpha} \| M_t^n \|^2)/2$, for any $t > s > 0$ there is

$$
\begin{aligned}
\eta^\beta \left\| M_{t \wedge \tau_n}^n \right\|^2 &\leq \int_{s \wedge \tau_n}^{t \wedge \tau_n} (-2\lambda_{\min} \eta^{\beta+\alpha-2} \| M_r^n \|^2 + C_1 \eta^{\beta+\alpha-2} \| M_r^n \| + C_1 \eta^{\beta+2\alpha-2}(\| M_r^n \|^2 + 1))\mathrm{d}Z_r^n \\
&\quad + \eta^\beta \left\| M_{s \wedge \tau_n}^n \right\|^2 + \eta^{\beta+2\alpha-2} \int_{s \wedge \tau_n}^{t \wedge \tau_n} \lambda_r^2 \operatorname{tr}(\boldsymbol{\Sigma}(X_r^n))\mathrm{d}Z_r^n + \eta^{\beta+2\alpha-2}(W_{t \wedge \tau_n}^n - W_{s \wedge \tau_n}^n) \\
&\leq \eta^\beta \left\| M_{s \wedge \tau_n}^n \right\|^2 + \int_{s \wedge \tau_n}^{t \wedge \tau_n} (-2\lambda_{\min} \eta^{\alpha-2} + \frac{1}{2}C_1 \eta^{-1} + C_1 \eta^{2\alpha-2})\eta^\beta \| M_r^n \|^2 \mathrm{d}Z_r^n \\
&\quad + \eta^{2\alpha+\beta-3}C_2(Z_{t \wedge \tau_n}^n - Z_{s \wedge \tau_n}^n) + \eta^{2\alpha+\beta-2}(W_{t \wedge \tau_n}^n - W_{s \wedge \tau_n}^n)
\end{aligned}
$$

By the Doob's inequality Lemma D.15,

$$
\begin{aligned}
W_t^{s,n} &= W_{t \wedge \tau_n}^n - W_{s \wedge \tau_n}^n \\
&= \int_{s \wedge \tau_n}^{t \wedge \tau_n} 2\gamma_r \eta^{1-\alpha}(M_r^n)^\top \boldsymbol{\Sigma}^{1/2}(X_r^n)\mathrm{d}Y_r^n + \gamma_r^2 \operatorname{tr}(\boldsymbol{\Sigma}(X_r^n)(\mathrm{d}[Y^n]_r - \boldsymbol{I}\mathrm{d}Z_r^n)) \\
&\quad - 2\gamma_r^2 \eta(M_r^n - \nabla \mathcal{L}(X_r^n))^\top \boldsymbol{\Sigma}^{1/2}(X_r^n)\mathrm{d}Y_r^n
\end{aligned}
$$

is a martingale, so there is a universal constant $A$ that

$$\mathbb{E}\sup_{r\in[s,t]}|W_r^{s,n}| \le 2\sqrt{\mathbb{E}|W_t^{s,n}|^2}$$

$$\le 2A\sqrt{Z_{t\wedge\tau_n}^n - Z_{s\wedge\tau_n}^n + \eta^{2-2\alpha}\int_{s\wedge\tau_n}^{t\wedge\tau_n}\|M_r^n\|^2\,\mathrm{d}Z_r^n}$$

$$\le 2A\eta^{-1}(Z_{t\wedge\tau_n}^n - Z_{s\wedge\tau_n}^n + \eta^{2-2\alpha}\int_{s\wedge\tau_n}^{t\wedge\tau_n}\|M_r^n\|^2\,\mathrm{d}Z_r^n)$$

Therefore let $K_t^n = \eta^\beta\mathbb{E}\sup_{s\in[0,t]}\left\|M_{s\wedge\tau_n}^n\right\|^2$,

$$K_t^n \le K_s^n + \int_s^t \kappa_n K_r^n \mathrm{d}Z_r^n + (2A+C_2)\eta^{2\alpha+\beta-3}(Z_t^n - Z_s^n)$$

Here $\kappa_n = -2\lambda_{\min}\eta^{\alpha-2} + \frac{1}{2}C_1\eta^{-1} + C_1\eta^{2\alpha-2} + 2A\eta^{-1}$, then when $\eta \to 0$ eventually $\kappa_n < 0$. Then by the Grönwall inequality Lemma D.12, with $\kappa_n' = \eta^{-2}\log(1+|\kappa_n|\eta^2)$,

$$K_t^n \le \eta^{2\alpha+\beta-3}(2A+C_2)[e^{-\kappa_n'Z_t^n}Z_t^n + \int_0^t|\kappa_n|e^{-\kappa_n'(Z_t^n-Z_s^n)}(Z_t^n - Z_s^n)\mathrm{d}Z_s^n]$$

By Lemma D.14,

$$K_t^n \le \eta^{2\alpha+\beta-3}(2A+C_2)(e^{-\kappa_n'Z_t^n}Z_t^n + |\kappa_n|(\kappa_n')^{-2})$$

Taking the limit gives $\lim_{n\to\infty}K_t^n = \widetilde{O}(\eta^{3\alpha+\beta-1}) = 0$.

For $\alpha = 0$, there is

$$\eta^2 d\|M_t^n\|^2 = \mathrm{d}W_t^n - 2\gamma_t(\|M_t^n\|^2 - (M_t^n)^\top\nabla\mathcal{L}(X_t^n))\mathrm{d}Z_t^n$$
$$+ \gamma_t^2\|M_t^n - \nabla\mathcal{L}(X_t^n)\|^2\,\mathrm{d}Z_t^n + \gamma_t^2\operatorname{tr}(\boldsymbol{\Sigma}(X_t^n))\mathrm{d}Z_t^n$$
$$\le \mathrm{d}W_t^n - \gamma_t\|M_t^n\|^2\,\mathrm{d}Z_t^n + \|\nabla\mathcal{L}(X_t^n)\|^2\,\mathrm{d}Z_t^n + \operatorname{tr}(\boldsymbol{\Sigma}(X_t^n))\mathrm{d}Z_t^n.$$

For $K_t^n = \eta^\beta\mathbb{E}\sup_{s\in[0,t]}\left\|M_{s\wedge\tau_n}^n\right\|^2$, let $\iota_n = -\lambda_{\min}\eta^{-2} + 2A\eta^{-1}$ and some constant $C_3$, there is

$$K_t^n \le K_s^n + \int_s^t \iota_n K_r^n\mathrm{d}Z_r^n + C_3\eta_n^{\beta-3}(Z_t^n - Z_s^n)$$

Similarly by Lemma D.12 and Lemma D.14, $\lim_{n\to\infty}K_t^n = O(\eta^{\beta-1}) = 0$.

For any jump $\Delta M_t^n$, and $f(M_t^n) = \|M_t^n\|^3$ there is $\theta \in [0,1]$ that $M = \theta M_{t-}^n + (1-\theta)M_t^n$ and

$$\Delta f(M_t^n) - \langle\partial f(M_{t-}^n), M_t^n\rangle - \frac{1}{2}\partial^2 f(M_{t-}^n)[\Delta M_t^n(\Delta M_t^n)^\top]$$
$$= \frac{1}{6}\partial^3 f(M)[\Delta M_t^n, \Delta M_t^n, \Delta M_t^n]$$
$$= -\frac{1}{2}\frac{\langle M, \Delta M_t^n\rangle^3}{\|M\|^3} + \frac{3}{2}\frac{\langle M, \Delta M_t^n\rangle}{\|M\|}$$
$$\le 2\|\Delta M_t^n\|^3.$$

Ito's formula on $\|M_t^n\|^3$ gives

$$d\|M_t^n\|^3 \le 3\|M_t^n\|\langle M_t^n, dM_t^n\rangle + \frac{3}{2}(\|M_t^n\|\,d\operatorname{tr}[M_t^n] + \|M_t^n\|^{-1}\operatorname{tr}(M_t^n(M_t^n)^\top d[M_t^n])) + 2\|\Delta M_t^n\|^3$$

At $t = k\eta^2$, there is a jump for the process $M_t^n$ as $\Delta M_t^n = \gamma_t\eta^\alpha(-M_{t-}^n + \nabla\mathcal{L}(X_{t-}^n) + \boldsymbol{\Sigma}^{1/2}(X_{t-}^b)\xi_k)$, so for constant $C_4 = 12\sup_{X\in K}(\|\nabla\mathcal{L}(X)\| + 1 + \|\boldsymbol{\Sigma}^{1/2}(X)\|)^3$

$$\mathbb{E}\left\|\Delta M_{t\wedge\tau_n}^n\right\|^3 \le C_4\eta^{3\alpha-2}(\left\|M_{t\wedge\tau_n}^n\right\|^3 + 1)\mathrm{d}Z_{t\wedge\tau_n}^n$$

When $\alpha > 0$, let $J_t^n = \eta^\beta \mathbb{E} \|M_t^n\|^3$, as $\eta^{\beta+\alpha-2} \|M_t^n\| \le \eta^{3\beta/2+\alpha-2} \|M_t^n\|^2 + \eta^{\beta/2+\alpha-2}$, there is

$$J_t^n \le J_s^n - \int_s^t (3\lambda_{\min}\eta^{\alpha-2} - \eta^{\alpha-2+\beta/2}C_5)J_r^n \mathrm{d}Z_r^n + C_5\eta^{\alpha-2+\beta/2}(Z_t^n - Z_s^n)$$

where $C_5 = 3C_4 + 3(\sup_{t\le T} |\lambda_t|)(1 + \sup_{X\in K} \max(\|\nabla\mathcal{L}(X)\|))$ is some universal constant. By Lemma D.12 and Lemma D.14, there is

$$J_t^n \le C_5 Z_t^n O(\eta^{\beta/2}).$$

And the conclusion follows. $\qquad\square$

A simpler proof may be given by consider the process $(M_t^n)^{\otimes 3}$ instead of $\|M_t^n\|^3$, and the proof idea is similar using the Grönwall's inequality.

**Lemma D.17.** *There exist a universal constant $C$ such that*

- $\left\|\Delta M_{t\wedge\tau_n}^n\right\| \le C\eta^\alpha(\left\|M_{t\wedge\tau_n}^n\right\| + 1)$.

- $\left\|\Delta X_{t\wedge\tau_n}^n\right\| \le C\eta(\left\|M_{t\wedge\tau_n}^n\right\| + \eta^\alpha)$.

*Proof.* Direct from the iterations Equation (64). $\qquad\square$

**Lemma D.18.** *When $n \to \infty$ (or $\eta \to 0$), $\delta(t \wedge \tau_n) \to 0$ weakly for all $\alpha \in (0,1)$ and $t \in [0,T]$.*

We wish to generalize the result a little bit.

**Lemma D.19.** *For any function $f \in C^3(K)$, as $n \to 0$,*

$$\mathbb{E} \sup_{t\in[0,\tau_n\wedge T]} \sum_{s\in[0,t]} \left| \Delta f(X_s^n) - \partial f(X_{s-}^n)\Delta X_s^n - \frac{1}{2}\partial^2 f(X_{s-}^n)[\Delta X_s^n, \Delta X_s^n] \right| \to 0$$

*Proof.* $\Delta X_s^n$ and $\Delta f(X_s^n)$ are non-zero at times $s = k\eta^2$ for $k \in \mathbb{Z}^+$. By the mean value theorem, there is $\theta \in [0,1]$ that

$$\begin{aligned}
&\left|\Delta f(X_s^n) - \partial f(X_{s-}^n)\Delta X_s^n - \frac{1}{2}\partial^2 f(X_{s-}^n)[\Delta X_s^n, \Delta X_s^n]\right| \\
=&\frac{1}{6}|\partial^3 f(\theta X_{s-}^n + (1-\theta)X_s^n)[\Delta X_s^n, \Delta X_s^n, \Delta X_s^n]| \\
\le&\frac{1}{6}(\sup_{X\in K} \left\|\partial^3 f(X)\right\|_F) \|\Delta X_s^n\|^3.
\end{aligned}$$

Here the norm is defined for tensors as $\left\|\partial^3 f(X)\right\|_F = \sqrt{\sum_{ijk}(\partial_i\partial_j\partial_k f(X))^2}$. The first term $\sup_{X\in K} \left\|\partial^3 f(X)\right\|_F$ is a constant independent of $n$ (as $K$ is compact). Notice that

$$\Delta X_s^n = -\lambda_s\eta(M_{s-}^n + \Delta M_s^n) = -\lambda_s\eta M_s^n.$$

Therefore there exists a constant $C = \frac{1}{6}(\sup_{X\in K} \left\|\partial^3 f(X)\right\|_F)(\sup_{t\in[0,T]}(\lambda_t)^3)$ that

$$\begin{aligned}
&\mathbb{E} \sup_{t\in[0,\tau_n\wedge T]} \sum_{s\in[0,t]} \left|\Delta f(X_s^n) - \partial f(X_{s-}^n)\Delta X_s^n - \frac{1}{2}\partial^2 f(X_{s-}^n)[\Delta X_s^n, \Delta X_s^n]\right| \\
\le&C \cdot \mathbb{E} \sup_{t\in[0,\tau_n\wedge T]} \eta^3 \sum_{s=k\eta^2\in[0,t]} \|M_s^n\|^3 \\
\le&C \cdot \mathbb{E}\eta \int_0^{T\wedge\tau_n} \|M_s^n\|^3 \, \mathrm{d}Z_t^n.
\end{aligned}$$

From Lemma D.16 we know $\eta \int_0^{T\wedge\tau_n} \mathbb{E} \|M_s^n\|^3 \, \mathrm{d}Z_t^n \to 0$, so the proof is done. $\qquad\square$

*Proof for Lemma D.18.* The result follows by applying Lemma D.19 to every coordinate of $\Phi(X_t^n)$.
$\qquad\square$

**Lemma D.20.** *For $\alpha \in (0,1)$, $\lim_{n\to\infty} \eta^\beta \mathbb{E} \int_0^{t\wedge\tau_n} \left\| M_{t\wedge\tau_n}^n \right\|^2 \mathrm{d}Z_s^n = 0$ for any $\beta > -\alpha$. moreover, $\sup_n \eta^{-\alpha} \mathbb{E} \int_0^{t\wedge\tau_n} \left\| M_{t\wedge\tau_n}^n \right\|^2 \mathrm{d}Z_s^n < \infty$*

*Proof.* Consider the energy function $G(X_t, M_t) = 2\frac{\gamma_t}{\lambda_t}\mathcal{L}(X_t) + \eta^{1-\alpha}\left\| M_t \right\|^2$, there is

$$
\begin{aligned}
\mathbb{E}G(X_t, M_t) = {}& \mathbb{E}G(X_0, M_0) + \mathbb{E}\int_0^t 2\frac{\gamma_t}{\lambda_t}\nabla\mathcal{L}(X_t)dX_t + 2d(\frac{\gamma_t}{\lambda_t})(\mathcal{L}(X_t) + \Delta\mathcal{L}(X_t)) \\
& + \frac{\gamma_t}{\lambda_t}\nabla^2\mathcal{L}(X_t)d[X]_t + d\delta \\
& - 2\gamma_t\eta^{-1}(\| M_t^n \|^2 - (M_t^n)^\top\nabla\mathcal{L}(X_t^n))\mathrm{d}Z_t^n + \\
& + \gamma_t^2\eta^{\alpha-1}\| M_t^n - \nabla\mathcal{L}(X_t^n) \|^2 \mathrm{d}Z_t^n + \gamma_t^2\eta^{\alpha-1}\operatorname{tr}(\mathbf{\Sigma}(X_t^n)\mathrm{d}[Y^n]_t) \\
= {}& G(X_0, M_0) + A_t - \int_0^t 2\gamma_t\eta^{-1}\mathbb{E}\| M_t^n \|^2 \mathrm{d}Z_t^n \\
& + \int_0^t \gamma_t^2\eta^{\alpha-1}\mathbb{E}\| M_t^n - \nabla\mathcal{L}(X_t^n) \|^2 \mathrm{d}Z_t^n + \gamma_t^2\eta^{\alpha-1}\operatorname{tr}(\mathbf{\Sigma}(X_t^n)\mathrm{d}[Y^n]_t)
\end{aligned}
$$

Here $A_t$ is some uniformly bounded process. Multiply both sides by $\eta^{1-\alpha}$ gives

$$
\begin{aligned}
\int_0^t 2\gamma_t\eta^{-\alpha}\mathbb{E}\| M_t^n \|^2 \mathrm{d}Z_t^n \leq {}& \eta^{1-\alpha}\mathbb{E}G(X_0, M_0) + \eta^{1-\alpha}A_t \\
& + \int_0^t \gamma_t^2\mathbb{E}\| M_t^n - \nabla\mathcal{L}(X_t^n) \|^2 \mathrm{d}Z_t^n + \gamma_t^2\operatorname{tr}(\mathbf{\Sigma}(X_t^n)\mathrm{d}[Y^n]_t).
\end{aligned}
$$

From Lemma D.16 we know the right-hand-side is uniformly bounded in $n$, and the conclusion follows. $\qquad\square$

### D.3.4 CONVERGENCE TO THE MANIFOLD

We wish to show that the process $X_{t\wedge\tau_n}^n \to \Phi_{t\wedge\tau_n}^n$ as $n \to \infty$ for any $t > 0$. First we need to show that as the learning rate $\eta \to 0$, there is a distance function $d$ that $d(\Phi_t, \Gamma) \to 0$ weakly as a stochastic process.

**Lemma D.21.** *As $n \to \infty$, $d(\Phi_t, \Gamma) \to 0$ weakly for all $\alpha \in (0,1)$.*

*Proof.* By Lemma D.2, we need to prove that for all $T > 0$, $\sup_{t\leq T} h(X_{t\wedge\tau_n}) \to 0$ in probability.

Ito's formula on $h(X_{t\wedge\tau_n})$ gives for some process $\delta \to 0$ $(n \to \infty)$

$$
\begin{aligned}
dh(X_t) = {}& \langle\nabla h(X_t), dX_t\rangle + \frac{1}{2}\nabla^2 h(X_t)d[X]_t + d\delta \\
= {}& \langle\nabla h(X_t), -\lambda_t\eta\mathrm{d}M_t^n + (\lambda_t/\gamma_t)\eta^{1-\alpha}\mathrm{d}M_t^n - \lambda_t\eta^{-1}\nabla\mathcal{L}(X_t^n)\mathrm{d}Z_t^n - \lambda_t\mathbf{\Sigma}^{1/2}(X_t^n)\mathrm{d}Y_t^n\rangle \\
& + \frac{1}{2}\nabla^2 h(X_t)d[X]_t + d\delta
\end{aligned}
$$

Let the process

$$
dS_t = \langle\nabla h(X_t), -\lambda_t\eta\mathrm{d}M_t^n + (\lambda_t/\gamma_t)\eta^{1-\alpha}\mathrm{d}M_t^n - \lambda_t\mathbf{\Sigma}^{1/2}(X_t^n)\mathrm{d}Y_t^n\rangle + \frac{1}{2}\nabla^2 h(X_t)d[X]_t + d\delta,
$$

there is

$$
\begin{aligned}
dh(X_t) &= dS_t - \lambda_t\eta^{-1}\langle\nabla h(X_t), \nabla\mathcal{L}(X_t)\rangle dZ_t \\
&\leq dS_t - \lambda_{\min}c^{-1}\eta^{-1}h(X_t)dZ_t
\end{aligned}
$$

Therefore by Lemma D.12, for $\kappa = \eta^{-2}\log(1 + \lambda_{\min}c^{-1}\eta)$, there is

$$
h(X_t) \leq e^{-\kappa Z_t}S_t + (\lambda_{\min}c^{-1}\eta^{-1})\int_0^t e^{-\kappa(Z_t - Z_s)}(S_t - S_s)dZ_s
$$

Clearly $e^{-\kappa Z_t} S_t \to 0$. Furthermore we have for

$$A_t = \int_0^{t \wedge \tau_n} (-\lambda_t \eta + (\lambda_t/\gamma_t)\eta^{1-\alpha})\langle \nabla h(X_t), \mathrm{d}M_t^n \rangle$$

$$B_t = \int_0^{t \wedge \tau_n} \langle \nabla h(X_t), -\lambda_t \mathbf{\Sigma}^{1/2}(X_t^n)\mathrm{d}Y_t^n \rangle$$

$$C_t = \int_0^{t \wedge \tau_n} \frac{1}{2}\nabla^2 h(X_t) d[X]_t + d\delta$$

we know $S_t = A_t + B_t + C_t$.

First, we show $\eta^{-1}\int_0^t e^{-\kappa(Z_t - Z_s)}(A_t - A_s)dZ_s \to 0$. as $A_t - A_s \le \int_s^t K[\eta^{-1}(\|M_r\|+1)dZ_r + dY_r]$ for some constant $K$, we know by Lemma D.14 and Lemma D.16,

$$\eta^{-2}\sup_r \|M_r\| \left| \int_0^t e^{-\kappa(Z_t - Z_s)}(Z_t - Z_s)dZ_s \right| \le \eta^{-2}\kappa^{-2}\sup_r \|M_r\| \le \eta^2 \sup_r \|M_r\| \to 0$$

and since $\mathbb{E}\sup_r \|Y_t - Y_s\| \le 2\sqrt{\mathbb{E}\|Y_t\|^2} \le Kt$, there is $\mathbb{E}\sup_n \eta^{-1}|\int_0^t e^{-\kappa(Z_t - Z_s)}(Y_t - Y_s)dZ_s| \le \frac{Kt}{\kappa\eta} \to 0$.

Next, we show $\eta^{-1}\int_0^t e^{-\kappa(Z_t - Z_s)}(B_t - B_s)dZ_s \to 0$. $B_t - B_s$ is a martingale so by Doob's inequality, $\mathbb{E}\sup_r |B_t - B_r| \le 2\sqrt{\mathbb{E}|B_t|^2} \le Kt$ for some constants $K$. Therefore $\mathbb{E}\sup_n \eta^{-1}|\int_0^t e^{-\kappa(Z_t - Z_s)}(B_t - B_s)dZ_s| \le \frac{Kt}{\kappa\eta} \to 0$.

Finally, there exists constant $K$ such that $\eta^{-1}\int_0^t e^{-\kappa(Z_t - Z_s)}(C_t - C_s)dZ_s \le \eta^{-1}\int_0^t e^{-\kappa(Z_t - Z_s)}K(Z_t - Z_s)dZ_s \le \frac{K}{\eta\kappa^2} \to 0$ by Lemma D.14. Therefore we concludes the proof by showing that $h(X_t) \to 0$. $\square$

## D.4 Averaging

**Lemma D.22.** $\lim_{n \to \infty} \eta^\beta \mathbb{E}\sup_{t \le T} |\int_0^{t \wedge \tau_n} \partial^2 \Phi(X_s^n)[\nabla \mathcal{L}(X_s^n)(M_s^n)^\top]\mathrm{d}Z_s^n| = 0$ for any $\beta > -1$.

To prove the result we need another lemma.

**Lemma D.23.** $\lim_{n \to \infty} \eta^\beta \mathbb{E}\sup_{t \le T} |\int_0^{t \wedge \tau_n} \|\nabla \mathcal{L}(X_s^n)\|^2 \mathrm{d}Z_s^n| = 0$ for any $\beta > -1$.

*Proof.* Use Ito on $\langle \nabla \mathcal{L}(X_t), M_t \rangle$, there is

$$d\langle \nabla \mathcal{L}(X_t), M_t \rangle = -\lambda_t \partial^2 \mathcal{L}(X_t)[\eta^{-1}M_t M_t^\top dZ_t + \eta M_t dM_t^\top] + \langle \Delta\nabla\mathcal{L}(X_t), \Delta M_t \rangle$$
$$+ \gamma_t \nabla\mathcal{L}(X_t)[\eta^{-2+\alpha}(\nabla\mathcal{L}(X_t) - M_t)dZ_t + \eta^{-1+\alpha}\mathbf{\Sigma}^{1/2}(X_t)dY_t].$$

Therefore we know there exists constant $K$ such that

$$\mathbb{E}\int_0^{t \wedge \tau_n} \|\nabla\mathcal{L}(X_s)\|^2 dZ_s \le K\eta^{2-\alpha}\mathbb{E}(\langle \nabla\mathcal{L}(X_{t\wedge\tau_n}), M_{t\wedge\tau_n} \rangle - \langle \nabla\mathcal{L}(X_0), M_0 \rangle)$$

$$+ \eta^{1-\alpha}K\mathbb{E}\int_0^{t\wedge\tau_n} \|M_s\|^2 dZ_s + K\eta^3 \mathbb{E}\sum_{\Delta M_t \neq 0} \|M_t\|^2$$

$$+ \mathbb{E}\int_0^{t\wedge\tau_n} \langle \nabla\mathcal{L}(X_t), M_t \rangle dZ_t.$$

The first four terms vanishes when multiplied $\eta^\beta$ for $\beta > -1$ by Lemma D.20. Note the last term

$$\int_0^{t\wedge\tau_n} \langle \nabla\mathcal{L}(X_t), M_t \rangle dZ_t = \int_0^{t\wedge\tau_n} \langle \nabla\mathcal{L}(X_t), -\eta\lambda_t^{-1}dX_t - \eta^2 dM_t \rangle$$

so

$$\lambda_t^{-1}\langle \nabla\mathcal{L}(X_t), dX_t \rangle = d(\lambda_t^{-1}\mathcal{L}(X_t)) - (\Delta\lambda_t^{-1})(\Delta\mathcal{L}(X_t) + \mathcal{L}(X_t)) - \frac{1}{2}\lambda_t^{-1}\partial^2\mathcal{L}(X_t)[d[X]_t] - d\delta.$$

$\lambda_t^{-1}\langle \nabla\mathcal{L}(X_t), dX_t \rangle$ is clearly a bounded process given the bounded variation of $\lambda_t^{-1}$ and boundedness of $X_{t\wedge\tau_n}$. Thereby we finished the proof. $\square$

*Proof of Lemma D.22.* Let $f(X_s) = \partial^2 \Phi(X_s) \nabla \mathcal{L}(X_s)$.

Ito's formula on $\partial^2 \Phi(X_s)[\nabla \mathcal{L}(X_s)(M_s)^\top] = f(X_s) M_s$ gives

$$
\begin{aligned}
d(f(X_s) M_s) &= df(X_s) M_s + f(X_s) dM_s + \Delta f(X_s) \Delta M_s \\
&= df(X_s) M_s + \Delta f(X_s) \Delta M_s \\
&\quad - \gamma_s \eta^{\alpha-2} f(X_s)(M_s - \nabla \mathcal{L}(X_s)) dZ_s + \gamma_s \eta^{\alpha-1} f(X_s) \mathbf{\Sigma}^{1/2}(X_s) dY_s.
\end{aligned}
$$

Therefore there is constant $K$ such that

$$
\int_0^t f(X_s) M_s dZ_s \leq \int_0^t f(X_s) \nabla \mathcal{L}(X_s) dZ_s
$$

$$
+ K(\int_0^t \eta^{2-\alpha} df(X_s) M_s + \eta f(X_s) \mathbf{\Sigma}^{1/2}(X_s) dY_s + \eta^{2-\alpha} \sum \Delta f(X_s) \Delta M_s)
$$

We know that $\eta^{2-\alpha} \sum \Delta f(X_s) \Delta M_s = O(\eta)$ and $\sup_t \int_0^t \eta f(X_s) \mathbf{\Sigma}^{1/2}(X_s) dY_s = O(\eta)$. Expansion gives $\int_0^t \eta^{2-\alpha} df(X_s) M_s = O(\eta)$ by Lemma D.20. Finally by Lemma D.23 we obtain the desired result. $\qquad\square$

Let $\phi_t = \lambda_t / \gamma_t - \eta^\alpha \lambda_t$.

**Lemma D.24.** $\int_{c_1}^{c_2} [\eta^{1-\alpha} \phi_t \partial \Phi(X_t) \mathrm{d}M_t + \phi_t \lambda_t (\eta^{-\alpha} - \lambda_t) \partial^2 \Phi(X_t)[M_t M_t^\top] \mathrm{d}Z_t \to 0$ as $\eta \to 0$.

*Proof.* Ito's formula on $\eta^{1-\alpha} \phi_t \partial \Phi(X_t) M_t$ gives

$$
\begin{aligned}
\mathrm{d}(\phi_t \partial \Phi(X_t) M_t) &= \mathrm{d}(\phi_t) \partial \Phi(X_t) M_t + (\Delta \phi_t) \Delta(\partial \Phi(X_{t-}) M_t) + \phi_t \partial \Phi(X_t) \mathrm{d}M_t \\
&\quad + \phi_t \partial^2 \Phi(X_t)[M_t \mathrm{d}X_t^\top] + \phi_t \Delta \partial \Phi(X_t) \Delta M_t + \mathrm{d}\delta
\end{aligned}
$$

where

$$
\mathrm{d}\delta = \phi_t (\Delta \partial \Phi(X_t) - \partial^2 \Phi(X_{t-}) \Delta X_t) M_t
$$

We know for $\alpha \in (0, 1)$, by Lemma D.16

- $\eta^{1-\alpha} \delta \to 0$ as $|\delta_t| \leq C \int_0^t \|M_s\|^3 \mathrm{d}Z_s$.

- $\int \eta^{1-\alpha} \mathrm{d}(\phi_t) \partial \Phi(X_t) M_t \to 0$ as $\int \mathrm{d}(\phi_t) < \infty$ by Assumption 4.4 and $\sup_t \eta^{1-\alpha} \|M_t\| \to 0$.

- $\sum \eta^{1-\alpha} (\Delta \phi_t) \Delta(\partial \Phi(X_{t-}) M_t) \to 0$ as $\sum_t \Delta \phi_t < \infty$ by Assumption 4.4 and $\int \eta^{1-\alpha} \|M_t\| \to 0$.

- $\int \eta^{1-\alpha} \phi_t \partial^2 \Phi(X_t)[M_t \mathrm{d}X_t^\top] + \int \eta^{-\alpha} \phi_t \lambda_t \partial^2 \Phi(X_t)[M_t M_t^\top] \mathrm{d}Z_t \to 0$

- $\sum \eta^{1-\alpha} \phi_t \Delta \partial \Phi(X_t) \Delta M_t - \int \phi_t \lambda_t \gamma_t \partial^2 \Phi(X_t)[M_t, M_t - \nabla \mathcal{L}(X_t)] \mathrm{d}Z_t \to 0$

- $\eta^{1-\alpha} \phi_t \partial \Phi(X_t) M_t \to 0$

Therefore adding them up we obtain the result. $\qquad\square$

**Lemma D.25.** $|\int_0^{t \wedge \tau_n} \partial^2 \Phi(X_s)[\mathrm{d}[X]_s] - \int_0^{t \wedge \tau_n} \lambda_t^2 \partial^2 \Phi(X_s)[M_t M_t^\top] \mathrm{d}Z_t| \to 0$.

*Proof.* This follows directly by the expansion of $[X]_t$ and Lemma D.16. $\qquad\square$

**Lemma D.26.** $\sup_{c_1,c_2} \left| \int_{c_1}^{c_2} \frac{\lambda_t^2/\gamma_t}{\eta^\alpha} \partial^2 \Phi(X_t)[M_t, M_t] \mathrm{d}Z_t - \int_{c_1}^{c_2} \frac{\lambda_t^2}{2} \partial^2 \Phi(X_t)[\mathbf{\Sigma}(X_t)] \mathrm{d}t \right| \to 0$ as $\eta \to 0$.

*Proof.* let $A_t = \phi_t \partial^2 \Phi(X_t)[M_t, M_t]$ for some schedule $\phi_t$, then for some uniformly bounded process $B_t$,

$$
\begin{aligned}
\mathrm{d}A_t = & \, d(\phi_t)(\partial^2 \Phi(X_t)[M_t, M_t] + \Delta \partial^2 \Phi(X_t)[M_t, M_t]) + \eta^{3\alpha-2} dB_t \\
& - \frac{\phi_t \lambda_t}{\eta} \partial^3 \Phi(X_t)[M_t, M_t, M_t] \mathrm{d}Z_t + \frac{\gamma_t^2 \phi_t}{\eta^{2-2\alpha}} \partial^2 \Phi(X_t)[\boldsymbol{\Sigma}(X_t)^{1/2} \mathrm{d}[Y]_t \boldsymbol{\Sigma}(X_t)^{1/2}] \\
& - \frac{2\phi_t \gamma_t}{\eta^{2-\alpha}} \partial^2 \Phi(X_t)[M_t, M_t] \mathrm{d}Z_t + \frac{2\phi_t \gamma_t}{\eta^{2-\alpha}} \partial^2 \Phi(X_t)[\nabla L(X_t), M_t] \mathrm{d}Z_t \\
& + \frac{2\phi_t \gamma_t}{\eta^{1-\alpha}} \partial^2 \Phi(X_t)[\boldsymbol{\Sigma}^{1/2}(X_t) \mathrm{d}Y_t, M_t] \\
& - \frac{2\phi_t \gamma_t^2}{\eta^{1-2\alpha}} \partial^2 \Phi(X_t)[(M_t - \nabla \mathcal{L}(X_t)) dY_t^\top \boldsymbol{\Sigma}^{1/2}(X_t)] \\
& + \gamma_t^2 \eta^{2\alpha-2} \phi_t \partial^2 \Phi(X_t)[(M_t - \nabla \mathcal{L}(X_t))(M_t - \nabla \mathcal{L}(X_t))^\top] \mathrm{d}Z_t.
\end{aligned}
$$

Multiply both sides by $\eta^{2\alpha-2}$, by Lemmas D.22 and D.23 we know $\int \partial^2 \Phi(X_t)[\nabla \mathcal{L}(X_t) M_t^\top] \mathrm{d}Z_t$ and $\int \partial^2 \Phi(X_t)[\nabla \mathcal{L}(X_t) \nabla \mathcal{L}(X_t)^\top] \mathrm{d}Z_t$ converges to 0. Bound the Martingale $W_t$ that

$$
dW_t = 2\phi_t \gamma_t \eta^{1-\alpha} \partial^2 \Phi(X_t)[\boldsymbol{\Sigma}^{1/2}(X_t) \mathrm{d}Y_t, M_t] - 2\phi_t \gamma_t^2 \eta \partial^2 \Phi(X_t)[(M_t - \nabla \mathcal{L}(X_t)) dY_t^\top \boldsymbol{\Sigma}^{1/2}(X_t)],
$$

Doob's inequality alongside with Lemma D.16 shows $W_t \to 0$. Then we know with $\phi_t = \lambda_t^2/\gamma_t^2$ that

$$
\sup_{c_1, c_2} \left| \int_{c_1}^{c_2} \frac{\lambda_t^2/\gamma_t}{\eta^\alpha} \partial^2 \Phi(X_t)[M_t, M_t] \mathrm{d}Z_t - \int_{c_1}^{c_2} \frac{\lambda_t^2}{2} \partial^2 \Phi(X_t)[\boldsymbol{\Sigma}(X_t)] \mathrm{d}t \right| \to 0
$$

$\square$

Finally we are ready to show the limiting dynamics as

**Theorem D.27.** *For any $t > 0$, $(X_{t \wedge \tau_n}^n, \tau_n)$ converges in distribution to $(X_{t \wedge \tau}, \tau)$ that $\tau = \inf\{t > 0 : X_t \notin K\}$, and that*

$$
X_t = \Phi(\boldsymbol{x}_0) + \int_0^t \lambda_t \partial \Phi(X_s) \boldsymbol{\Sigma}^{1/2}(X_s) \mathrm{d}W_s + \int_0^t \frac{\lambda_t^2}{2} \partial^2 \Phi(X_s)[\boldsymbol{\Sigma}(X_s)] \mathrm{d}s.
$$

*Proof.* Recall the process $\Phi_t^n$ as

$$
\mathrm{d}\Phi_t^n = \partial \Phi(X_t^n)((-\lambda_t \eta + \lambda_t \eta^{1-\alpha}/\gamma_t) \mathrm{d}M_t^n - \lambda_t \boldsymbol{\Sigma}^{1/2}(X_t^n) dY_t^n) + \frac{1}{2} \partial^2 \Phi(X_t^n)[\mathrm{d}[X^n]_t] + \mathrm{d}\delta_t^n
$$

Therefore we know

$$
X_{t \wedge \tau_n}^n = \Phi(\boldsymbol{x}_0) + (X_{t \wedge \tau_n}^n - \Phi_{t \wedge \tau_n}^n) + \delta_t^n \tag{68}
$$
$$
+ \int_0^{t \wedge \tau_n} \partial \Phi(X_t^n)((-\lambda_t \eta + \lambda_t \eta^{1-\alpha}/\gamma_t) \mathrm{d}M_t^n - \lambda_t \boldsymbol{\Sigma}^{1/2}(X_t^n) dY_t^n) + \frac{1}{2} \partial^2 \Phi(X_t^n)[\mathrm{d}[X^n]_t]. \tag{69}
$$

By Lemma D.21 we know the process $X_t^n - \Phi_t^n$ weakly converges to zero. By Lemma D.18 we know $\delta_t \to 0$. By Lemmas D.24 to D.26 we know

$$
\left| \int_0^{t \wedge \tau_n} \partial \Phi(X_t^n)((-\lambda_t \eta + \lambda_t \eta^{1-\alpha}/\gamma_t) \mathrm{d}M_t^n + \frac{1}{2} \partial^2 \Phi(X_t^n)[\mathrm{d}[X^n]_t] \right.
$$
$$
\left. - \int_0^{t \wedge \tau_n} \frac{\lambda_t^2}{2} \partial^2 \Phi(X_s^n)[\Sigma(X_s^n)] \mathrm{d}[Y^n]_s \right| \to 0.
$$

We know the process $Y_t^n$, $Z_t^n$ and $[Y^n]_t$ are of bounded quadratic variation, so are $\widetilde{Y}_t^n = Y_t^n - \sum_{s \le t} h_\delta(|\Delta Y_s^n|) \Delta Y_s^n$, $\widetilde{Z}_t^n = Z_t^n - \sum_{s \le t} h_\delta(|\Delta Z_s^n|) \Delta Z_s^n$ and $[\widetilde{Y}^n]_t = [Y^n]_t - \sum_{s \le t} h_\delta(|\bar{\Delta}[Y^n]_s|) \Delta[Y^n]_s$. Furthermore by the Donsker's theorem $Y_t^n \to W_t$ in the uniform metric, where $W_t$ is a Brownian motion, and $Z_t^n \to t$. By the law of large numbers we know

$[Y^n]_t \to t$. Additionally, the process $X_t^n$, $Z_t^n$ and $Y_t^n$ always share jumps at the same locations. This implies we can write

$$X_t^n = X_0 + P_{t \wedge \tau_n}^n + \int_0^t F_n(X_s^n)\mathrm{d}Y_s^n + G_n(X_s^n)\mathrm{d}[Y^n]_s + H_n(X_s^n)\mathrm{d}Z_s^n$$

for the functions $F_n, G_n$ and $H_n$ that describe the iterates Equation (68) and

$$P_t^n = (X_t^n - \Phi_t^n) + \delta_t^n +$$
$$\int_0^t \partial\Phi(X_t^n)((-\lambda_t\eta + \lambda_t\eta^{1-\alpha}/\gamma_t)\mathrm{d}M_t^n + \frac{1}{2}\partial^2\Phi(X_t^n)[\mathrm{d}[X^n]_t] - \frac{\lambda_t^2}{2}\partial^2\Phi(X_s^n)[\mathbf{\Sigma}(X_s^n)]\mathrm{d}[Y^n]_s.$$

Notice that the stopping time is incorporated into the functions that $F_n(x) = G_n(x) = H_n(x) = 0$ for $x \notin K$. Notice that the processes $(P_t^n, Y_t^n, [Y^n]_t, Z_t^n, F_n(X_t), G_n(X_t), H_n(X_t))$ converges in the uniform metric to $(0, W_t, t, t, F(X_t), G(X_t), H(X_t))$ for any process $X_t$, then by Theorem D.5, the limit of $X_t^n$ can be denoted by

$$X_t = X_0 + \int_0^t F(X_s)\mathrm{d}W_s + G(X_s)\mathrm{d}s + H(X_s)\mathrm{d}s.$$

Plugging in the above results gives the limit

$$X_t = \Phi(\boldsymbol{x}_0) + \int_0^t \lambda_t\partial\Phi(X_s)\mathbf{\Sigma}^{1/2}(X_s)\mathrm{d}W_s + \int_0^t \frac{\lambda_t^2}{2}\partial^2\Phi(X_s)[\mathbf{\Sigma}(X_s)]\mathrm{d}s.$$

$\square$

*Proof for Theorem D.6.* The result is a natural corollary of Theorem D.7 and Theorem D.27. $\square$

# E EXPERIMENTAL DETAILS

## E.1 LANGUAGE MODEL FINE-TUNING

We fine-tune RoBERTa-large (Liu et al., 2019) on several tasks using the code provided by Malladi et al. (2023). We are interested in comparing SGD and SGDM, so we use a coarser version of the learning rate grid proposed for fine-tuning masked langauge models with SGD in (Malladi et al., 2023). We randomly sample 512 examples per class using 5 seeds, following the many shot setting in Gao et al. (2021). Then, we fine-tune for 4 epochs with batch sizes $2, 4$, and $8$ and learning rates $1e-4, 1e-3$, and $1e-2$. We select the setting with the best dev set performance and report its performance on a fixed 1000 test examples subsampled from the full test set, which follows (Malladi et al., 2023). We also compare the trajectories of SGDM and SGD when fixing the data seed and see that the two closely track each other (see Figure 3) on five different tasks.

Past work (Malladi et al., 2023) has suggested that using a suitable prompt can fundamentally modify the dynamics of fine-tuning, so we also report results when using a suitable prompt. Prompts are taken from Gao et al. (2021). Results in Table 2 demonstrate that SGD and SGDM track each other closely in the prompt-based fine-tuning setting as well.

| Task | SST-2 | SST-5 | SNLI | TREC | MNLI |
|------|-------|-------|------|------|------|
| Zero-shot | 79.0 | 35.5 | 50.2 | 51.4 | 48.8 |
| SGD | 93.1 (0.9) | 54.9 (0.7) | 87.9 (0.8) | 97.0 (0.2) | 82.4 (1.4) |
| SGDM | 93.1 (0.4) | 55.3 (0.9) | 87.3 (0.5) | 96.8 (0.7) | 82.8 (1.4) |

Table 2: SGD and SGDM for fine-tuning RoBERTa-large on 5 tasks using 512 examples from each class (Gao et al., 2021; Malladi et al., 2023). We use the simple task-specific prompt introduced in Gao et al. (2021). Results are averaged over 5 random subsets of the full dataset. These findings confirm that SGD and SGDM approximate each other in noisy settings.

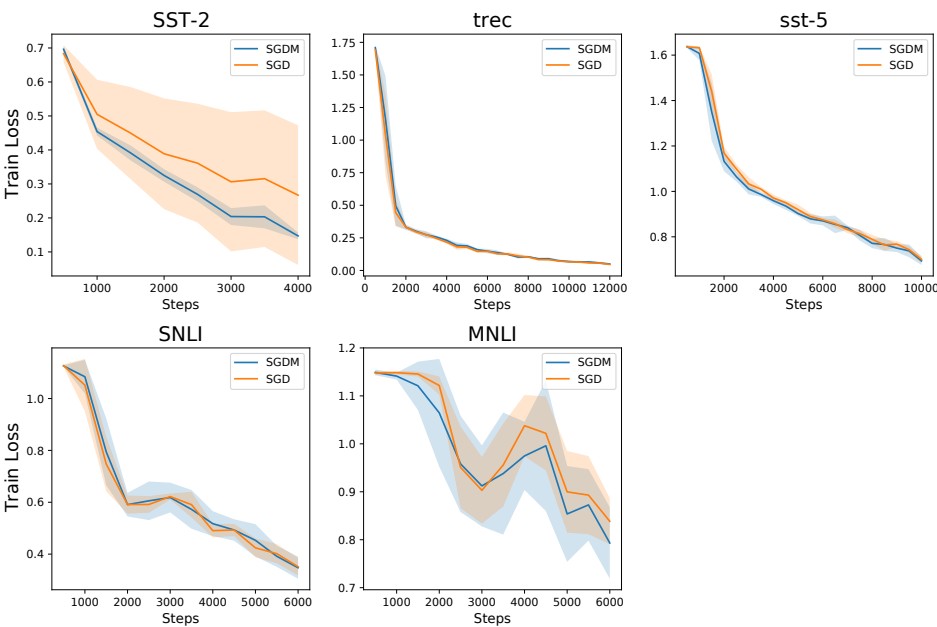

Figure 3: SGD and SGDM trajectories when fine-tuning RoBERTa-large on five downstream tasks. We ensure the effective learning rate is fixed in both cases, so SGDM trajectories are with learning rate 0.001 and SGD trajectories are with learning rate 0.01. We fix the data seed and fine-tune using five different optimization seeds. The results show that SGD and SGDM track each other closely on average over the course of language model fine-tuning.

## E.2 CIFAR-10 EXPERIMENTS

We report the results of training ResNet-32 on CIFAR-10 with and without momentum. First, we grid search over learning rates between $0.1$ and $12.8$, multiplying by factors of 2, to find the best learning rate for SGD with momentum. We find this to be $\eta = 0.2$. Then, we run SGD and SGDM with SVAG to produce the figure. We obey the SGDM formulation Definition 2.3 by setting $\gamma = \frac{\eta}{1-\beta}$ with $\beta = 0.9$.

Standard SGD and SGDM exhibit different test accuracies ($89.35\%$ vs $92.68\%$), suggesting that momentum exhibits a different implicit regularization than SGD. However, as we increase the gradient noise by increasing $\ell$ in SVAG (Definition 5.1), we see that the two trajectories get closer. At $\ell = 4$, the final test accuracies for SGD and SGDM are $91.96\%$ and $91.94\%$, respectively, verifying our theoretical analysis.

