# OpenReview forum: "The Marginal Value of Momentum for Small Learning Rate SGD"
_ICLR.cc/2024/Conference — ICLR 2024 poster_

### Official Review · Reviewer_EHVB · 2023-10-27

**Soundness:** 2 fair
**Presentation:** 3 good
**Contribution:** 2 fair
**Rating:** 5
**Confidence:** 3

**Summary:**

This paper considers the role of momentum in SGD training. When the learning rate is small but the gradient noise is not small, the paper shows that the trajectories of SGD and SGDM are close. Therefore, the momentum component in SGDM adds only a small effect on top of SGD. This paper also conducts experiments to verify that the optimization and generalization of SGDM and SGD are close when the learning rate is small and the mini-batch size is not large (such that the gradient noise is not small).

**Strengths:**

Theoretically, momentum provably improves the (minimax) convergence rate of GD for (strongly convex) smooth optimization. Practically, momentum is often used with SGD (as GD is expensive in practical machine learning optimization). In the presence of gradient noise, will momentum still offer benefits? This is the central question studied in this work.

First, this work presents a theory, showing that when the learning rate tends to zero, the trajectories of SGD and SGDM are close in two regimes. The first regime is when the gradient noise is large (with $O(1/\eta)$ variance) but the total running distance is constant. The second regime allows the total running steps to be large but requires an overparameterized model where the global solutions form a manifold.

Second, this work conducts experiments to verify that for both training and test, SGDM and SGD are close when the learning rate is small and when the batch size is not too large.

The writing is clear overall. Some places require clarifications. See more in the next section.

I did not check the proof.

**Weaknesses:**

1. I feel the introduction/motivation could use some polishment. Specifically, the benefit of momentum is for stabilizing GD with a large learning rate ($>2/L$, in terms of improving the minimax convergence rate for (strongly convex) smooth optimization) as discussed in the first paragraph of the introduction. However, I did not recall a theoretical result that claims acceleration of momentum for GD with a learning rate.

One of the main contributions of this paper is to show that momentum does not change SGD trajectory much when the learning rate is sufficiently small. Therefore, momentum is not very helpful.

I am not sure how surprising this is, since the benefit of momentum is only claimed for GD with a very large learning rate.

Also, the condition that $1-\\beta\^{(n)}_k = \\Theta ((\\eta^{(n)})\^{\\alpha} )$ (in definition 3.1) for $\alpha \in [0,1)$ (in Theorem 3.5) basically says that the history gradient will be forgotten fast (in a nearly exponential rate) when SGDM runs $\\Theta(1/\\eta)$ steps. So it's not very surprising that SGDM will be close to SGD.


2. In the last paragraph of Section 3, the paper briefly mentions a phenomenon for $\\alpha>1$. However, the theorem only covers $\\alpha <1$. Could the author elaborate on what happens to SGDM when $\\alpha>1$?


3. In the paragraph after Lemma 2.4, $\\sigma$ is set to be $1/\sqrt{\eta}$ to ensure a non-trivial gradient noise scaling. However, later in Theorem 3.5, it allows that $\\sigma \le \eta^{-1/2}$, and in Theorem 4.5, it allows $\\alpha = \\Theta(1)$. These seem to be different from the prior discussions. Could the authors clarify the scaling of the noise?  In particular, can Theorem 3.5 be applied to GD and GDM (i.e., $\\sigma = 0$)?


4. This paper mentions two versions of SGDM, that is, eq (1) used in practice and eq (3) used in theory. I appreciate the usage of $\\gamma$ and $\\eta$ to differentiate the learning rate for those two versions, which helps highlight the difference in the effective optimization length.

In the ImageNet experiment, the learning rate for eq (1) is manually rescaled when compared to SGD (to align with the theory setup in eq (3)), which is good. However, for training ResNet-32 on CIFAR-10, the paper writes that "We first grid search to find the best learning
rate for the standard SGDM ($\\ell=1$), and then we perform SGD and SGDM with that same learning rate for different levels of $\\ell$". I am wondering if the learning rate for SGD needs a rescaling here.

**Questions:**

See above.

---

> ### Author Response · Authors · 2023-11-21
>
> Thank you for your comments.
>
> ---
>
> **Q1:** I did not recall a theoretical result that claims acceleration of momentum for GD with a learning rate.
>
> **A1:** The result quoted by the introduction comes from [1] (Theorem 9). Specifically it shows that the heavy-ball momentum with optimal hyperparameter constants enjoys a faster convergence rate than GD in the neighborhood of a local minimizer, and when the hessian is ill-conditioned the optimal learning rate with momentum can exceed $2/L$.
>
> - [1] B.T. Polyak. Some methods of speeding up the convergence of iteration methods. USSR Computational Mathematics and Mathematical Physics, 4(5):1–17, 1964. ISSN 0041-5553.
>
> ---
>
> **Q2:** Is it surprising that momentum does not help when the LR is sufficiently small? The benefit of momentum over standard GD has only been claimed in the large LR regime
>
> **A2:** There are works that claim the role of momentum (for instance [2]) for GD in terms of the implicit regularization in the setting where the LR is sufficiently small, and we are writing to further show that when there are enough gradient noises, then adding momentum does not help implicit regularization.
>
> - [2] Avrajit Ghosh, He Lyu, Xitong Zhang, and Rongrong Wang. Implicit regularization in heavyball momentum accelerated stochastic gradient descent. In The Eleventh International Conference on Learning Representations, 2023. URL https://openreview.net/forum?id= ZzdBhtEH9yB
>
> ---
>
> **Q3:** The condition on $1-\beta$ in Definition 3.1 says that the past gradient will be quickly forgotten, so it is not surprising that SGDM is close to SGD in that case. What happens when $\alpha > 1$?
>
> **A3:** When $\alpha>1$, there will be no limiting dynamics as $\eta\to 0$.  For any non-zero initial momentum, the parameter will follow the initial momentum up to the scale $O(\eta^{1-\alpha})$, so when $\eta$ is small enough the trajectory will tend to explode as the parameters go unbounded. In general when $\alpha>1$, it takes longer time for the momentum to adapt to local curvature than for the parameter to go out of local areas, therefore conceptually such a momentum choice makes optimization uncontrollable.
>
> Even when the initial momentum is zero and the initial point is near a strongly-convex local minimizer, the asymptotic convergence rate will still be $O(1/\eta^{(1+\alpha)/2})$ for $\alpha>1$, strictly slower than the rate $O(1/\eta)$ for the case when $\alpha<1$, which is not preferred for optimization.
>
> Empirically, setting $\alpha>1$ at LR annealing also produces very bad performance. We believe the above are the reasons.
>
> ---
>
> **Q4:** How does the scaling of the noise work? Lemma 2.4 and Theorem 3.5 appear to contradict. Can Theorem 3.5 be applied to full-batch GD with and without momentum?
>
> **A4:** Yes Theorem 3.5 can be applied to full-batch GD where $\sigma = 0$. We write $\sigma\leq \eta^{-1/2}$ to make the theorem more general, but the most interesting case is $\sigma= \eta^{-1/2}$. As Lemma 2.4 shows, when $\sigma= \eta^{-1/2}$, the gradient noise has non-negligible impact on the loss curve, while when $\sigma\ll \eta^{-1/2}$, the impact of the gradient noise is negligible and SGD will have a similar trajectory as GD.
>
> We realized that the paragraph after Lemma 2.4 may cause confusion, as our results actually hold for a range of $\sigma$ as stated in the theorems. We will fix the confusion in the revision, and thank you for the suggestion.
>
> ---
>
> **Q5:** Does the learning rate for SGD need a rescaling in the CIFAR-10 experiments?
>
> **A5:** Yes in all the experiments the learning rate needs to be rescaled so that $\eta$ matches for SGD and SGDM (or $\eta = \gamma/(1-\beta)$). We will add that into the experiment details in the revision.

---

> > ### Comment · Reviewer_EHVB · 2023-11-22
> > **Follow up questions**
> >
> > Thank you for the comments.
> >
> > Q1. I was trying to say that "I did not recall a theoretical result that claims acceleration of momentum for GD with a **small** learning rate." Sorry for missing "small" in my initial comment.
> >
> > As the acceleration of momentum is only justified for GD with a large learning rate ($>2/L$), I find it not surprising that momentum does not contribute when the learning rate is small.
> >
> >
> > Q2. Thank you for pointing out [2]. As Theorem 3.5 can be applied to GD by taking $\\sigma\^2 = 0$, would Theorem 3.5 contradict the message in [2] in this case? Have I misunderstood anything?
> >
> >
> > Q5. Could you please clarify whether or not the learning rate has been properly rescaled in the current experiments? Would some of the current experiments need to be redone for a fair comparison?

---

> > > ### Author Response · Authors · 2023-11-23
> > >
> > > Thanks for your clarification on Q1.
> > >
> > > For Q2, there is no contradiction between our Theorem 3.5 and the main result in [2], because we are saying Momentum SGD (in this case Momentum GD) is a first-order approximation to GD and their result says momentum GD is a second-approximation to a gradient flow of modified loss. However, the gradient flow of the modified loss itself is a first-order approximation to the gradient flow of the original loss. The first-order implicit regularization of the modified loss is only significant when the learning rate is large, e.g., when running full-batch GD. In the main scenario that we are interested in, where the noise covariance scales up in the order $\sigma^2\sim 1/\eta$ and $\eta$ is very small, the first-order implicit regularization due to momentum is much weaker than the implicit regularization brought by noise, which does not vanish as learning rate $\eta$ goes to 0.
> > >
> > > For Q5, we confirm that all the learning rate has been properly rescaled and no current experiments need to be rerun. Thanks for asking.

---

### Official Review · Reviewer_fSFX · 2023-10-31

**Soundness:** 3 good
**Presentation:** 3 good
**Contribution:** 3 good
**Rating:** 6
**Confidence:** 3

**Summary:**

This paper aims to investigate the advantages of incorporating momentum in the Stochastic Gradient Descent with Momentum (SGDM) method for mitigating the variance associated with stochastic gradients. To mitigate oscillations along the high-curvature direction, the authors make the assumption that the learning rate is small. Within this context, they demonstrate that, subject to specific assumptions, Stochastic Gradient Descent (SGD) and SGDM yield identical training error results. Additionally, similar to previous studies, they construct a Stochastic Differential Equation (SDE) for the continuous version of SGDM/SGD and, through an analysis of the SDE's limiting behavior, reveal that both SGD and SGDM exhibit a similar implicit bias.

**Strengths:**

Prior research has noted that the advantage of incorporating momentum in the training of neural networks is somewhat limited, and there hasn't been a definitive theoretical outcome regarding the effectiveness of momentum in the stochastic setting. Therefore, demonstrating that momentum does not significantly reduce variance or improve generalization can lead to savings in computational resources and memory usage during deep neural network training. A noteworthy technical innovation in this work is the introduction of a novel Stochastic Differential Equation (SDE) to represent the diminishing learning rate in Stochastic Gradient Descent with Momentum (SGDM) and Stochastic Gradient Descent (SGD).

**Weaknesses:**

The assumptions that the paper considers to show that SGDM approximate SGD is too restrictive. It would be great if they can justify these too restrictive assumptions.

**Questions:**

In Theorem 3.5, the paper demonstrates that the trajectory of Stochastic Gradient Descent with Momentum (SGDM) approximates that of Stochastic Gradient Descent (SGD) with a specific learning rate. While it may be somewhat expected that these two methods, both being first-order optimization techniques, would exhibit some similarity, the importance of this result lies in its ability to establish a specific condition under which SGDM behaves similarly to SGD.

However, as you rightly suggest, a more generalized result would be valuable. If the paper could establish conditions where, for certain small learning rates, SGDM and SGD consistently approximate each other, it would offer broader insights into the relationship between these methods and provide more guidance on when to expect their similarities. This would enhance our understanding of the interplay between learning rates and optimization algorithms, potentially leading to more informed choices in practical applications.

---

> ### Author Response · Authors · 2023-11-21
>
> Thank you for your comments.
>
> ---
>
> **Q1:** The assumptions that the paper considers to show that SGDM approximate SGD is too restrictive. It would be great if they can justify these too restrictive assumptions.
>
> **A1:** The assumptions we make are standard from a series prior works [1-4] on the trajectory analysis of stochastic updates, and theoretical findings from those works have been predictive of complex real-world settings (e.g., training vision models and fine-tuning language models, see [2,5] for instance).
>
> Specifically, for training smooth loss on a neural network with smooth activations (Swish, sigmoid, GeLU etc) and weight decay, our assumptions will hold. In practical settings, all the data points and networks weights are actually mathematically bounded for a fixed number of steps, and the corresponding assumptions mostly serve for mathematical rigor.
>
> References:
> - [1] Qianxiao Li, Cheng Tai, and Weinan E. Stochastic modified equations and dynamics of stochastic gradient algorithms i: Mathematical foundations. Journal of Machine Learning Research, 20(40): 1–47, 2019.
> - [2] Zhiyuan Li, Sadhika Malladi, and Sanjeev Arora. On the validity of modeling SGD with stochastic differential equations (SDEs). Advances in Neural Information Processing Systems, 34:12712– 12725, 2021a.
> - [3] Li, Zhiyuan, Tianhao Wang, and Sanjeev Arora. "What Happens after SGD Reaches Zero Loss?--A Mathematical Framework." In International Conference on Learning Representations. 2021.
> - [4] Sadhika Malladi, Kaifeng Lyu, Abhishek Panigrahi, and Sanjeev Arora. On the SDEs and scaling rules for adaptive gradient algorithms. Advances in Neural Information Processing Systems, 2022.
> - [5]  Elkabetz, Omer, and Nadav Cohen. "Continuous vs. discrete optimization of deep neural networks." Advances in Neural Information Processing Systems 34 (2021): 4947-4960. Blog http://www.offconvex.org/2022/01/06/gf-gd/.

---

### Official Review · Reviewer_RZuq · 2023-11-01

**Soundness:** 3 good
**Presentation:** 3 good
**Contribution:** 3 good
**Rating:** 6
**Confidence:** 3

**Summary:**

This work compares SGDM and SGD in the setting of small learning rate and gradient noise is large enough to produce instability. Two main results are presented, both indicating that the training trajectory of SDGM is close to that of SGD. The first result states that within $O(1/\eta)$ steps of training, the trajectories of SGD and SGDM approximate each other with distance $O(\sqrt{\eta/(1-\beta)})$. The second result states that within $O(1/\eta^2)$ steps of training, the trajectories of both SGD and SGDM move slowly after reaching a manifold where the local minimizer is located.

**Strengths:**

1. The main results, i.e., Theorem 3.5 and Theorem 4.5, are strong in terms of both implications and proof techniques.
2. The results are presented in an orderly manner.
3. The result that the training trajectories of SGDM and SGD are similar is intriguing.

**Weaknesses:**

1. Although the implications of Theorem 3.5 is clear, the statement of the theorem is a bit confusing. Specifically, the averaged learning schedule $\bar\eta_k$ is introduced, and it does not appear again until the appendix.
2. It would be interesting to see other concrete types of hyperparameter schedule apart from the constant ones, both theoretically and empirically.
3. It would also be interesting to see experiments demonstrating the main theoretical contribution of this paper, i.e., the trajectories rather than the loss / accuracy stay close.

**Questions:**

1. Do the two regimes, i.e. $t=O(1/\eta)$ and $t=O(1/\eta^2)$ have any connections? Can we understand from the results provided in this work what's going on when $t=\Omega(1/\eta)$ but $t=o(1/\eta^2)$?
2. Could the authors clarify why a **class** of hyperparameter schedule $\{\eta_k^{(n)}, \beta_k^{(n)}\}$, indexed by $n$, is considered in both main theorems? It seems more straightforward to me to consider only one choice of hyperparameter schedule.

---

> ### Author Response · Authors · 2023-11-21
>
> Thank you for your comments.
>
> ---
>
> **Q1:** Although the implications of Theorem 3.5 is clear, the statement of the theorem is a bit confusing. Specifically, the averaged learning schedule is introduced, and it does not appear again until the appendix.
>
> **A1:** The averaged learning rate schedule is defined in our theorem as the formula in the parentheses directly followed. We will find a better way to introduce this notion in the revision to eliminate the confusion.
>
> ---
>
> **Q2:** It would be interesting to see other concrete types of hyperparameter schedule apart from the constant ones, both theoretically and empirically. It would also be interesting to see experiments demonstrating the main theoretical contribution of this paper, i.e., the trajectories rather than the loss / accuracy stay close.
>
> **A2:** Our theory is proved for a general hyperparameter scheduling (as long as its scale is controlled), so the conclusions hold for non-constant schedules. Also, for the experiments, we used the 3-stage annealing learning rate schedule and added a linear warm-up, which is technically not constant. In the future, we can surely provide more experiments with more varying hyperparameter schedules.
>
> Our theory prescribes that the trajectories are close according to any test function (Definition 3.4), and we used training and test curves which are the most empirically concerned test functions. In the future we can add more test functions to show the closeness of the trajectory distributions.
>
> ---
>
> **Q3:** Can we understand what’s going on when $t=\Omega(1/\eta)$ and $t=O(1/\eta^2)$? That is, between the two regimes?
>
> **A3:** The two regimes are separate in our results because they entail different assumptions on the landscape. The $\Omega(1/\eta)$-step result holds for general smooth landscapes, while for the trajectory to be tractable for $O(1/\eta^2)$ steps, we need stronger assumptions on the landscape (i.e. the existence of a minimizer manifold). The two regimes also have different levels of tolerance of the noise scale $\sigma$.
>
> When the $O(1/\eta^2)$-trajectory is tractable, the first $O(1/\eta)$ steps in the trajectory actually follows an SGD trajectory that decrease the loss from initial to almost zero. At the end of the first $O(1/\eta)$ steps, the parameter will be $o(1)$ close to the minimizer manifold, and it stays in the neighborhood of the manifold for $O(1/\eta^2)$ steps and do a $O(\eta)$ progression along the manifold every $O(1/\eta)$ steps.
>
> ---
>
> **Q4:** Why is a class of hyperparameter schedules used in the main theorems?
>
> **A4:** For a specific hyperparameter scheduling, we can indeed use our theorem to bound the distance between SGDM and SGD trajectories.
>
> We write a class of schedules in our theorem to make the theorem more general and to emphasize the dependence of our result on the scale of $\eta$, e.g. when $\eta\to 0$ the trajectories indeed have limiting distributions at a rate $O(\eta^{(1-\alpha)/2})$ for some limiting hyperparameter schedule.

---

### Official Review · Reviewer_ReS6 · 2023-11-03

**Soundness:** 3 good
**Presentation:** 2 fair
**Contribution:** 2 fair
**Rating:** 5
**Confidence:** 3

**Summary:**

The authors have proved theoretically the marginal benefit of momentum in training when the learning rate is small and the noise is dominant. The theory has been verified with some experiments.

**Strengths:**

1. The paper has demonstrated the limited benefit of momentum theoretically in some regimes, which helps suggest when not to use momentum.
2. The article has illustrated the idea with warmup examples, which makes it easier to digest.

**Weaknesses:**

1. The analysis is based on the fact that the momentum effect is dominated by other factors when the learning rate is negligible. This is somewhat intuitive and needs to dive deeper into what the special role of momentum is.

2. There needs to be clear examples of what scenarios the theory fits in. For example, with what class of loss and neural network does the theory fit in?

3. The paper seems rushed in polishing. There are some links of reference in the paper that need to be added, for example, in line 2 on page 8.

**Questions:**

1. Could you show some examples of what kind of networks, loss..etc the theory fits in.
2. It will be nice to make more clear comparison of when with and without momentum.

**Details Of Ethics Concerns:**

No ethics concern

---

> ### Author Response · Authors · 2023-11-21
>
> Thank you for your comments.
>
> ---
>
> **Q1:** The analysis is based on the fact that the momentum effect is dominated by other factors when the learning rate is negligible. This is somewhat intuitive and needs to dive deeper into what the special role of momentum is.
>
> **A1:** We are not sure about what is excluded from “other factors”, and we appreciate it if more clarifications are provided.
>
> The main message we wish to convey through this paper is that momentum has no special role when the learning rates are small, and tuning the momentum parameter has very limited benefits as model performance is agnostic to the choice of momentum parameters over a range of hyperparameter scales. This conclusion is supported by both our theoretical and empirical findings.
>
> For some clarifications of our results, first, we believe that it is incorrect to claim that the momentum effect is dominated by changing the curvature-induced instability, rather than changing the noise-induced instability (defined in Lemma 2.4). Adding momentum actually modifies the noise part of parameter updates more than the curvature part. Second, simple intuitions like Section 3.1 cannot offer any suggestions on the number of steps when adding momentum accumulates non-negligible deviations. For instance, as our setting involves a $O(\sqrt{\eta})$-normed update per step, two adjacent SGD updates may differ by $O(\sqrt{\eta})$, therefore after $O(1/\sqrt{\eta})$ steps an SGDM trajectory will deviate from an SGD trajectory by the above intuition, which only accounts for $O(\sqrt{\eta})$ training loss decrease. Therefore even when the learning rate is small enough, it is not trivial to conclude that momentum has a negligible effect over the whole trajectory of $O(1/\eta)$ or $O(1/\eta^2)$ steps, when the training loss goes from initial to almost optimal and plateaus.
>
> ---
>
> **Q2**: There needs to be clear examples of what scenarios the theory fits in. For example, with what class of loss and neural network does the theory fit in?
>
> **A2:** For a short clarification, as our theory is a proof of the general non-convex landscape, it aims to model model training with any smooth loss, any neural networks architecture with any smooth activations (with polynomial growth) and small learning rates. For instance training with square loss for regression or logistic loss for classification falls into the assumptions. The boundedness of noise covariance and hessian are satisfied when the weights of the network have bounded moments during the course of the training, which is satisfied when weight decay is involved for general neural networks.
>
> The assumptions of the paper are standard in a series of trajectory analysis works [1-4].
>
> References
> - [1] Qianxiao Li, Cheng Tai, and Weinan E. Stochastic modified equations and dynamics of stochastic gradient algorithms i: Mathematical foundations. Journal of Machine Learning Research, 20(40): 1–47, 2019.
> - [2] Zhiyuan Li, Sadhika Malladi, and Sanjeev Arora. On the validity of modeling SGD with stochastic differential equations (SDEs). Advances in Neural Information Processing Systems, 34:12712– 12725, 2021a.
> - [3] Li, Zhiyuan, Tianhao Wang, and Sanjeev Arora. "What Happens after SGD Reaches Zero Loss?--A Mathematical Framework." In International Conference on Learning Representations. 2021.
> - [4] Sadhika Malladi, Kaifeng Lyu, Abhishek Panigrahi, and Sanjeev Arora. On the SDEs and scaling rules for adaptive gradient algorithms. Advances in Neural Information Processing Systems, 2022.
>
> Furthermore, our experiment results further corroborate that our conclusion holds for general small-batch training from image tasks to natural language tasks, even when the learning rate is large. In practice, this implies that model performance is agnostic to the choice of momentum parameters when the batch sizes are scaled down.

---

### Meta-Review · Area_Chair_1kP1 · 2023-12-22

**Metareview:**

The paper analyses heavy ball momentum's trajectory and shows that in the small learning rate regime where the main source of instability in training is arising from the noise in the stochastic gradient oracle, the trajectories of sgd and agd with momentum tend to approximate each other closely. The practical understanding of this result is that in low batch regimes sgd and sgdm tend to perform similarly. The result is established in two regimes for 1/eta steps and for 1/eta^2. The latter regime emerges with motivation from recent works where there is an assumption of a manifold of minimizers and the optimization is assumed to have converged to a minimum. In both settings (can be bradly thought of as optimization and generalization respectively) it can be seen that momentum and sgd closely approximate each other as long as the noise scaling is a certain function of the learning rate.

Overall, the paper makes a nice theoretical contribution of capturing the ineeficacy of momentum in small learning rate regime. The thesis of the result is not surprising and has reported several times before. Howerver the trajectory analysis tends to shed a stronger light on this problem. I read the paper ,yself and found the writing to be confusing in multiple parts. Especially the parts where schedules are introduced around the main theorems. I understand they add generality to the result but I believe the presentation in terms of notation etc can be improved there. Experimental results support the theory but to my knowledge they are not necessarily 'new'.

**Justification For Why Not Higher Score:**

The reviewers had mixed reviews towards the paper however no reviewer provided a score higher than marginal accept.

**Justification For Why Not Lower Score:**

The paper is a borderline paper given the reviews and my understanding of the paper. The analysis adds a solid base to a somewhat understood/believed concept but is not very surprising and did not particularly add new tools to the space (imo primarily builds on  line of work from Li et al and Malladi et al).

---

### Decision · Program_Chairs · 2024-01-16

Accept (poster)